# Direct Alcoholysis of Carbohydrate Precursors and Real Cellulosic Biomasses to Alkyl Levulinates: A Critical Review

**Anna Maria Raspolli Galletti** *[ID]**, Claudia Antonetti**[ID]**, Sara Fulignati**[ID] **and Domenico Licursi**[ID]

Department of Chemistry and Industrial Chemistry, University of Pisa, Via Giuseppe Moruzzi 13, 56124 Pisa, Italy; claudia.antonetti@unipi.it (C.A.); sara.fulignati@for.unipi.it (S.F.); domenico.licursi@unipi.it (D.L.)
* Correspondence: anna.maria.raspolli.galletti@unipi.it; Tel.: +39-50-2219290

**Abstract:** Alkyl levulinates (ALs) represent outstanding bio-fuels and strategic bio-products within the context of the marketing of levulinic acid derivatives. However, their synthesis by acid-catalyzed esterification of pure levulinic acid, or by acid-catalyzed alcoholysis of furfuryl alcohol, although relatively simple, is still economically disadvantageous, due to the high costs of the pure precursors. The direct one-pot alcoholysis of model C6 carbohydrates and raw biomass represents an alternative approach for the one-step synthesis of ALs. In order to promote the market for these bio-products and, concurrently, the immediate development of new applications, it is necessary to speed up the intensification of their production processes, and this important achievement is onlypossible by using low-cost or, even better, waste biomasses, as starting feedstocks. This review provides an overview of the most recent and promising advances on the one-pot production of ALs from model C6 carbohydrates and real biomasses, in the presence of homogeneous or heterogeneous acid catalysts. The use of model C6 carbohydrates allows for the identification of the best obtainable ALs yields, resulting in being strategic for the development of new smart catalysts, whose chemical properties must be properly tuned, taking into account the involved reaction mechanism. On the other hand, the transition to the real biomass now represents a necessary choice for allowing the next ALs production on a larger scale. The improvement of the available synthetic strategies, the use of raw materials and the development of new applications for ALs will contribute to develop more intensified, greener, and sustainable processes.

**Keywords:** alkyl levulinates; one-pot alcoholysis; solvothermal processes; levulinic acid; bio-fuels; process intensification

---

## 1. Introduction

Levulinic acid (4-oxopentanoic acid, gamma ketovaleric acid or 3-acetylpropionic acid) is a linear C5-alkyl carbon chain, which is considered to be one of the top 12 most promising chemicals derived from biomass [1–4]. Unfortunately, this platform chemical can be scarcely used for immediate uses, mainly requiring needful upgrading step(s) in order to obtain marketable bio-products of higher added-value, such as bio-fuel additives, fragrances, solvents, pharmaceuticals, plasticizers, and other polymers [5–7]. Therefore, the growth of the levulinic acid market needs the possibility of synthesizing its derivatives by the simplest synthetic strategies, thus preferring the development of the process intensification in the short term. Among these possible derivatives, alkyl levulinates (ALs) fully satisfy these requirements, in principle being easily synthesized by acid-catalyzed esterification of pure levulinic acid [8], according to the reaction that is reported in Figure 1.

**Figure 1.** AL synthesis by acid-catalized esterification of levulinic acid.

The choice of this class of levulinic acid derivatives as starting feedstock for further up-grade is very attractive, due to the moderate reactivity of the carboxylate group, leading to remarkable advantages for the selectivity towards the final product(s) of interest. This is the reason why some levulinic acid derivatives of increasing industrial interest, such as ketals [9], alkoxypentanoates [10], γ-valerolactone, 2-methyltetrahydrofuran, valeric acid/alkyl valerates, 1,4-pentanediol, and N-substituted pyrrolidinones [11,12], are preferably produced, starting from AL, rather than from levulinic acid. As an additional noteworthy advantage, esters have a lower boiling point than the corresponding carboxylic acid, thus allowing for their easier and cheaper separation/purification by distillation [13]. Moreover, integrating the synthesis and recovery of the ester starting from levulinic acid is a concrete possibility, achieved by reactive distillation, which improves the overall economy of the process and favors its intensification [14]. Regarding the economic evaluations, up to now, the market attention has mainly been focused on ethyl levulinate (EL) production, which is a valuable fuel-additive and a potential biomass-derived platform molecule [15], having a global market size of $ 10.5 Million in 2019, favorably increasing up to about $ 11.8 Million by 2022, with an estimated growth rate of 3.6% [16]. In this context, other independent economic evaluations have confirmed a promising economic outlook for the EL production [17], but only if low-cost or, even better, waste biomasses, will be used as starting feedstocks [2].

## 1.1. Reaction Mechanisms

The syntheses of ALs and levulinic acid occur with similar chemistry, but the chemical structures of the reaction intermediates that are obtained from C6 feedstocks are different, thus expanding the possible developable applications [18–21] (Figure 2).

**Figure 2.** Comparison between the C6 routes for the production of levulinic acid and alkyl levulinate (AL).

The hydrothermal process follows a series of acid-catalyzed hydrolysis/dehydration steps:(i)hydrolysis of cellulose to glucose, (ii)next dehydration of glucose to 5-hydroxymethylfurfural, and (iii) hydrolysis of 5-hydroxymethylfurfural to levulinic acid. Instead, in the alcohol medium, the overall reaction occurs by:(i) cellulose alcoholysis to the alkyl glucoside, (ii) dehydration of the alkyl glucoside to 5-alkoxymethyfurfural, and(iii)final alcoholization of the 5-alkoxymethyfurfural to give equimolar amounts of AL and alkyl formate [18]. However, Figure 2 also shows that the synthesis of ALs can be advantageously performed, starting from different reaction intermediates

derived from the hydrothermal process, proving the good flexibility of these solvothermal processes, depending on the market availability and the costs of the precursors. The main obstacle to the efficient use of 5-hydroxymethylfurfural for AL production is its high reactivity, which makes it prone to fast degradation. To solve this drawback, a possible solution involves 5-chlorometylfurfural production from C6 carbohydrates in a hydrogen chloride-saturated organic phase [22]. This furanic derivative is more stable and hydrophobic than 5-hydroxymethylfurfural, thus allowing for its quicker simultaneous extraction into the organic phase of a biphasic systemand making its isolation easier and more efficient. Additionally, the commercial production of 5-hydroxymethylfurfural mainly provides the use of fructose as the starting feedstock, which is more expensive than raw cellulosic biomass, whilst the production of 5-chlorometylfurfural directly from raw biomass is more feasible [23,24]. Based on these requirements, the choice of the real biomass as starting feedstock should be the most appropriate for ALs production, significantly improving the process intensification, but this goal is more difficult to achieve [25]. Starting from backward C6 precursors, and particularly from real biomasses, the issue of the by-product formation becomes of paramount importance, consuming carbon atoms and significantly worsening the selectivity to the desired AL, with complex product separation and significantly increased process cost. It is necessary to properly tune the main reaction conditions and the properties of the adopted catalysts, in particular in the specific case of the alcoholysis by C6 pathway, the Brønsted and Lewis acidities, similarly to the corresponding hydrothermal synthesis of levulinic acid in order to minimize the by-product formation [26]. Starting from glucose, Figure 3 reports the possible main pathways of C6 alcoholysis, taking into account both the Brønsted and Lewis acidity.

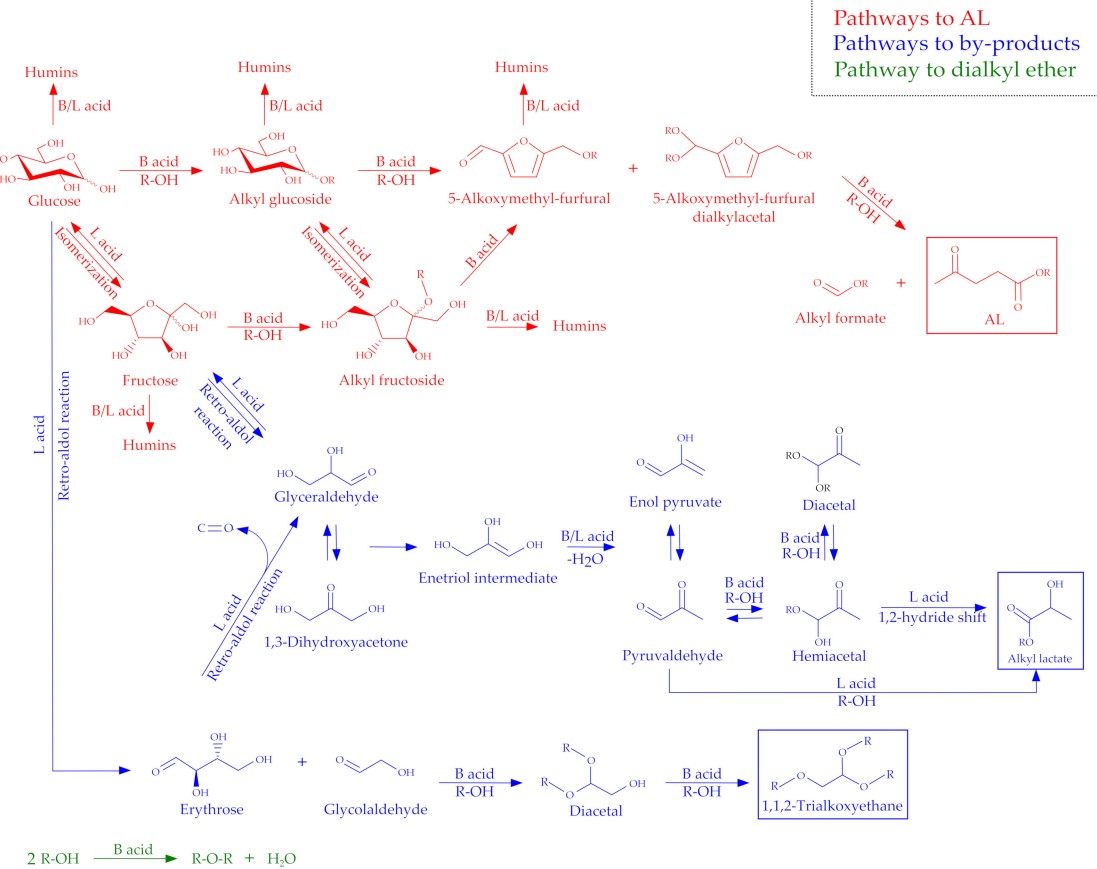

**Figure 3.** Possible pathways for the glucose alcoholysis, depending on the Brønsted (B) and Lewis (L) acidity.

The cascade conversion of C6 carbohydrates in the alcohol medium is much more complex than levulinic acid esterification, with the former being carried out in the presence of a Brønsted acid or, more advantageously, with bifunctional Brønsted–Lewis acid catalysts, in order to minimize the formation of reaction by-products. In this regard, according to Figure 3, C6 alcoholysis mainly involves three reaction pathways, the first one leading to the formation of the AL, the second one to that of the alkyl lactate, and the third one to that of 1,1,2-trialkoxyethane [18]. In detail, the formation of AL is mainly catalyzed by Brønsted acid sites and it is further improved by the moderate presence of Lewis acid sites, which enables the glucose isomerization to fructose, together with that of the corresponding alkyl glucoside to alkyl fructoside, in this way reducing the formation of undesired by-products and shifting the reaction towards the production of the target ester [26–29]. On this basis, the balanced presence of Lewis acid sites is of paramount importance for catalyzing the aldose–ketose isomerization [30]. The subsequent Brønsted-catalysed dehydration of alkyl glucosides/fructosides leads to the formation of the 5-alkoxymethylfurfural, together with its dialkylacetal derivative [15], which undergo ring-opening in the alcohol medium, in order to give the alkyl formate and the desired AL. Therefore, it is evident that Brønsted catalysts, such as traditional sulfuric or hydrochloric acid, are more effective for fructose conversion to AL, rather than for that of glucose, which is slower [31], and that fructose should be the ideal feedstock for obtaining the highest AL yield. Instead, the reactivity of di- and poly-saccharides depends on the reactivity of their constituent monosaccharide units, up to the cellulose, which is the most complex polysaccharide to convert [15]. A proper tuning of the Brønsted–Lewis acid properties of the catalyst must take into account the interaction of the catalyst with the reaction medium, as occurs for the metal salts, which can undergo hydrolysis in the presence of the formed/added water, leading to different Brønsted–Lewis acid properties [32], which also depend on both involved cations and anions [8,33]. The main by-products deriving from the C6 path are insoluble polymeric furans, named humins, deriving from acid-catalyzed condensation reactions of the reactive furanic intermediates/carbohydrate precursors [31,34]. However, their formation is particularly favored in the water medium, whilst it is limited in the alcohol [35–37]. This aspect represents a remarkable advantage in favor of the alcoholysis approach. Certainly, excessive Brønsted acidity is the main reason for the humin production [38]. Therefore, when considering the sulfuric acid as the catalyst of reference, to limit the humin formation, the combination of a very low acid concentration (up to 0.01 mol/L) and a higher reaction temperaturehas been the preferred choice for AL production [39]. Besides, the use of catalysts with strong Lewis acid sites can also activate the humin formation [40], thus highlighting that this side-reactionmust be controlled by properly tuning the total amount of acid sites of the catalyst, the ratio of Brønsted/Lewis acid sites and their strength, as well as reaction temperature and duration. Becausethe generation of alkyl lactate and 1,1,2-trialkoxyethane from hexoses is generally catalyzed by Lewis acids, the ratio of alkyl lactate and 1,1,2-trialkoxyethane to AL is strongly dependent on the Lewis-to-Brønsted acidity ratio of the adopted catalyst. A further side-reaction thatcertainly occurs is the alcohol etherification to the corresponding dialkyl ether (Figure 3) [41]. This is a well-known issue and it is one of the main obstacles for the development of the AL production on a greater scale, because it could cause a significant solvent loss, which cannot be recovered and recycled [42]. The addition of water as a reaction co-solvent may reduce the alcohol etherification and the coke/humin formation [32]. Moreover, this side reaction can be controlled by preferring mild acid catalysis, therefore working at a very low mineral acid concentration [43] or using solid catalysts [44], thus, once again, highlighting the key role of catalysis in optimizing this process. However, when possible, dialkyl ether can be considered to be a coproduct of alcoholysis and recovered by distillation to be used for other valuable applications, in particular as bio-additives for fuels [45], thus justifying the alcohol loss.

## 1.2. Possible AL Applications and Aim of the Review

Regarding the possible applications of ALs, in principle, they can be used as biofuels, biofuel additives, green solvents, flavoring agents, lubricants, fragrances, and polymer plasticizers [8,41]. The use of short-chain ALs, in particular, methyl levulinate (ML) and EL, as sustainable oxygenated

fuel-additives, has been deeply investigated, resulting in being really promising for this purpose [8,41]. Shrivastav et al. [46] blended C1-C4 ALs with conventional gasoline fuelmaintaining up to 18 mol%, density, viscosity, and compressibility within the recommended limits. The tested ALs showed good octane ratings, similar C/H ratio to that of aromatics, and better local oxygen concentration than that of traditional methyl *tert*-butyl ether. The authors increased the amount of the blended AL up to 35 mol%, reducing the aromatic content of gasoline. However, low-chain ALs suffer from some limitations, including high oxygen content, good water solubility, and low-energy density. For this reason, new research trends are rather directed towards the synthesis of longer-chain ALs with higher carbon content and stronger hydrophobicity, thus improving the energy density and water insolubility. These "biodiesel-like" ALs result in being more appropriate as oxygenated additives for diesel blends [47–51]. However, given the more difficult synthesis of the levulinates with increasing length of the alcohol residue, only a few papers are reported for these esters, mostly synthesized starting from the more costly and pure levulinic acid, often preferring elegant catalysts, which have been synthesizedad hocon the laboratory scale. In order to increase the development of high-volume automotive applications, it is necessary to find a compromise between the synthesis of these bio-products, which should be easily achievable, and their motor performances, which should be (at least) satisfactory. Up to now, butyl levulinate (BL) meets both of these requirements, due to its feasible synthesis, even starting from real biomass, and to the related good diesel performances, also allowing for a significant reduction of both CO and soot emissions [52]. In this context, Kremer et al. [53] have discussed, in detail, the engineering aspects that are related to the motor applications of this levulinate, also positively re-evaluating those of the di-*n*-butyl ether. This last compound, which is obtained as the main by-product from the same alcoholysis process, can be used, in addition to a pure diesel alternative, as an ignition enhancer for low-cetane biofuels, due to its high self-ignitability (Cetane Number = 100) [53].

In principle, alcoholysis can be applied as mild biomass pre-treatment, in a biorefinery perspective, such as for the selective depolymerization/liquefaction of its components, e.g., cellulose, hemicellulose, and lignin, which can be effectively fractionated and converted into valuable bioproducts, such as alkyl glucosides/xylosides and soluble aromatics [54–56], to be used for niche added-value applications. In particular, the use of long-chain alkyl glucosides as bio-surfactants has been widely demonstrated, showing the remarkable advantages of performance, biodegradability, low-toxicity, and environmental compatibility [57,58]. On the other hand, alcoholysis alsoenables the breakdown of ether linkages of lignin, to give smaller aromatics [21,59], which can be isolated and further functionalized to obtain more value-added products, such as polymer building blocks/pharmaceuticals, or defunctionalized to simpler *drop-in* molecules (BTX, phenol, catechol, and cyclohexane), which have a large market potential [60].

This review provides an overview of the most recent and promising advances on the one-pot production of ALs by alcoholysis of both model C6 carbohydrates and real cellulosic biomasses, in the presence of homogeneous or heterogeneous catalysts. Up to now, the choice of real biomass feedstocks has not been properly emphasized in the literature, whilst it is particularly attractive and strategic to promote the development/intensification of this process, and it should be preferred, as well as the one-pot cascade approach, occurring without the separation/purification of the reaction intermediates [61]. Other practical aspects, such as the strategic use of high feedstock loadings and diluted acids, will be specifically considered, because of the related environmental and economic concerns, ofgreater industrial relevance [52,62–64]. Furthermore, the issue of the catalyst heterogenization, which is already highly desired for the levulinic acid process [65], but industrially still unsolved, will be considered also for the production of ALs, in this last case being (theoretically) simpler for model and pure feedstocks, thanks to their better mass transfer with the catalyst and the reduced formation of humin by-products [44].

To make the comparison among the different available dataeasier, the AL yield from model C6 carbohydrates has been calculated on a molar basis, according to Equation (1), this basis of calculation being more useful for academic evaluations. Instead, the AL yield from real biomasses was calculated

respect to the weight of the dry biomasses, according to Equation (2), which is of greater practical utility in the industrial field.

$$\text{AL yield from C6 model compound } (Y_{AL}) \text{ mol\%} = [\text{AL recovered (mol)/C6 units (mol)}] \times 100 \quad (1)$$

$$\text{AL yield from real biomass } (Y_{AL}) \text{ wt\%} = [\text{AL recovered (g)/dry biomass (g)}] \times 100 \quad (2)$$

## 2. AL Synthesis from Model Compounds

In the following paragraphs, the state of the art regarding the synthesis of ALs from C6 model compounds of different complexity will be discussed, while taking into account the most promising available data that were reached with both homogeneous and heterogeneous catalysts. For a clearer discussion, ALs will be considered separately, methyl levulinate (ML), ethyl levulinate (EL), propyl levulinate (PL), butyl levulinate (BL), pentyl levulinate (PeL), and hexyl levulinate (HL).

### 2.1. ML Synthesis from Model Carbohydrates

Table 1 summarizes the available data for ML synthesis from C6 model carbohydrates with inorganic mineral acids or metal salts.

**Table 1.** Inorganic mineral acids or metal salts for the methyl levulinate (ML) production from C6 model carbohydrates.

| Entry | Feedstock | Catalyst | Cat.(g)/MeOH (g) [1] | T (°C) | t (h) | Heat [2] | $Y_{ML}$ (mol%) | Ref. |
|---|---|---|---|---|---|---|---|---|
| ML_1 | Fructose | $H_2SO_4$ (100 wt%) | 0.2/15.8 | 200 | 2 | Conv. | 85 | [66] |
| ML_2 | Fructose | $H_2SO_4$ (96 wt%) | 0.2/5.0 | 160 | 0.5 | MW | 90 | [67] |
| ML_3 | Fructose | $H_2SO_4$ (96 wt%) | 0.3/8.0 | 160 | 0.67 | MW | 93 | [68] |
| ML_4 | Glucose | $H_2SO_4$ (96 wt%) | 0.2/5.0 | 160 | 1 | MW | 70 | [67] |
| ML_5 | Glucose | $H_2SO_4$ (96 wt%) | 0.3/8.0 | 160 | 0.67 | MW | 72 | [68] |
| ML_6 | Cellulose | $H_2SO_4$ (100 wt%) | 0.2/6.9 | 179 | 0.25 | Conv. | 31 | [69] |
| ML_7 | Cellulose | $H_2SO_4$ (100 wt%) | 0.2/3.4 [3] | 194 | 0.083 | Conv. | 45 | [69] |
| ML_8 | Cellulose | $H_2SO_4$ (100 wt%) | 0.1/39.6 | 210 | 2 | Conv. | 50 | [70] |
| ML_9 | Cellulose | $H_2SO_4$ (100 wt%) | 0.1/39.5 | 190 | 4.2 | Conv. | 55 | [71] |
| ML_10 | Cellulose | $H_2SO_4$ (100 wt%) | 0.2/39.5 | 180 | 4 | Conv. | 42 | [71] |
| ML_11 | Cellulose | $H_2SO_4$ (96 wt%) | 0.3/8.0 | 180 | 0.67 | MW | 46 | [68] |
| ML_12 | Cellulose | $H_2SO_4$ (96 wt%) | 0.2/5.0 | 160 | 1 | MW | 70 | [67] |
| ML_13 | Fructose | $HReO_4$ | 0.1/22.2 | 160 | 16 | Conv. | 76 | [72] |
| ML_14 | Fructose | $Fe_2(SO_4)_3$ | 2.2/9.0 | 170 | 2 | Conv. | 25 | [73] |
| ML_15 | Glucose | $Fe_2(SO_4)_3$ | 2.2/9.0 | 170 | 2 | Conv. | 26 | [73] |
| ML_16 | Glucose | $La_2(SO_4)_3$ | 3.2/9.0 | 170 | 2 | Conv. | 30 | [73] |
| ML_17 | Glucose | $Fe_2(SO_4)_3$ | 0.8/15.8 | 200 | 2 | Conv. | 43 | [66] |
| ML_18 | Glucose | $Al_2(SO_4)_3$ | 0.7/15.8 | 200 | 2 | Conv. | 54 | [66] |
| ML_19 | Fructose | $Al_2(SO_4)_3$ | 0.6/29.8 | 160 | 1 | MW | 70 | [67] |
| ML_20 | Glucose | $Al_2(SO_4)_3$ | 0.6/29.8 | 160 | 1 | MW | 70 | [67] |
| ML_21 | Cellulose | $Al_2(SO_4)_3$ | 0.6/29.8 | 160 | 1 | MW | 20 | [67] |
| ML_22 | Cellulose | $Al_2(SO_4)_3$ | 0.6/29.8 | 160 | 4 | MW | 49 | [67] |
| ML_23 | Cellulose | $Al_2(SO_4)_3$ | 0.6/29.8 | 180 | 1 | MW | 61 | [67] |
| ML_24 | Glucose | $Al_2(SO_4)_3$ | 0.4/35.0 | 160 | 2.5 | Conv. | 64 | [29] |
| ML_25 | Sucrose | $Al_2(SO_4)_3$ | 0.4/35.0 | 160 | 2.5 | Conv. | 55 | [29] |
| ML_26 | Cellulose | $Al_2(SO_4)_3$ | 0.4/35.0 | 180 | 5 | Conv. | 44 | [29] |
| ML_27 | Sucrose | $Al_2(SO_4)_3$ | 0.4/22.4 [4] | 180 | 0.25 | MW | 83 | [32] |
| ML_28 | Cellulose | $Al_2(SO_4)_3$ | 0.4/22.4 [4] | 170 | 1 | MW | 44 | [32] |
| ML_29 | Cellulose | $Al_2(SO_4)_3$ | 0.4/22.4 [4] | 180 | 0.67 | MW | 71 | [32] |
| ML_30 | BM-cellulose [5] | $Al_2(SO_4)_3$ | 0.6/29.8 | 160 | 1 | MW | 58 | [74] |
| ML_31 | BM-cellulose [5] | $Al_2(SO_4)_3$ | 0.6/29.8 | 170 | 0.75 | MW | 65 | [74] |
| ML_32 | Cellulose [6] | $Al_2(SO_4)_3 \cdot 18H_2O$ | 0.6/35.0 | 180 | 3 | Conv. | 52 | [75] |

[1] The amounts of catalyst and MeOH have been normalized to 1 g of feedstock. [2] "Conv." and "MW" stand for "Conventional" and "Microwave", respectively. [3] 6.9 g of $CCl_4$ were added to this reaction mixture. [4] 1.2 g of water were added to this reaction mixture. [5] "BM" stands for "Ball-Milled". [6] Cellulose (2.0 g) was pre-oxidized at 200 °C for 10 h under $O_2$ flow (50 mL min.$^{-1}$) before the alcoholysis.

Sulfuric acid has been widely employed as a homogeneous Brønsted acid catalyst for the methanolysis of C6 model feedstocks, thanks to its excellent compromise between high activity and low-cost. Higher ML molar yields were achieved, starting from fructose (runs ML_1—ML_3, Table 1), rather than from glucose (runs ML_4—ML_5, Table 1), due to the simpler conversion of the first feedstock, which, contrarily to the glucose, does not need of the Lewis acid catalysis. The use of sulfuric acid for the one-step alcoholysis of model cellulose was proposed for the first time by Garves [69], who reported good ML yields, working under subcritical conditions (runs ML_6—ML_7, Table 1). However, sulfuric acid is also an excellent catalyst for alcohol etherification, which may cause a progressive and significant consumption of the solvent/reactant. To solve this issue, etherification must be controlled and limited as much as possible, in particular by minimizing the acid concentration, a process solution that also improves the environmental sustainability and reduces the plant costs of the alcoholysis process. Peng et al. [43] investigated this aspect more in-depth for the glucose alcoholysis at 200 °C, in the presence of two different $H_2SO_4$ concentrations (0.1 and 0.005 mol/L). The authors found that methanol conversion to dimethyl ether, after 2.5 h, and in the presence of 0.1 mol/L of $H_2SO_4$, was remarkable (about 59 mol%), whilst it strongly decreased (to about 2 mol%), if the use of diluted $H_2SO_4$ was preferred. More recently, the use of very dilute sulfuric acid ($\leq$0.01 mol/L) has been rightly preferred by Li et al. [70], reporting similar ML yields (about 50 mol%), only extending the reaction time (run ML_8, Table 1). Besides, the authors also discussed the possible ML recovery, combining atmospheric and vacuum distillation, while using *n*-dodecane for improving the separation of the heavy fraction (ML) from the light one (MeOH). Wu et al. [71] achieved similar ML yields, always preferring a very low sulfuric acid concentration ($\leq$0.04 mol/L), and by properly optimizing reaction temperature and time (runs ML_9—ML_10, Table 1). Besides, the authors compared the catalytic performances of sulfuric acid with those of other common Brønsted acids, such as phosphoric, formic, and acetic acids, working at the same molar concentration (0.04 mol/L). The higher catalytic activity of the sulfuric acid over phosphoric acid was attributed the stronger acidity of the former [76], whilst organic acids, such as formic and acetic ones, were not efficient incatalyzing this reaction, due to their unwanted esterification, occurring under the typical reaction conditions that were required for ML production [77]. The use of MW as an efficient heating system is better exploitable if applied to the conversion of more recalcitrant substrates, such as cellulose, achieving better ML yields with respect to the conventional heating systems and reduced formation of the reaction by-products, mainly humins. In this context, the best ML yields achieved by Feng et al. [68] and Chen et al. [67] (runs ML_11—ML_12, Table 1) fully support these premises, highlighting that such efficient heating allows for the use of milder reaction conditions (temperature and time) than conventional heating systems. Up to now, the highest ML yield from model cellulose was reported by Chen et al. [67] (run ML_12, Table 1), amounting to 70 mol%, which is the best value to consider as reference for the development of new efficient catalysts. The innovation of the catalytic system is a very hot topic and many alternatives have been already evaluated. Perrhenic acid ($HReO_4$) is just an example, but, certainly, this is not the simplest and cheapest catalyst to propose as an alternative to the sulfuric acid. Its good efficiency towards fructose methanolysis has been recently demonstrated by Bernardo et al. (run ML_13, Table 1) [72], but its real use on a greater scale is not sustainable and competitive with that of the sulfuric acid. Despite the high activity of sulfuric acid, separation and corrosion issues hinder its use, especially at high concentrations, thus the research is moving towards the use of other kinds of catalysts, in particular inorganic salts. In this context, cheap metal sulfates ($Na_2SO_4$, $K_2SO_4$, $MnSO_4$, $CoSO_4$, $NiSO_4$, $ZnSO_4$, $CuSO_4$, $Fe_2(SO_4)_3$, $La_2(SO_4)_3$, and $Ce(SO_4)_2$) have been recently tested by Sun et al. [73] for the solvothermal conversion of glucose and fructose, evaluating different reaction media, including MeOH. As a general consideration, the metal salts provides both Lewis acid sites (due to the unsaturated metal center) and Brønsted acid ones (due to the hydrolysis/methanolysis of the salt), which catalyze the isomerization of glucose to fructose and the dehydration of fructose to ML, respectively, whilst sulfate anions generally have a positive role towards this reaction, strongly chelating with the carbohydrates, thus preventing the formation of unwanted by-products, such as humins [73]. Among the tested metal sulfates, $Fe_2(SO_4)_3$ and $La_2(SO_4)_3$ gave the

highest ML molar yields at 170 °C, in the range 25–30 mol% (runs ML_14—ML_16, Table 1), preferring the cheaper $Fe_2(SO_4)_3$, which suppressed the humin formation. However, despite this catalyst seeming interesting for ML synthesis, the reported ML yields from glucose and fructose are too low, especially if compared with the traditional sulfuric acid. A possible solution to improve the catalysis in the presence of metal salts provides the additional use of modest quantities of sulfuric acid, which favors the Brønsted acid-catalyzed steps. Peng et al. [66] investigated the catalytic effect of various metal sulfates, who confirmed a significant effect of these salts on the reaction selectivity. In particular, $K_2SO_4$ and $Na_2SO_4$ were particularly effective for stabilizing the methyl glucoside intermediate, suppressing its further transformationand minimizing the humin formation. Instead, $Fe_2(SO_4)_3$ and $Al_2(SO_4)_3$ gave the highest ML yields (runs ML_17—ML_18, Table 1), but also favored the formation of humins and dimethyl ether by-products. However, the formation of these by-products (in particular dimethyl ether) was significant also in the sole presence of these two metal sulfates, being the conversion of methanol to dimethyl ether equal to 19 and 23 mol%, adopting $Fe_2(SO_4)_3$ and $Al_2(SO_4)_3$, respectively, highlighting the importance of further optimizing the alcoholysis reaction with these more active catalytic systems [66]. The optimization of the alcoholysis of different C6 model substrates in the presence of $Al_2(SO_4)_3$ was investigated more in-depth (runs ML_19—ML_32, Table 1) [29,32,67,74,75]. The addition of water reduces humin/coke formation and solvent consumption (runs ML_28—ML_29, Table 1) [32]. The Lewis acid species $[Al(OH)_x(H_2O)_y]^{n+}$ and Brønsted acid species $H^+$, which were generated by in-situ hydrolysis of $Al_2(SO_4)_3$, were responsible for the improved catalytic performances. The amount of water in MeOH could affect the equilibrium of metal salt hydrolysis, as well as the acid density of the reaction mixture, which would affect the salt reactivity towards cellulose conversion and related product distribution. In particular, the addition of water significantly inhibited the etherification reaction of MeOH to dimethyl ether, which was unavoidable in the presence of Brønsted acidic catalysts at elevated temperatures. When no water was added, the dimethyl ether yield was approximately20 mol%, whilst it decreased below 5 mol% by adding a low amount of water (0.6 mL). Moreover, water addition improved the cellulose solvolysis, moving the reaction towards the hydrolysis of the glycosidic C-O bonds, rather than to its thermal decomposition to coke, whose yield was reduced from about 40 wt%, in the absence of water, to about 10 wt%, when the water content was over 0.6 mL. The combined use of $(Al_2(SO_4)_3 + H_2O)$, together with MW heating as the efficient heating system, and milder reaction conditions, allowed for the improvement of the ML yield from model cellulose up to a maximum of about 70 mol% (run ML_29, Table 1), a value that is similar to that reached with the traditional sulfuric acid. Similar results were obtained under conventional heating but employing longer reaction times (run ML_24, Table 1) [29], while harsher reaction conditions resulted in being detrimental for ML production (run ML_26, Table 1). Some improvements have been recently proposed for enhancing the ML yield, always in the presence of $Al_2(SO_4)_3$ as the main catalytic system. Chen et al. [74] performed a ball-milling pretreatment on the cellulose feedstock, for reducing its particle size and crystallinity index, thus improving the catalyst accessibility during the alcoholysis step. Anyway, this pre-treatment did not lead to significant improvements in ML yield, which reached the maximum value of about 65 mol%, working at 170 °C and for 45 min., under MW heating (runs ML_30—ML_32, Table 1). Very recently, an oxidation pretreatment has been considered for improving ML production from cellulose, aimed at the conversion of its hydroxymethyl groups into carboxylic ones, thus providing the Brønsted acid sites for improving the catalysis that is related to the steps of interest [75]. However, by adopting this interesting approach, the maximum ML yield was only 52 mol%, working at 180 °C for 3 h (run ML_32, Table 1), highlighting that, up to now, this pre-treatment is not very functional for this purpose.

Sulfonic acids or/and sulfonate salts or resins have been investigated for the ML production from C6 model carbohydrates, and the best available data are summarized in Table 2.

**Table 2.** Sulfonic acids, sulfonate salts, and their combinations for the ML production from C6 model carbohydrates, under conventional heating.

| Entry | Feedstock | Catalyst | Cat.(g)/MeOH(g) [1] | T (°C) | t (h) | $Y_{ML}$ (mol%) | Ref. |
|---|---|---|---|---|---|---|---|
| ML_33 | Cellulose | PTSA [2] | 0.05/39.5 | 180 | 5 | 20 | [27] |
| ML_34 | Cellulose | PTSA [2] | 0.12/6.9 | 210 | 0.5 | 34 | [69] |
| ML_35 | Cellulose | PTSA [2] | 0.35/39.5 | 180 | 4 | 46 | [71] |
| ML_36 | Cellulose | In(OTf)$_3$ | 0.03/39.5 | 180 | 5 | 52 | [27] |
| ML_37 | Cellulose | PTSA [2] + In(OTf)$_3$ | 0.05 + 0.03/39.5 | 180 | 5 | 70 | [27] |
| ML_38 | Cellulose | 2-NSA [3] + In(OTf)$_3$ | 0.05 + 0.03/39.5 | 180 | 5 | 75 | [27] |
| ML_39 | Glucose | BSA [4] + In(OTf)$_3$ | n.a.[5] | 180 | 5 | 70 | [78] |
| ML_40 | Mannose | BSA [4] + In(OTf)$_3$ | n.a.[5] | 180 | 5 | 76 | [78] |
| ML_41 | Galactose | BSA [4] + In(OTf)$_3$ | n.a.[5] | 180 | 5 | 70 | [78] |
| ML_42 | Cellulose | PTSA [2] + Al(OEt)$_3$ | 0.08 + 0.01/39.5 | 180 | 5 | 69 | [79] |
| ML_43 | Cellulose | PTSA [2] + Al(acac)$_3$ | 0.08 + 0.02/39.5 | 180 | 5 | 72 | [79] |
| ML_44 | Cellulose | 2-NSA [3] + Al(OH)$_3$ | 0.10 + 0.01/39.5 | 180 | 5 | 74 | [79] |
| ML_45 | Fructose | Amberlyst-15 | 0.40/64.0 | 100 | 24 | 54 | [80] |
| ML_46 | Fructose | Amberlyst-15 | 0.28/48.3 | 170 | 15 | 68 | [81] |
| ML_47 | Glucose | Amberlyst-15 | 0.27/32.4 | 160 | 5 | 12 | [82] |
| ML_48 | Glucose | Amberlyst-15 | 0.54/32.4 | 160 | 5 | 75 | [83] |
| ML_49 | Fructose | Nafion NR50 | 0.28/48.3 | 170 | 15 | 73 | [81] |
| ML_50 | Fructose | PD-En-SO$_3$H [6] | 0.28/48.3 | 170 | 15 | 78 | [81] |
| ML_51 | Fructose | PSSA-g-CNT [7] | 0.40/64.0 | 100 | 24 | 69 | [80] |
| ML_52 | Fructose | PSSA-g-CNF [8] | 0.40/64.0 | 100 | 24 | 53 | [80] |
| ML_53 | Fructose | BSA-g-CMK-5 [9] | 0.40/64.0 | 100 | 24 | 49 | [80] |
| ML_54 | Fructose | BSA-g-CNT [10] | 0.40/64.0 | 100 | 24 | 12 | [80] |
| ML_55 | Fructose | 5-Cl-SHPAO [11] | 0.25/20.0 | 160 | 1 | 79 | [84] |
| ML_56 | Glucose | 5-Cl-SHPAO [11] | 0.25/20.0 | 160 | 12 | 60 | [84] |
| ML_57 | Inulin | 5-Cl-SHPAO [11] | 0.25/20.0 | 160 | 8 | 71 | [84] |
| ML_58 | Cellulose | Sulfonated char | 0.50/31.6 | 200 | 1.25 | 30 | [85] |
| ML_59 | Cellulose | Sulfonated char | 0.50/31.6 | 225 | 0.75 | 30 | [85] |

[1] The amounts of catalysts and MeOH have been normalized to 1 g of feedstock. [2] "PTSA" stands for "*p*-toluenesulfonic acid". [3] "2-NSA" stands for "2-naphthalenesulfonic acid". [4] "BSA" stands for "benzenesulfonic acid". [5] "n.a." stands for "not-available". [6] "PD-En-SO$_3$H" stands for "sulfonic acid-grafted ethylenediamine-functionalized mesoporous polydivinylbenzene". [7] "PSSA-g-CNT" stands for "poly(*p*-styrenesulfonic acid)-grafted carbon nanotubes". [8] "PSSA-g-CNF" stands for "poly(*p*-styrenesulfonic acid)-grafted carbon nanofibers". [9] "BSA-g-CMK-5" stands for "benzenesulfonic acid-grafted carbon mesostructured by KAIST-5". [10] "BSA-g-CNT" stands for "benzenesulfonic acid-grafted carbon nanotubes". [11] "5-Cl-SHPAO" stands for "sulfonated hyperbranched poly(aryleneoxindole) with chloride substituent in the fifth position of isatin".

*p*-Toluenesulfonic acid (PTSA) has been proposed for a long time as a reference Brønsted acid catalyst for studying the cellulose alcoholysis (runs ML_33—ML_35, Table 2), as an effective alternative to the already discussed sulfuric acid, achieving similar maximum ML yield (about 50 mol%). Although single catalysts have been widely used for this reaction, achieving good results, recent catalytic studies are rather oriented towards the appropriate tuning of the Brønsted–Lewis acidity, generally achieved in the presence of binary catalysts. Therefore, as previously observed, Lewis acids have been used to improve the ML yield, both alone or in combination with small amounts of sulfonic acids, as Brønsted acids. In this context, In(OTf)$_3$ has been proposed by Tominaga et al. [27] as an efficient Lewis acid for the cellulose alcoholysis, both alone and in combination with PTSA or 2-naphthalenesulfonic acid (2-NSA), as active Brønsted acids (runs ML_36—ML_38, Table 2). The sole In(OTf)$_3$ showed catalytic performances that were similar to those of PTSA, whilst the ML yield increased up to 70–75mol%, when In(OTf)$_3$ and PTSA or 2-NSA were used in combination. The adopted catalytic system resulted in being stable and recyclable, being recovered as a residue after the distillation of the solvent/products. Similar promising results were obtained by Nemoto et al. [78] with simpler feedstocks, such as glucose, mannose, or galactose, highlighting a similar reactivity of these carbohydrates (runs ML_39—ML_41, Table 2). In a more recent work of Tominaga et al. [79], binary catalytic systems that were composed of aluminum compounds (Al(OEt)$_3$, Al(acac)$_3$, or Al(OH)$_3$) and organic sulfonic acids (PTSA or 2-NSA), were found to be particularly efficient for direct ML synthesis from microcrystalline cellulose (runs ML_42—ML_44, Table 2), achieving maximum ML yields that were similar to those of the best triflate-based catalytic systems (runs ML_37—ML_38, Table 2). Amberlyst-15 and Nafion NR50 have been mainlyconsidered as catalysts of reference, for justifying the development of new heterogeneous catalysts, despite the available data beinglimited to the conversion soluble C6 carbohydrates (runs ML_45—ML_49, Table 2). A novel and efficient sulfonic acid-grafted ethylenediamine-functionalized mesoporous polydivinylbenzene (PD-En-SO$_3$H) heterogeneous catalyst was synthesized by Pan et al. [81]. PD-En-SO$_3$H showed excellent catalytic performances for the conversion of fructose to ML, being more active than commercial Amberlyst-15 and Nafion NR50 (compare run ML_50 with runs ML_46 and ML_49, Table 2). Moreover, this new catalyst could be reused for four times, maintaining its high catalytic activityunaltered. A series of sulfonic acid-functionalized carbon materials, including poly(*p*-styrenesulfonic acid)-grafted carbon nanotubes (PSSA-g-CNT), poly(*p*-styrenesulfonic acid)-grafted carbon nanofibers (PSSA-g-CNF), benzenesulfonic acid-grafted CMK-5 (BSA-g-CMK-5), and benzenesulfonic acid-grafted carbon nanotubes (BSA-g-CNT), have been applied for fructose conversion to ML (runs ML_51—ML_54, Table 2) [80]. The catalytic activities of these sulfonic acid-functionalized carbon materials, applied to the conversion of fructose to ML, follow the order of their acid strength and PSSA-g-CNT exhibited the highest acid density and the best catalytic performances to ML (maximum yield of 69 mol%). The authors claimed the high thermal stability and ease of recovery of these catalysts. Sulfonated hyperbranched poly(aryleneoxindole)s (SHPAOs) were also used for the conversion of simple monosaccharides and inulin to ML, with high yields up to 79 mol% (runs ML_55—ML_57, Table 2) [84]. Becauseofthe soluble character of the hyperbranched catalyst in the alcoholic solvent, it was easily separated from the solid humins, and recovered from the solution over a commercial low molecular weight cut-off filter. Moreover, the recovered catalyst showed in a recycle run a comparable catalytic activity (per catalyst weight) and product selectivity. To exploit more sustainable carbon bio-materials, an amorphous carbon-based catalyst was prepared by the sulfonation of the bio-char obtained from fast pyrolysis (N$_2$atm; 550 °C) of *jatrophacurcas* de-oiled seed cake and tested for the cellulose methanolysis, reaching a maximum ML yield of 30 mol% (runs ML_58—ML_59, Table 2) [85]. The authors demonstrated that this functionalized carbon catalyst was stable for five cycles with a slight loss in catalytic activity.

Heteropolycompounds, or heteropolyoxometalates (POMs), consist of metal oxide clusters of early transition metals [86]. These could be heteropolyacids (HPAs) or their salts, containing one heteroatom (X = P(V), As(V), Si(IV), and B(III)) and addenda atoms (M = W(VI), Mo(VI), and V(V)). The main heteropolyanion that is used in the field of biomass conversion is the Keggin structure (XM$_{12}$O$_{40}$$^{n-}$), but

catalysts with Wells–Dawson structure ($X_2M_{18}O_{62}^{m-}$) have also found applications [87]. These catalysts are widely used in acid-catalyzed and oxidation reactions, with economic and green benefits, but generally exhibit poor efficiency in their bulk form, due to the limited number of exposed active sites [88]. HPAs are usually soluble in aqueous and organic solvents, making their separation/regeneration from the reaction mixturedifficult, and, for this reason, many attempts for their heterogenization have been made by immobilization on support or by the formation of an insoluble ionic material [86]. The use of these catalysts has also been proposed for the ML production, and the most interesting available data are summarized in Table 3.

**Table 3.** Polyoxometalate (POM)-based catalysts for the ML production from C6 model carbohydrates.

| Entry | Feedstock | Catalyst | Cat.(g)/MeOH(g) [1] | T (°C) | t (h) | Heat [2] | $Y_{ML}$ (mol%) | Ref. |
|---|---|---|---|---|---|---|---|---|
| ML_60 | Fructose | $H_3PW_{12}O_{40}$ | 0.6/32.0 | 130 | 2 | Conv. | 60 | [89] |
| ML_61 | Cellulose | $H_3PW_{12}O_{40}$ | 2.0/63.0 | 160 | 5 | Conv. | 42 | [90] |
| ML_62 | Cellulose | $H_6P_2W_{18}O_{62}$ | 3.1/63.0 | 160 | 5 | Conv. | 52 | [90] |
| ML_63 | Cellulose | $H_4SiW_{12}O_{40}$ | 0.1/31.6 | 195 | 1 | Conv. | 12 | [91] |
| ML_64 | Sucrose | $H_5PW_{11}TiO_{40}$ | 1.5/42.0 | 100 | 5 | Conv. | 58 | [92] |
| ML_65 | Cellobiose | $H_5PW_{11}TiO_{40}$ | 1.5/42.0 | 120 | 5 | Conv. | 51 | [92] |
| ML_66 | Starch | $H_5PW_{11}TiO_{40}$ | 1.5/42.0 | 130 | 5 | Conv. | 47 | [92] |
| ML_67 | Cellulose | $H_5PW_{11}TiO_{40}$ | 1.5/42.0 | 160 | 7 | Conv. | 51 | [92] |
| ML_68 | Cellulose | $H_5PW_{11}TiO_{40}$ | 1.5/42.0 | 160 | 2 | MW | 63 | [92] |
| ML_69 | Fructose | $FePW_{12}O_{40}$ | 0.6/32.0 | 130 | 2 | Conv. | 74 | [89] |
| ML_70 | Glucose | $FePW_{12}O_{40}$ | 0.6/32.0 | 130 | 2 | Conv. | 14 | [89] |
| ML_71 | Sucrose | $FePW_{12}O_{40}$ | 0.6/32.0 | 130 | 2 | Conv. | 44 | [89] |
| ML_72 | Inulin | $FePW_{12}O_{40}$ | 0.6/32.0 | 130 | 2 | Conv. | 92 | [89] |
| ML_73 | Cellulose | $FePW_{12}O_{40}$ | 0.6/32.0 | 220 | 2 | Conv. | 14 | [89] |
| ML_74 | Fructose | $Sn_2SiW_{12}O_{40}$ | 1.7/65.8 | 150 | 3 | Conv. | 57 | [93] |
| ML_75 | Fructose | $AlPW_{12}O_{40}$ | 0.6/7.5 | 160 | 0.5 | MW | 70 | [94] |
| ML_76 | Glucose | $AlPW_{12}O_{40}$ | 0.6/7.5 | 160 | 0.5 | MW | 64 | [94] |
| ML_77 | Sucrose | $AlPW_{12}O_{40}$ | 0.6/7.5 | 160 | 0.5 | MW | 65 | [94] |
| ML_78 | Cellulose | $AlPW_{12}O_{40}$ | 0.6/7.5 | 160 | 0.5 | MW | 45 | [94] |
| ML_79 | Cellulose | $Cs_{2.5}H_{0.5}PW_{12}O_{40}$ | 0.4/71.4 | 300 | 0.02 | Conv. | 20 | [95] |
| ML_80 | Fructose | $[TMEDAPS]_3 [PW_{12}O_{40}]_2$ [3] | 1.0/64.0 | 120 | 12 | Conv. | 80 | [96] |
| ML_81 | Sucrose | $[PyPS]_3PW_{12}O_{40}$ [4] | 5.0/46.5 | 150 | 4.5 | Conv. | 76 | [97] |
| ML_82 | Glucose | $[PyPS]_3PW_{12}O_{40}$ [4] | 9.4/87.8 | 150 | 4 | Conv. | 59 | [97] |
| ML_83 | Starch | $[PyPS]_3PW_{12}O_{40}$ [4] | 10.6/98.8 | 150 | 5 | Conv. | 51 | [97] |
| ML_84 | Cellulose | $[PyPS]_3PW_{12}O_{40}$ [4] | 10.6/98.8 | 150 | 5 | Conv. | 71 | [97] |
| ML_85 | Cellulose | $[C16TA]H_5P_2W_{18}O_{62}$ [5] | 3.3/63.0 | 160 | 7 | Conv. | 58 | [90] |
| ML_86 | Fructose | 3-FPyPW [6] | 0.5/22.2 | 120 | 10 | Conv. | 82 | [98] |
| ML_87 | Fructose | 3-PhPyPW [7] | 0.5/22.2 | 140 | 8 | Conv. | 71 | [99] |
| ML_88 | Fructose | $Cs_{10.6}[H_{2.4}GeNb_{13}O_{41}]$ | 1.3/48.0 | 130 | 3 | Conv. | 53 | [100] |
| ML_89 | Glucose | $Cs_{10.6}[H_{2.4}GeNb_{13}O_{41}]$ | 1.3/48.0 | 130 | 3 | Conv. | 85 | [100] |
| ML_90 | Starch | $Cs_{10.6}[H_{2.4}GeNb_{13}O_{41}]$ | 1.3/48.0 | 150 | 4 | Conv. | 60 | [100] |
| ML_91 | Cellulose | $Cs_{10.6}[H_{2.4}GeNb_{13}O_{41}]$ | 1.3/48.0 | 165 | 10 | Conv. | 46 | [100] |
| ML_92 | Fructose | $Cs_{10.6}[H_{2.4}GeNb_{13}O_{41}]$ | 1.3/48.0 | 110 | 1 | MW | 55 | [100] |
| ML_93 | Glucose | $Cs_{10.6}[H_{2.4}GeNb_{13}O_{41}]$ | 1.3/48.0 | 110 | 1.5 | MW | 89 | [100] |
| ML_94 | Starch | $Cs_{10.6}[H_{2.4}GeNb_{13}O_{41}]$ | 1.3/48.0 | 120 | 3 | MW | 60 | [100] |
| ML_95 | Cellulose | $Cs_{10.6}[H_{2.4}GeNb_{13}O_{41}]$ | 1.3/48.0 | 165 | 3 | MW | 50 | [100] |

[1] The amounts of catalyst and MeOH have been normalized to 1 g of feedstock. [2] "Conv." and "MW" stand for "Conventional" and "Microwave", respectively. [3] "TMEDAPS" stands for "*N,N,N',N'*-tetramethyl-*N,N'*-dipropanesulfonic acid-1,6-hexanediammonium". [4] "PyPS" stands for "1-(3-sulfopropyl)pyridinium". [5] "C16TA" stands for $C_{16}H_{33}N(CH_3)_3$. [6] "3-FPyPW" stands for "3-fluoropyridine phosphotungstate". [7] "3-PhPyPW" stands for "3-phenylpyridine phosphotungstate".

In recent years, HPAs have attracted great interest as homogeneous acid catalysts, due to their strong Brønsted acidity, good structural mobility, and marked multi-functionality. Among these, the Keggin-type tungstosilicious and tungstophosphoric HPAs ($H_4SiW_{12}O_{40}$ and $H_3PW_{12}O_{40}$) have been proposed as efficient homogeneous catalysts for the alcoholysis of C6 model carbohydrates. In particular, $H_3PW_{12}O_{40}$ and $H_6P_2W_{18}O_{62}$ showed good catalytic performances for ML synthesis, starting from different model feedstocks, including cellulose (runs ML_60—ML_62, Table 3), whilst $H_4SiW_{12}O_{40}$ was selective for the formation of methyl glucoside intermediate and dimethyl ether by-product, rather than for ML synthesis, leading to the maximum methyl glucoside yield of 57 mol%, and the corresponding conversion of methanol to dimethyl ether of 28 mol% (run ML_63, Table 3) [91]. HPAs were modified by partially substituting the proton with larger cations, such as $Cu^{II}$, $Zn^{II}$, $Cr^{III}$, $Fe^{III}$, $Sn^{IV}$, $Ti^{IV}$, and $Zr^{IV}$, in order to improve the Lewis acidity of the HPAs, which helps the isomerization step from glucose to fructose, and to limit their solubility in alcohols, which complicates their work-up procedures. The best ML yields (in the range 50–60 mol%) were achieved in the presence of the $H_5PW_{11}TiO_{40}$ catalyst, adopting feedstocks of different complexity, including cellulose (runs ML_64—ML_68, Table 3) [92]. In these cases, the addition of efficient Lewis acid sites, together with the appropriate balance between Brønsted and Lewis acidity, was responsible for the promising performances of the catalyst. Contrarily, too strong Lewis acid sites promoted C-C bond cleavage, so that, for example, Fe-based HPWs gave lower ML yields than Ti-based HPWs. However, in another work of Liu et al. [89], the Brønsted–Lewis acidity of Fe-based HPWs catalysts was properly tuned in favor of the methanolysis pathway, thus allowing for a complete conversion of simple carbohydrates, reaching good ML yields, working at 130 °C for 2 h, and ensuring satisfactory recycling tests (runs ML_69—ML_72, Table 3) [89]. Anyway, the use of this catalyst for cellulose conversion gave an unsatisfactory ML yield (run ML_73, Table 3), which limits its application to simpler model feedstocks. Good performances were also reported for Sn(II) exchanged HPA Keggin $Sn_2SiW_{12}O_{40}$, leading to the maximum ML yield of about 57 mol%, starting from fructose, working at 150 °C for 3 h (run ML_74, Table 3) [93]. Instead, Zhang et al. [94] investigated the performances of the $AlPW_{12}O_{40}$ catalyst, demonstrating its effectiveness for the MW-assisted ML synthesis (ML yields in the range 45–70 mol%), starting from various carbohydrates, working at 160 °C for 30 min. (runs ML_75—ML_78, Table 3). $AlPW_{12}O_{40}$ acts as a bulk-type catalyst in a pseudo-liquid system, improving the accessibility of the active catalyst sites. The absorption of high-polar MeOH favors this liquefaction, increasing the distance between the heteropolyanion-based species, while increasing the proton mobility [94]. Among the other possible proposals of metal-exchanged HPAs, the catalytic performances of $Cs_xH_{3-x}PW_{12}O_{40}$ catalysts were tested for one-pot dissolution of microcrystalline cellulose and subsequent conversion to ML, in the presence of supercritical MeOH [95]. The authors identified $Cs_{2.5}H_{0.5}PW_{12}O_{40}$ as the best performing catalyst for ML production. In the absence of a catalyst, under supercritical conditions and for short reaction times (300 °C/10 MPa/1 min.), the cellulose dissolution was successful, but the next conversion to ML was inefficient. On the other hand, the addition of $Cs_xH_{3-x}PW_{12}O_{40}$ activated the methanolysis pathway to ML (maximum yield of 20 mol%), due to its well-balanced Lewis–Brønsted acidic properties (run ML_79, Table 3). The authors highlighted the improved accessibility of the solubilized carbohydrates to the Brønsted sites and the acid strength of $Cs_{2.5}H_{0.5}PW_{12}O_{40}$ catalyst. However, despite cellulose solubilization was significantly improved, the performances of all these catalysts were modest, especially if compared with those that were achieved under milder subcritical conditions. Heteropolyanion-based ionic liquid catalysts were tested for the conversion of different model compounds [96,97]. In particular, $[TMEDAPS]_3[PW_{12}O_{40}]_2$ gave high ML yield from fructose (run ML_80, Table 3), whilst $[PyPS]_3PW_{12}O_{40}$ was used also for more complex substrates (runs ML_81—ML_84, Table 3), including cellulose, in this case reaching the maximum ML yield of about 71 mol%, working for 5 h at 150 °C. In particular, the remarkable efficiency of $[PyPS]_3PW_{12}O_{40}$ towards the cellulose alcoholysis was ascribed to the high acidic strength of the sulfonic-functionalized cation and to its synergistic effects with the corresponding heteropolyanion. Besides, the proper acidic strength of this catalyst favored the whole process consisting of cellulose hydrolysis, levulinic acid formation, and

its esterification. Moreover, the authors demonstrated that $[PyPS]_3PW_{12}O_{40}$ can be effectively recovered by self-separation, through simple temperature control, also showing excellent reusability, even after ten runs. $H_6P_2W_{18}O_{62}$ was properly modified with surfactants, in order to prepare a micellar assembly $(C_{16}TA)_xH_{6-x}P_2W_{18}O_{62}$ [90], which gave the maximum ML yield of about 58 mol%, starting from cellulose (run ML_85, Table 3). In particular, the promising performances of the $(C_{16}TA)H_5P_2W_{18}O_{62}$ catalyst were attributed to its micellar structure, high acidic content, and good oxidizing ability. The assembly of HPAs with basic organic species is another possibility forsolidifying the homogeneous acid and modulate its acidity. Novel substituted pyridine phosphotungstates were prepared and tested for the direct conversion of fructose to ML [98,99]. 3-fluoropyridine phosphotungstate (3-FPyPW) displayed superior catalytic activity for the synthesis of ML from fructose (run ML_86, Table 3), which was ascribed to its relatively higher acidity, and the maximum ML yield of 82 mol% was achieved [98]. In another work of Fang et al. [99], 3-phenylpyridine phosphotungstate (3-PhPyPW) hybrid catalyst displayed good catalytic performances in the upgrade of fructose to ML (yield of 71 mol%), after 8h at 140 °C, which was attributed to its relatively large pore size and high hydrophobicity (run ML_87, Table 3). As a further improvement of HPA-based catalysis, very recently, a trifunctional polyoxometalate $Cs_{10.6}[H_{2.4}GeNb_{13}O_{41}]$ catalyst, which included Brønsted acid sites, Lewis acid sites, and basic sites, was synthesized and used for this reaction, starting from different model compounds, under conventional and MW heating (runs ML_88—ML_95, Table 3) [100]. The further addition of basic sites improves the isomerization from methyl glucoside to methyl fructoside and the subsequent formation of 5-methoxymethylfurfural. This interesting approach for the production of ML from cellulose allowed achieving maximum ML yields of 53 and 55 mol%, under conventional and MW heating, respectively, and the catalyst maintained almost unaltered its activity after six recycling runs.

Zeolites are highly porous aluminosilicates, exhibiting a well-defined channel and cage-based structure, and, less commonly, a lamellar structure [101]. The molecular dimensions of their channels and cages greatly contribute to their catalytic potential. The catalytic properties of zeolites are due, in part, to the exchangeable ions and water molecules trapped in their structure, as well as their commonly high adsorption capacity. In general, although zeolites are strong acids at high temperatures, their acidity is relatively modest at the common temperature ranges of biomass conversions [101]. However, it is possible to functionalize the zeolites and modulate their existing electronic features, giving rise to zeolites with enhanced acidic properties, which are more suitable for this reaction. The most significant examples of zeolites applied to the ML production are reported in Table 4.

Table 4. Zeolite-based catalysts for the ML production from C6 model carbohydrates.

| Entry | Feedstock | Catalyst | Cat.(g)/MeOH(g) [1] | T(°C) | t (h) | Heat [2] | $Y_{ML}$(mol%) | Ref. |
|---|---|---|---|---|---|---|---|---|
| ML_96 | Glucose | H-β | 0.3/32.4 | 160 | 5 | Conv. | 17 | [82] |
| ML_97 | Cellulose | HZSM-5 | 0.4/71.4 | 300 | 0.02 | Conv. | 12 | [95] |
| ML_98 | Cellulose | HY | 0.4/71.4 | 300 | 0.02 | Conv. | 17 | [95] |
| ML_99 | Fructose | H-USY (6) | 0.6/31.6 | 160 | 20 | Conv. | 51 | [102] |
| ML_100 | Glucose | H-USY (6) | 0.6/31.6 | 160 | 20 | Conv. | 49 | [102] |
| ML_101 | Cellobiose | H-USY (6) | 0.6/31.6 | 160 | 20 | Conv. | 53 | [102] |
| ML_102 | Maltose | H-USY (6) | 0.6/31.6 | 160 | 20 | Conv. | 51 | [102] |
| ML_103 | Inulin | H-USY (6) | 0.6/31.6 | 160 | 20 | Conv. | 50 | [102] |
| ML_104 | Starch | H-USY (6) | 0.6/31.6 | 160 | 20 | Conv. | 31 | [102] |
| ML_105 | Cellulose | H-USY (6) | 0.6/31.6 | 160 | 20 | Conv. | 13 | [102] |
| ML_106 | Fructose | H-USY-0.2 | 0.3/32.4 | 160 | 5 | Conv. | 40 | [82] |
| ML_107 | Glucose | H-USY-0.2 | 0.3/32.4 | 160 | 5 | Conv. | 32 | [82] |
| ML_108 | Mannose | H-USY-0.2 | 0.3/32.4 | 160 | 5 | Conv. | 21 | [82] |
| ML_109 | Sucrose | H-USY-0.2 | 0.3/32.4 | 160 | 5 | Conv. | 38 | [82] |
| ML_110 | Cellobiose | H-USY-0.2 | 0.3/32.4 | 160 | 5 | Conv. | 20 | [82] |
| ML_111 | Glucose | ZrY6 (0.5) | 0.3/40.0 | 180 | 3 | MW | 68 | [103] |
| ML_112 | Mannose | ZrY6 (0.5) | 0.3/40.0 | 180 | 3 | MW | 70 | [103] |
| ML_113 | Galactose | ZrY6 (0.5) | 0.3/40.0 | 180 | 3 | MW | 73 | [103] |
| ML_114 | Sucrose | ZrY6 (0.5) | 0.3/40.0 | 180 | 3 | MW | 78 | [103] |
| ML_115 | Cellobiose | ZrY6 (0.5) | 0.3/40.0 | 180 | 3 | MW | 46 | [103] |
| ML_116 | Starch | ZrY6 (0.5) | 0.3/40.0 | 180 | 6 | MW | 53 | [103] |
| ML_117 | Cellulose | ZrY6 (0.5) | 0.3/40.0 | 180 | 6 | MW | 27 | [103] |
| ML_118 | Glucose | Sn-Al-Beta | 0.6/33.3 | 160 | 5 | Conv. | 41 | [104] |
| ML_119 | Mannose | Sn-Beta + $H_4SiW_{12}O_{40}$ | 0.2 + 0.3/29.0 | 150 | 3 | Conv. | 65 | [105] |
| ML_120 | Starch | Sn-Beta + $H_4SiW_{12}O_{40}$ | 0.2 + 0.3/29.0 | 150 | 5 | Conv. | 58 | [105] |
| ML_121 | Cellulose | Sn-Beta + $H_4SiW_{12}O_{40}$ | 0.2 + 0.3/29.0 | 160 | 10 | Conv. | 62 | [105] |

[1] The amounts of catalysts and MeOH have been normalized to 1 g of feedstock. [2] "Conv." and "MW" stand for "Conventional" and "Microwave", respectively.

Unmodified zeolites, such as H-β, HZSM-5, and HY, were mainly proposed as catalysts of reference, in order to demonstrate the effectiveness of ad hoc synthesized catalysts [82,95]. However, the low ML yields that were obtained with these zeolites (runs ML_96—ML_98, Table 4) show that the modification of their acidic properties is fundamental to improve the ML production. Saravanamurugan et al. [102] demonstrated that unmodified zeolite H-USY (6) can directly transform model mono-, disaccharides, and inulin polysaccharide to ML, with good yields (about 50 mol%) (runs ML_99—ML_103, Table 4). The H-USY (6) zeolite was preferred by the authors for its high content of Lewis acid sites, which facilitated the isomerization of glucose to fructose and the further conversion to ML. Moreover, this zeolite maintained unaltered its structural integrity in the alcohol medium, and it was reused five times without significant changes in the ML yield. However, despite these valuable data, the reported ML yields from more complex feedstocks, e.g., starch and cellulose, were too low (runs ML_104—ML_105, Table 4), indicating that a modification of the catalyst was necessary to improve the catalysis. In fact, the microporosity of the zeolites represents an important drawback for achieving an efficient diffusion of these bulky bio-molecules. In order to solve these shortcomings, some additional mesopores can be introduced into the zeolite framework, for example through nitric acid treatment, which removes the extra framework aluminum species, leading to an increase of mesoporosity and a slight decrease of acidity. Hierarchical H-USY was modified in such a way by Zhou et al. [82], and tested for the conversion of model carbohydrates to ML (runs ML_106—ML_110, Table 4). However, despite the good performances declared by the authors, the comparison with the other data discussed up to now shows that these ML yields for simple model feedstocks are not striking and, even more so, cellulose was not studied. Li et al. [103] proposed the use of a zirconia-zeolite hybrid ZrY6(0.5) catalyst, showing a moderate acid-base site content (0.97 and 0.08 mmol/g), high stability and porosity (average mesopore diameter: 6.2 nm). It was demonstrated that metal content/type and acid-base bifunctionality were closely correlated with substrate conversion and ML yield, respectively. The catalytic activity for ML production was higher than that of other zeolites (runs ML_111—ML_117, Table 4), but the issue of the low ML yield from cellulose remained unsolved (run ML_117, Table 4). Anyway, the catalyst could be reused five times, maintaining stable conversion rates and ML yields. Similarly, Yang et al. [104] enhanced the Lewis acidity of Al-β zeolite by the loading of tin species. Mesopores were generated by the hydrothermal treatment of Sn/Al-β zeolite, via desilication with tetraethylammonium hydroxide, in order to restructure the zeolite and enhance the porosity of Sn-Al-β, which facilitated the diffusion of the reactant and, consequently, the ML production. The dual effects of Lewis acidity and mesoporosity improved ML yield about 2.3 folds from glucose if compared to the parent zeolite. However, the best ML yield from glucose (run ML_118, Table 4) is in agreement with those reported for other discussed zeolites; hence, also in this case, the improvement deriving from the use of this catalyst is not remarkable. Lastly, a bifunctional catalytic system that was composed of commercial homogeneous $H_4SiW_{12}O_{40}$ as the Brønsted HPA, and Sn-Beta zeolite as the Lewis acid, was recently designed by Zhou et al. [105], to catalyze the direct conversion of model mono- and polysaccharides to ML (runs ML_119—ML_121, Table 4). The strong Brønsted acidity and the solubility of HPA, together with the superiority of Sn-Beta zeolite towards the isomerization reaction, make this bifunctional catalyst system highly active for ML production, even when employing cellulose (run ML_121, Table 4, yield to ML 62 mol%), achieving the best catalytic performances within these kinds of catalysts.

Another group of catalysts of great interest is that of montmorillonites (clays), which are aluminosilicates constituted of multiple layers of polyhedrons [106]. Tetrahedral silicon oxide and octahedral hydrous alumina are the common building blocks of these catalysts. Clays have demonstrated remarkable catalytic activity towards many biomass conversion routes, due to their porous structure that provides a unique environment in which molecules can interact in specific ways, allowing for reactions to take place. Other remarkable advantages are high abundance, versatility, smart modulation of the textural properties, and environmental inertness. Table 5 reports the main advances towards the ML synthesis from model compounds in the presence of these catalysts.

**Table 5.** Montmorillonite (clay)–based catalysts the ML production from C6 model carbohydrates, under conventional heating.

| Entry | Feedstock | Catalyst | Cat.(g)/MeOH(g) [1] | T(°C) | t (h) | $Y_{ML}$(mol%) | Ref. |
|---|---|---|---|---|---|---|---|
| ML_122 | Glucose | Al-clay | 0.5/80.0 | 220 | 6 | 61 | [107] |
| ML_123 | Glucose | Cu-clay | 0.5/80.0 | 220 | 6 | 59 | [107] |
| ML_124 | Glucose | In-clay | 0.5/80.0 | 220 | 6 | 52 | [107] |
| ML_125 | Fructose | Sn-clay | 0.5/80.0 | 220 | 6 | 66 | [108] |
| ML_126 | Glucose | Sn-clay | 0.5/80.0 | 220 | 6 | 60 | [108] |
| ML_127 | Sucrose | Sn-clay | 0.5/80.0 | 220 | 6 | 62 | [108] |
| ML_128 | Inulin | Sn-clay | 0.5/80.0 | 220 | 6 | 55 | [108] |
| ML_129 | Starch | Sn-clay | 0.5/80.0 | 220 | 6 | 46 | [108] |
| ML_130 | Cellulose | Sn-clay | 0.5/80.0 | 220 | 6 | 19 | [108] |
| ML_131 | Fructose | $20\text{-}SO_4^{2-}$/clay | 0.8/87.8 | 200 | 4 | 65 | [109] |
| ML_132 | Glucose | $20\text{-}SO_4^{2-}$/clay | 0.8/87.8 | 200 | 4 | 48 | [109] |
| ML_133 | Sucrose | $20\text{-}SO_4^{2-}$/clay | 0.8/87.8 | 200 | 4 | 60 | [109] |
| ML_134 | Starch | $20\text{-}SO_4^{2-}$/clay | 0.8/87.8 | 200 | 4 | 41 | [109] |
| ML_135 | Cellulose | $20\text{-}SO_4^{2-}$/clay | 0.8/87.8 | 200 | 4 | 24 | [109] |
| ML_136 | Fructose | 4-HPWFe-MMTSi [2] | 0.6/24.8 | 180 | 1 | 74 | [110] |

[1] The amounts of catalysts and MeOH have been normalized to 1 g of feedstock. [2] "4-HPWFe-MMTSi" stands for "iron-modified tungstophosphoric acid supported on silica pillared montmorillonite".

Various metal ion-exchanged montmorillonite catalysts were prepared, characterized, and evaluated in glucose conversion to ML (runs ML_122—ML_124, Table 5) [107]. $Al^{3+}$-exchanged montmorillonite gave the best ML yield (61 mol%), due to the presence of a large number of acid sites and a good balance of Brønsted and Lewis acid sites. The montmorillonite catalyst was easily recovered from the reaction mixture by filtration and reused at least five times without any loss of activity/selectivity after treatment with $H_2O_2$ solution in order to remove carbon species deposited on the catalyst surface. Similar results were reported in another work with tin-exchanged montmorillonite catalysts (runs ML_125—ML_130, Table 5) [108]. A high ML yield (60–65 mol%) was obtained when simple monosaccharides, such as glucose or fructose, were adopted as the starting substrate, whilst it significantly decreased to 19 mol%, starting from the insoluble cellulose. In a work conducted byXu et al. [109], montmorillonites were modified by $H_2SO_4$ treatment, thus introducing sulfate groups for improving their acidity and, therefore, catalytic activity for ML production from different model C6 feedstocks (runs ML_131—ML_135, Table 5). Under the optimal conditions, the conversions of fructose and glucose were complete, whilst the corresponding ML yields were 65 and 48 mol% (runs ML_131—ML_132, Table 5). However, also in this case cellulose conversion gave only moderate ML yield (24 mol%, according to run ML_135, Table 5), which restricted the use of this catalyst to simpler soluble carbohydrates. Lastly, silica-pillared montmorillonites functionalized by iron-modified tungstophosphoric acid were prepared by Lai et al. [110] and tested for fructose methanolysis. The characterization demonstrated the high dispersion and Keggin structure of HPWFe in the framework of the MMTSi. An optimized ML yield of around 74 mol% was obtained at 180 °C for 1 h (run ML_136, Table 5), and the catalyst recovered after calcination was active after five recyclings. These catalysts showed advantageous porosity, tuned Brønsted–Lewis acidity, and high thermal stability, making its use promising if compared with other catalysts that belongto this group.

It is well-known that metal oxides can be converted into more stable solid-acid catalysts by the sulfation treatment [111]. These modified oxides show good stability in solvents, also showing enhanced Brønsted and Lewis acidities. Their Lewis acidity is attributed to the metal atoms, whilst the Brønsted one derives from the hydroxyl groups that are located on the surface of the oxides [112]. Generally, the hydroxyl groups on the surface of unmodified metal oxides have very weak Brønsted acidity, but, after having performed a sulfation treatment, the S–O bonds on the sulfuric groups could bind strongly to the metal atoms, forming a coordination of the S=O bond with the surface hydroxyl groups on metal oxides, thus improving the Brønsted acidity. Even these catalysts have been tested for ML production from C6 model carbohydrates, and Table 6 reports the main available data.

**Table 6.** Modified metal oxides and tungsten disulfide catalysts for the ML production from C6 model carbohydrates.

| Entry | Feedstock | Catalyst | Cat.(g)/MeOH(g) [1] | T(°C) | t (h) | Heat [2] | $Y_{ML}$(mol%) | Ref. |
|---|---|---|---|---|---|---|---|---|
| ML_137 | Fructose | TiO$_2$nanop. | 0.6/87.8 | 175 | 1 | Conv. | 80 | [113] |
| ML_138 | Glucose | TiO$_2$nanop. | 0.6/87.8 | 175 | 9 | Conv. | 61 | [113] |
| ML_139 | Starch | TiO$_2$nanop. | 1.1/175.6 | 175 | 20 | Conv. | 40 | [113] |
| ML_140 | Cellulose | TiO$_2$nanop. | 1.1/175.6 | 175 | 20 | Conv. | 42 | [113] |
| ML_141 | Fructose | SO$_4{}^{2-}$/TiO$_2$ | 0.5/15.8 | 200 | 2 | Conv. | 59 | [44] |
| ML_142 | Glucose | SO$_4{}^{2-}$/TiO$_2$ | 0.5/15.8 | 200 | 2 | Conv. | 33 | [44] |
| ML_143 | Sucrose | SO$_4{}^{2-}$/TiO$_2$ | 0.5/15.8 | 200 | 2 | Conv. | 43 | [44] |
| ML_144 | Starch | SO$_4{}^{2-}$/TiO$_2$ | 0.5/15.8 | 200 | 2 | Conv. | 28 | [44] |
| ML_145 | Cellulose | SO$_4{}^{2-}$/TiO$_2$ | 0.5/15.8 | 200 | 2 | Conv. | 10 | [44] |
| ML_146 | Glucose | GraftedSO$_4{}^{2-}$/ZrO$_2$/SBA-15 | 0.5/18.0 | 140 | 24 | Conv. | 24 | [114] |
| ML_147 | Fructose | SO$_4{}^{2-}$/TiO$_2$-ZrO$_2$ | 0.6/87.8 | 200 | 1 | Conv. | 71 | [115] |
| ML_148 | Glucose | SO$_4{}^{2-}$/TiO$_2$-ZrO$_2$ | 0.6/87.8 | 200 | 1 | Conv. | 23 | [115] |
| ML_149 | Sucrose | SO$_4{}^{2-}$/TiO$_2$-ZrO$_2$ | 0.6/87.8 | 200 | 1 | Conv. | 54 | [115] |
| ML_150 | Fructose | SO$_4{}^{2-}$/ZrO$_2$ + Sn-Beta | 0.4 + 0.1/32.4 | 160 | 5 | Conv. | 54 | [83] |
| ML_151 | Glucose | SO$_4{}^{2-}$/ZrO$_2$+ Sn-Beta | 0.4 + 0.1/32.4 | 160 | 5 | Conv. | 59 | [83] |
| ML_152 | Mannose | SO$_4{}^{2-}$/ZrO$_2$ + Sn-Beta | 0.4 + 0.1/32.4 | 160 | 5 | Conv. | 55 | [83] |
| ML_153 | Sucrose | SO$_4{}^{2-}$/ZrO$_2$ + Sn-Beta | 0.4 + 0.1/32.4 | 160 | 5 | Conv. | 31 | [83] |
| ML_154 | Cellobiose | SO$_4{}^{2-}$/ZrO$_2$+ Sn-Beta | 0.4 + 0.1/32.4 | 160 | 5 | Conv. | 37 | [83] |
| ML_155 | Cellulose | NbPO$_4$ | 0.2/19.0 [3] | 180 | 24 | Conv. | 56 | [18] |
| ML_156 | Fructose | WS$_2$ | 0.4/22.2 | 160 | 0.25 | MW | 37 | [116] |

[1] The amounts of catalysts and MeOH have been normalized to 1 g of feedstock. [2] "Conv." and "MW" stand for "Conventional" and "Microwave", respectively. [3] 1.0 g of water were added to this reaction mixture.

Acidic $TiO_2$ (anatase) nanoparticles gave good ML yields, starting from different model carbohydrates (runs ML_137—ML_140, Table 6) [113]. Remarkably, also for the challenging case of cellulose conversion, the best ML yield was good (42 mol%), but this achievement required a very long reaction time (20 h). The good catalytic activity of the nanoparticles was attributed to the *in-situ* solvation on the surface and their better dispersion in the reaction medium. Additionally, in this case, the catalysts showed high recyclability, with only a minor loss of performance. Peng et al. [44] tested many heterogeneous acids (ZSM-5(25), ZSM-5(36), NaY, H-mordenite, $Zr_3(PO_4)_4$, $SO_4^{2-}/ZrO_2$, $SO_4^{2-}/TiO_2$, and $TiO_2$), identifying $SO_4^{2-}/TiO_2$ as the most promising one for ML production (runs ML_141—ML_145, Table 6). Besides, $SO_4^{2-}/TiO_2$ avoided dimethyl ether formation, which, instead, significantly occurred, for example, with ZSM-5(25) and ZSM-5(36) catalysts, which led to a consumption of about half of the starting methanol, in favor of dimethyl ether. The heterogeneous $SO_4^{2-}/TiO_2$ catalyst was easily recovered by filtration and it exhibited good catalytic activities after calcination in five cycles of reusing. The surface structure and acidity variations of the fresh and recycled $SO_4^{2-}/TiO_2$ catalysts after calcination were characterized by XRD and $NH_3$-TPD techniques. The authors reported that the catalyst crystal structure was preserved after multiple cycles, but the amount and strength of the acid sites of the catalyst gradually decreased with the increase of consecutive recycling runs, due to the progressive loss of sulfur, mainly occurring by solvation during the alcoholysis and by calcination necessary for the catalyst regeneration [44]. However, the low ML yield achieved with cellulose (10 mol%, according to run ML_145, Table 6) demonstrated that a further modification of this catalyst was necessary for improving the catalysis. For this purpose, high surface area and thermally robust $SO_4^{2-}/ZrO_2$ conformal monolayers, with tunable Lewis–Brønsted acid site densities, were grown over a mesoporous SBA-15 template, via sequential grafting and hydrolysis cycles, employing a zirconium isopropoxide precursor [114]. The enhanced low-temperature activity of grafted SZ/SBA-15 was attributed to the presence of strong Lewis acid sites that drove the glucose isomerization to fructose. Catalyst reusability was confirmed over three consecutive runs, performing the calcination of the spent catalyst at 550 °C to remove organic deposits, thus overcoming the extended leaching problems of commercial $SO_4^{2-}/ZrO_2$ catalyst. However, the optimized balance between the Brønsted–Lewis acidity and density of the acid sites was not enough for improving the ML catalysis, leading only to a moderate ML yield, starting from glucose and working under mild reaction conditions (run ML_146, Table 6). Njagi et al. proposed further modification of these catalysts [115], who synthesized sulfated mixed-metal oxides ($SO_4^{2-}/TiO_2$–$ZrO_2$) of high acidity, mainly ascribed to the sulfate species, and new acidic sites generated from the charge excess. This catalyst was suitable for the high yield obtained in the conversion of fructose or sucrose to ML (runs ML_147 and ML_ 149, Table 6), but only low yields (23 mol%) were obtained from run ML_148, Table 6),also confirming that this mixed catalyst was not efficient for the isomerization of glucose to fructose. However, the authors declared a high selectivity to ML, which was attributed to the presence of large mesopores, whilst the dimethyl ether formation was reported to be negligible. The spent catalyst was reused after a calcination step for removing insoluble humans from the surface, maintaining its catalytic performances almost unaltered. Another proposal of catalyst improvement was done by Jiang et al. [83], who reported the maximum ML yield of about 60 mol% from fructose, by combining $SO_4^{2-}/ZrO_2$ and Sn-Beta zeolite (run ML_150, Table 6). ML yields for more complex feedstocks are moderate (ML_151—ML_154, Table 6), in agreement with those that were achieved with the zeolite-based catalysts. Sn-Beta zeolite has Lewis acidity for allowing the isomerization of glucose to fructose and poor catalytic activity for the retro-aldol reaction, whilst the Lewis acid sites of $SO_4^{2-}/ZrO_2$ cannot catalyze the isomerization of glucose to fructose. Recyclability studies indicated that the combined catalyst could be reused five times without a significant decrease in product yield, proving its easy recovery and thermal stability during regeneration. Ding et al. synthesized niobium phosphate catalysts [18] and tested for ML production from cellulose, showing good performances in the conversion of cellulose (maximum ML yield of 56 mol%) (run ML_155, Table 6). This investigation showed that the mechanism and type of intermediates of cellulose alcoholysis in MeOH were different from those in water and that the high Brønsted/Lewis

acid ratio of these solid catalysts is needed in order to prevent the generation of by-products, in particular methyl lactate and 1,1,2-trimethoxyethane. Additionally, the heterogeneous system $WS_2$ was tested for fructose conversion. Quereshi et al. synthesized this catalyst [116] in a tubular furnace (600 °C), while using elemental tungsten and sulfur, obtaining multilayered flakes/sheets of $WS_2$. Only a moderate ML yield was reached (run ML_156, Table 6) and the catalyst resulted in being very stable and showed similar activity after five consecutive runs.

*2.2. EL Synthesis from Model Carbohydrates*

The synthesis of EL from C6 model carbohydrates have been widely investigated, and many catalysts have been tested in order to increase the EL yield. The interest in this ester is enhanced by the possible use of bioethanol as alcohol, thus obtaining a fully bio-derived product. As for the synthesis of ML ester, among the employed catalysts, mineral acids and metal salts represent the simplest and cheapest alternatives to sulfuric acid. The most promising data for these kinds of catalysts arereported in Table 7.

Several works have investigated the production of EL from mono- and polysaccharides, preferring the use of diluted sulfuric acid (0.002–0.3 mol/L) (runs EL_1—EL_10, Table 7) [69,117–121]. Temperatures that are higher than 170 °C and relatively short reaction times (below 2 h) have been generally adopted to achieve good EL yields. Taking into account the different feedstocks, as expected fructose led to the highest EL yield (about 70 mol%), due to the easier conversion of this feedstock, in the presence of Brønsted acids (runs EL_1, EL_4—EL_7, Table 7) [117]. A binary reaction medium composed of ethanol-glycerol was proposed for the sulfuric acid-catalyzed conversion of glucoseto inhibit the humin formationviathe use of non-aqueous green solvents, but an improvement of the EL yield was not achieved (run EL_3, Table 7) [119]. In addition, the available data for the conversion of C6 carbohydrates to EL with $H_2SO_4$, confirm that cellulose is the most recalcitrant substrate, leading to the lowest EL yields (runs EL_7—EL_8, Table 7), except when higher catalyst concentrations (0.1–0.3 mol/L) were employed (runs EL_9—EL_10, Table 7) and the reaction conditions were properly modulated [69,117,120,121]. Bernardo et al. proposed perrhenic acid [72] as strong Brønsted homogeneous acid for the synthesis of EL (runs EL_11—EL_14, Table 7).However, this catalyst was more expensive than $H_2SO_4$ and required longer reaction time to achieve interesting yields. Analogously to the reactions that were performed with $H_2SO_4$, the highest EL yield of 80 mol% was obtained starting from fructose after 16 h, whilst under the same conditions the yield from glucose was only 27 mol% and the extension of the reaction time up to 72 h was necessary to reach the same yield obtained starting from fructose. This catalyst was also tested for the EL synthesis from inulin and sucrose, achieving the yields of 65 and 52 mol%, respectively, demonstrating that $HReO_4$ can hydrolyze the glycosidic bonds of di- and polysaccharides. As reported for methanolysis, in addition to the mineral acids, the inorganic salts, in particular sulfates, have also been successfully employed for the EL synthesis from model C6 carbohydrates. Sun et al. [73] investigated different inorganic salts, such as $Fe_2(SO_4)_3$, $La_2(SO_4)_3$ and $Ce(SO_4)_3$ as the catalyst for the alcoholysis of fructose and glucose, working at 170 °C for 2 h (runs EL_15—EL_18, Table 7). It was found that $Fe_2(SO_4)_3$ was the best catalyst for the glucose conversion to EL, leading to the highest EL yield of 39 mol%. Moreover, the EL yield that was obtained starting from glucose was higher than that from fructose, proving that the different chemical structures of these two sugars were affected by the chelation with $Fe_2(SO_4)_3$. For the cellulose alcoholysis $Al_2(SO_4)_3$ was largely employed. Zhou et al. [75] reported the maximum EL yield of 45 mol%, working at 180 °C for 5 h (run EL_19, Table 7), whilst Huang et al. [32] achieved the best EL yield up to 70 mol%, working at the same temperature, but only prolonging the reaction for 0.9 h under MW irradiation with an EtOH-water medium (runs EL_20 and EL_21, Table 7). This yield value from a recalcitrant substrate is remarkable and analogous to that obtained by the authors in the methanolysis of cellulose under the same reaction conditions (compare run ML_29, Table 1, with run EL_21, Table 2). As already discussed for the ML synthesis, the water addition is generally advantageous for this reaction, leading to an increase of the AL yield and

improving the kinetics, at the same time reducing the humin formation and solvent consumption to give the dialkyl ether.

Sulfonic acids and sulfonate salts have been employed as homogeneous or heterogeneous catalysts for the ethanolysis of C6 carbohydrates. Table 8 reports the most interesting available data.

**Table 7.** Inorganic mineral acids and metal salts for the ethyl levulinate (EL) production from C6 model carbohydrates.

| Entry | Feedstock | Catalyst | Cat.(g)/EtOH(g) [1] | T (°C) | t (h) | Heat [2] | $Y_{EL}$ (mol%) | Ref. |
|---|---|---|---|---|---|---|---|---|
| EL_1 | Fructose | $H_2SO_4$ (100 wt%) | 0.04/39.2 | 200 | 1.5 | Conv. | 72 | [117] |
| EL_2 | Glucose | $H_2SO_4$ (100 wt%) | 0.01/39.4 | 180 | 0.5 | Conv. | 45 | [118] |
| EL_3 | Glucose | $H_2SO_4$ (100 wt%) | 0.02/26.5 [3] | 200 | 6 | Conv. | 34 | [119] |
| EL_4 | Glucose | $H_2SO_4$ (100 wt%) | 0.04/39.2 | 200 | 1.5 | Conv. | 37 | [117] |
| EL_5 | Sucrose | $H_2SO_4$ (100 wt%) | 0.04/39.2 | 200 | 1.5 | Conv. | 40 | [117] |
| EL_6 | Inulin | $H_2SO_4$ (100 wt%) | 0.04/39.2 | 200 | 1.5 | Conv. | 51 | [117] |
| EL_7 | Cellulose | $H_2SO_4$ (100 wt%) | 0.04/39.2 | 200 | 1.5 | Conv. | 25 | [117] |
| EL_8 | Cellulose | $H_2SO_4$ (100 wt%) | 0.09/6.9 | 190 | 0.25 | Conv. | 12 | [69] |
| EL_9 | Cellulose | $H_2SO_4$ (100 wt%) | 0.13/11.8 | 190 | 0.5 | Conv. | 43 | [120] |
| EL_10 | Cellulose | $H_2SO_4$ (96 wt%) | 0.42/10.0 | 170 | 2 | Conv. | 51 | [121] |
| EL_11 | Fructose | $HReO_4$ | 0.14/21.7 | 160 | 16 | Conv. | 80 | [72] |
| EL_12 | Glucose | $HReO_4$ | 0.14/21.7 | 160 | 16 | Conv. | 27 | [72] |
| EL_13 | Sucrose | $HReO_4$ | 0.14/21.7 | 160 | 16 | Conv. | 52 | [72] |
| EL_14 | Inulin | $HReO_4$ | 0.14/21.7 | 160 | 16 | Conv. | 65 | [72] |
| EL_15 | Fructose | $Fe_2(SO_4)_3$ | 2.24/9.0 | 170 | 2 | Conv. | 29 | [73] |
| EL_16 | Glucose | $Fe_2(SO_4)_3$ | 2.24/9.0 | 170 | 2 | Conv. | 39 | [73] |
| EL_17 | Glucose | $La_2(SO_4)_3$ | 2.24/9.0 | 170 | 2 | Conv. | 29 | [73] |
| EL_18 | Glucose | $Ce(SO_4)_2$ | 2.24/9.0 | 170 | 2 | Conv. | 29 | [73] |
| EL_19 | Cellulose | $Al_2(SO_4)_3$ | 1.00/34.7 | 180 | 5 | Conv. | 45 | [75] |
| EL_20 | Cellulose | $Al_2(SO_4)_3$ | 0.40/22.0 [4] | 180 | 0.6 | MW | 54 | [32] |
| EL_21 | Cellulose | $Al_2(SO_4)_3$ | 0.40/22.0 [4] | 180 | 0.9 | MW | 70 | [32] |

[1] The amounts of catalyst and EtOH have been normalized to 1 g of feedstock. [2] "Conv." and "MW" stand for "Conventional" and "Microwave", respectively. [3] 14.3 g of glycerol were added to this reaction mixture. [4] 1.2 g of water were added to this reaction mixture.

**Table 8.** Sulfonic acids, sulfonate salts, and their combinations for the EL production from C6 model carbohydrate.

| Entry | Feedstock | Catalyst | Cat.(g)/EtOH(g) [1] | T (°C) | t (h) | Heat [2] | $Y_{EL}$ (mol%) | Ref. |
|-------|-----------|----------|---------------------|--------|-------|----------|------------------|------|
| EL_22 | Cellulose | Al(OTf)$_3$ | 0.47/64.0 | 160 | 4 | Conv. | 32 | [122] |
| EL_23 | Cellulose | In(OTf)$_3$ | 0.56/64.0 | 160 | 4 | Conv. | 20 | [122] |
| EL_24 | Cellulose | Sn(OTf)$_2$ | 0.42/64.0 | 160 | 4 | Conv. | 23 | [122] |
| EL_25 | Cellulose | Hf(OTf)$_4$ | 0.77/64.0 | 160 | 4 | Conv. | 24 | [122] |
| EL_26 | Cellulose | Y(OTf)$_3$ + H$_3$PO$_4$ (100 wt%) | 0.60 + 0.10/64.0 | 180 | 2 | Conv. | 75 | [122] |
| EL_27 | Fructose | PSDVB-SO$_3$H [3] | 0.90/80.0 | 120 | 2 | Conv. | 26 | [123] |
| EL_28 | Fructose | Amberlyst-15 | 0.40/64.0 | 120 | 24 | Conv. | 73 | [80] |
| EL_29 | Fructose | Amberlyst-15 | 0.80/78.0 | 150 | 3.5 | Conv. | 75 | [124] |
| EL_30 | Cellulose | Acid resin D008 | 2.40/10.0 | 170 | 2 | Conv. | 20 | [121] |
| EL_31 | Fructose | PSSA-g-CNT [4] | 0.40/64.0 | 120 | 24 | Conv. | 84 | [80] |
| EL_32 | Fructose | PSSA-g-CNF [5] | 0.40/64.0 | 120 | 24 | Conv. | 69 | [80] |
| EL_33 | Fructose | BSA-g-CMK-5 [6] | 0.40/64.0 | 120 | 24 | Conv. | 60 | [80] |
| EL_34 | Fructose | BSA-g-CNT [7] | 0.40/64.0 | 120 | 24 | Conv. | 45 | [80] |
| EL_35 | Fructose | HDS-3.6 [8] | 0.80/78.0 | 150 | 3.5 | Conv. | 70 | [124] |
| EL_36 | Glucose | HDS-3.6 [8] | 0.40/390.0 | 170 | 8 | Conv. | 25 | [124] |
| EL_37 | Inulin | HDS-3.6 [8] | 0.40/390.0 | 170 | 6 | Conv. | 51 | [124] |
| EL_38 | Starch | HDS-3.6 [8] | 0.40/390.0 | 170 | 8 | Conv. | 18 | [124] |
| EL_39 | Cellulose | HDS-3.6 [8] | 0.40/390.0 | 170 | 10 | Conv. | 12 | [124] |
| EL_40 | Fructose | 5-Cl-SHPAO [9] | 0.25/20.0 | 160 | 1 | Conv. | 68 | [84] |
| EL_41 | Glucose | 5-Cl-SHPAO [9] | 0.25/20.0 | 160 | 8 | Conv. | 61 | [84] |
| EL_42 | Sucrose | 5-Cl-SHPAO [9] | 0.25/20.0 | 160 | 6 | Conv. | 62 | [84] |
| EL_43 | Cellobiose | 5-Cl-SHPAO [9] | 0.25/20.0 | 160 | 8 | Conv. | 60 | [84] |
| EL_44 | Cellulose | 5-Cl-SHPAO [9] | 0.25/20.0 | 160 | 8 | Conv. | 58 | [84] |
| EL_45 | Fructose | AC-Fe-SO$_3$H [10] | 0.50/52.7 | 195 | 3 | Conv. | 58 | [125] |
| EL_46 | Fructose | AC-Fe-SO$_3$H [10] | 0.50/52.7 | 200 | 3 | Conv. | 47 | [125] |
| EL_47 | Glucose | AC-Fe-SO$_3$H [10] | 0.50/52.7 | 200 | 3 | Conv. | 19 | [125] |
| EL_48 | Sucrose | AC-Fe-SO$_3$H [10] | 0.50/52.7 | 200 | 3 | Conv. | 28 | [125] |
| EL_49 | Inulin | AC-Fe-SO$_3$H [10] | 0.50/52.7 | 200 | 3 | Conv. | 35 | [125] |
| EL_50 | Starch | AC-Fe-SO$_3$H [10] | 0.50/52.7 | 200 | 3 | Conv. | 12 | [125] |
| EL_51 | Sucrose | 20 wt% Zn-SC [11] | 0.50/47.3 [12] | 100 | 12 | Conv. | 64 | [126] |
| EL_52 | Sucrose | 20 wt% Zn-SC [11] | 0.50/47.3 [12] | 100 | 1 | US | 72 | [126] |
| EL_53 | Glucose | Sulfonated char | 0.20/26.5 [13] | 200 | 6 | Conv. | 37 | [119] |

[1] The amounts of catalyst and EtOH have been normalized to 1 g of feedstock. [2] "Conv." and "US" stand for "Conventional" and "Ultrasound", respectively. [3] "PSDVB-SO$_3$H" stands for "Polystyrene-co-divinylbenzene resin sulfonated". [4] "PSSA-g-CNT" stands for "poly(*p*-styrenesulfonic acid)-grafted carbon nanotubes". [5] "PSSA-g-CNF" stands for "poly(*p*-styrenesulfonic acid)-grafted carbon nanofibers". [6] "BSA-g-CMK-5" stands for "benzenesulfonic acid-grafted carbon mesostructured by KAIST-5". [7] "BSA-g-CNT" stands for "benzenesulfonic acid-grafted carbon nanotubes". [8] "HDS-3.6" stands for "Hyper-cross-linked polymer-based carbonaceous materials". [9] "5-Cl-SHPAO" stands for "sulfonated hyperbranched poly(aryleneoxindole) with chloride substituent in the fifth position of isatin". [10] "AC-Fe-SO$_3$H" stands for "Fe-impregnated sulfonated carbon". [11] "20 wt% Zn-SC" stands for "20 wt% ZnO doped onto acid-sulfonated carbon". [12] 6.0 g of tetrahydrofuran were added to this reaction mixture. [13] 14.3 g of glycerol were added to this reaction mixture.

Regarding the organic salts, Bodachivskyi et al. [122] studied the EL synthesis from cellulose, in the presence of metal triflates as catalysts (runs EL_22—EL_26, Table 8). In particular, the authors proved that harder Lewis acids, such as $Al(OTf)_3$, $In(OTf)_3$, $Sn(OTf)_2$, and $Hf(OTf)_4$, were able to catalyze this reaction, reaching the maximum EL yield of 32 mol%, with $Al(OTf)_3$. The catalytic performance is ascribed to the Brønsted acidity that is generated from the harder Lewis acids, as a consequence of the complexation of the protic solvent with the metal center, rather than hydrolysis, with the latter being responsible for the Brønsted acidity of inorganic salts. On the other hand, softer Lewis acids, such as $Y(OTf)_3$, $AgOTf$, $La(OTf)_3$, and $Yb(OTf)_3$, showed low activity towards the direct conversion of cellulose to EL, but their combination with a Brønsted acid, such as $H_3PO_4$, significantly increased their catalytic activity, in particular for the combined catalytic system ($Y(OTf)_3$ + $H_3PO_4$) [122]. This synergic effect derived from the favorable complexation of soft Lewis acid and $H_3PO_4$ increased the Lewis acid-assisted Brønsted acidity, leading to the highest EL yield of 75 mol% starting from microcrystalline cellulose. Regarding the other sulfonated systems, both commercial and *ad-hoc* synthesized heterogeneous catalysts have been employed for the production of EL from C6 carbohydrates. For example, Zhang et al. [123] prepared a sulfonic acid resin by the condensation of styrene and divinylbenzene ($PSDVB-SO_3H$), which was employed for the fructose ethanolysis, obtaining the maximum EL yield of 26 mol% (run EL_27, Table 8). However, better EL results have been reported in the presence of commercial styrene-divinylbenzene acid resins. In fact, Liu et al. [80] used the commercial Amberlyst-15 for the fructose ethanolysis, working at 120 °C for 24 h, achieving the highest EL yield of 73 mol% (run EL_28, Table 8). Besides, the same authors prepared several sulfonated carbonaceous materials, such as poly(*p*-styrenesulfonic acid)-grafted carbon nanotubes (PSSA-g-CNT), poly(*p*-styrenesulfonic acid)-grafted carbon nanofibers (PSSA-g-CNF), benzenesulfonic acid-grafted CMK-5 (BSA-g-CMK-5), and benzenesulfonic acid-grafted carbon nanotubes (BSA-g-CNT), employing them under the same reaction conditions of the Amberlyst-15 (runs EL_31—EL_34, Table 8) [80]. The concentration of Brønsted acid sites for the synthesized catalysts decreased, as follows: PSSA-g-CNT > PSSA-g-CNF > BSA-g-CMK-5 > BSA-g-CNT, and the same the trend was observed for EL yield, which was the highest for the PSSA-g-CNT catalyst (84 mol%) and lowest for the BSA-g-CNT one (45 mol%). Moreover, PSSA-g-CNT gave a higher EL yield than that with the commercial Amberlyst-15, underlining that this synthesized catalyst was particularly efficient. Additionally, Gu et al. [124] employed Amberlyst-15 as a catalyst for the fructose ethanolysis at 150 °C for 3.5 h, ascertaining the EL yield of 75 mol% (run EL_29, Table 8). These catalytic performances were compared with those of several *ad-hoc* synthesized sulfonated hyper-cross-linked polymers, working under the same reaction conditions, and the authors proved that the catalyst that was obtained from 4,4′-bis(chloromethyl)-1,1′-biphenyl as the precursor, was the most active, due to the higher acid density and surface area (run EL_35, Table 8) [124]. This catalyst (HDS-3.6) led to the maximum EL yield of 70 mol%, analogous to that ascertained with Amberlyst-15. The synthesized catalyst was also employed for the ethanolysis of other feedstocks, such as glucose, inulin, starch, and cellulose. The EL yields achieved starting from aldose sugars were lower than those obtained starting from ketose, due to the lack of Lewis acid sites in the adopted catalyst, which are of paramount importance for the isomerization step (runs EL_36—EL_39, Table 8) [124]. The same conclusion was reported by Ming et al. [121] for the EL synthesis from cellulose, employing the commercial sulfonic acid resin D008 (run EL_30, Table 8). This catalyst led to the EL yield of only 20 mol%, which was lower than that obtained under the same reaction conditions with the mineral acid $H_2SO_4$ (51 mol%, according to run EL_10, Table 7) [121]. The lower EL yield that was reported with the commercial resin was attributed to the expected mass transfer limitations occurring between the protons of the heterogeneous catalyst and the solid cellulose. The detrimental absence of Lewis acid sites for EL production was confirmed for other synthesized sulfonic acid, such as the 5-chloro-sulfonated hyperbranched poly(aryleneoxindole) catalyst (5-Cl-SHPAO), whichwas tested for the ethanolysis of several saccharides (runs EL_40—EL_44, Table 8) [84]. Under the optimized reaction conditions, the highest EL yield of 68 mol% was obtained from fructose and it decreased, as follows: fructose > sucrose > glucose >cellobiose> cellulose. Zhang

et al. [125] synthesized Fe-impregnated sulfonated carbon of high surface area, pore volume and $-SO_3H$ density, testing it for the ethanolysis of fructose, glucose, sucrose, inulin, and starch (runs EL_45—EL_50, Table 8). Additionally, in this case, the highest EL yield was obtained starting from fructose and the EL yield progressively decreased by converting aldoses of increasing complexity. The problem of the low alkyl levulinate yields obtained from aldose carbohydrates was partially overcome by Karnjanakom et al. [126], who prepared different sulfonated carbon doped with metal oxides, which were tested for the sucrose conversion to EL (runs EL_51—EL_52, Table 8). These catalysts showed higher activity than the corresponding undoped sulfonated carbon, underlining that the synergy between Lewis and Brønsted acidity, derived from oxides and sulfonic groups, respectively, was fundamental for this reaction. In particular, the sulfonated carbon doped with $ZrO_2$ (Zn-SC) showed the highest selectivity towards EL, due to its acidity, whichresulted sufficient to promote the reaction, but not excessive to catalyze also the humin formation. The employment of THF as the reaction co-solvent improved the EL yield from 60 to 72 mol%, because this low-polar solvent prevented the next conversion of the desired EL. The authors also exploited the ultrasound technology as an alternative heating system for promoting this reaction, which was proven to be particularly efficient in reducing reaction times. Bosilji et al. [119] carried out the glucose ethanolysis in an (ethanol-glycerol) solvent, testing a heterogeneous catalyst prepared by hydrothermal carbonization of the same substrate (glucose) itself, through the addition of sodium borate. This is an interesting example, because the adopted feedstock is also the precursor of the catalytic system, in principle making the whole process cheaper and sustainable. The synthesized catalyst had a high specific surface area and led to the maximum EL yield of 37 mol% (run EL_53, Table 8). Moreover, the catalytic performances were comparable with those of $H_2SO_4$ (34 mol%, run EL_3, Table 7).

Polyoxometalates (POMs) are another type of emerging acid catalystadopted for the synthesis of EL from C6 model carbohydrates. Table 9 summarizes the significant results obtained with these catalysts.

**Table 9.** Polyoxometalate (POM)-based catalysts for the EL production from C6 model carbohydrates.

| Entry | Feedstock | Catalyst | Cat.(g)/EtOH(g) [1] | T (°C) | t (h) | Heat [2] | $Y_{EL}$ (mol%) | Ref. |
|---|---|---|---|---|---|---|---|---|
| EL_54 | Fructose | $H_3PW_{12}O_{40}$ | 0.20/39.4 | 160 | 2 | Conv. | 50 | [127] |
| EL_55 | Fructose | $HPW_4Mo_{10}O_x$ | n.a. [3] | 170 | 0.3 | MW | 74 | [128] |
| EL_56 | Glucose | $HPW_4Mo_{10}O_x$ | n.a. [3] | 180 | 0.5 | MW | 62 | [128] |
| EL_57 | Cellulose | $H_4SiW_{12}O_{40}$ | 0.10/31.6 | 180 | 1 | Conv. | 19 | [91] |
| EL_58 | Fructose | $KH_2PW_{12}O_{40}$ | 0.75/35.5 [4] | 150 | 2 | Conv. | 69 | [129] |
| EL_59 | Glucose | $KH_2PW_{12}O_{40}$ | 0.75/35.5 [4] | 150 | 2 | Conv. | 15 | [129] |
| EL_60 | Sucrose | $KH_2PW_{12}O_{40}$ | 0.75/35.5 [4] | 150 | 2 | Conv. | 35 | [129] |
| EL_61 | Inulin | $KH_2PW_{12}O_{40}$ | 0.75/35.5 [4] | 150 | 2 | Conv. | 52 | [129] |
| EL_62 | Cellulose | $KH_2PW_{12}O_{40}$ | 0.75/35.5 [4] | 220 | 2 | Conv. | 15 | [129] |
| EL_63 | Fructose | $Ti_{0.75}PW_{12}O_{40}$ | 0.43/17.5 | 120 | 6 | Conv. | 63 | [130] |
| EL_64 | Glucose | $Ti_{0.75}PW_{12}O_{40}$ | 0.43/17.5 | 120 | 6 | Conv. | 21 | [130] |
| EL_65 | Fructose | $Sn_2SiW_{12}O_{40}$ | 1.67/65.8 | 150 | 2 | Conv. | 71 | [93] |
| EL_66 | Sucrose | $Sn_2SiW_{12}O_{40}$ | 0.91/35.9 | 150 | 2 | Conv. | 78 | [93] |
| EL_67 | Inulin | $Sn_2SiW_{12}O_{40}$ | 2.00/79.0 | 150 | 2 | Conv. | 61 | [93] |
| EL_68 | Fructose | 3-PhPyPW [5] | 0.50/21.7 | 140 | 8 | Conv. | 30 | [99] |
| EL_69 | Fructose | $[TMEDAPS]_3[PW_{12}O_{40}]_2$ [6] | 1.00/64.0 | 120 | 12 | Conv. | 80 | [96] |
| EL_70 | Glucose | $[TMEDAPS]_3[PW_{12}O_{40}]_2$ [6] | 1.00/64.0 | 150 | 24 | Conv. | 20 | [96] |
| EL_71 | Sucrose | $[TMEDAPS]_3[PW_{12}O_{40}]_2$ [6] | 1.00/64.0 | 120 | 12 | Conv. | 45 | [96] |
| EL_72 | Cellobiose | $[TMEDAPS]_3[PW_{12}O_{40}]_2$ [6] | 1.00/64.0 | 150 | 24 | Conv. | 18 | [96] |
| EL_73 | Inulin | $[TMEDAPS]_3[PW_{12}O_{40}]_2$ [6] | 1.00/64.0 | 120 | 12 | Conv. | 67 | [96] |
| EL_74 | Cellulose | $[TMEDAPS]_3[PW_{12}O_{40}]_2$ [6] | 1.00/64.0 | 150 | 24 | Conv. | 14 | [96] |
| EL_75 | Cellulose | $[PyPS]_3PW_{12}O_{40}$ [7] | 10.49/97.5 | 150 | 5 | Conv. | 57 | [97] |

[1] The amounts of catalyst and EtOH have been normalized to 1 g of feedstock. [2] "Conv." and "MW" stand for "Conventional" and "Microwave", respectively. [3] "n.a." stands for "not-available". [4] 4.4 g of toluene were added to this reaction mixture. [5] "3-PhPyPW" stands for "3-phenylpyridine phosphotungstate". [6] "TMEDAPS" stands for "*N,N,N',N'*-tetramethyl-*N,N'*-dipropanesulfonic acid-1,6-hexanediammonium". [7] "PyPS" stands for "1-(3-sulfopropyl)pyridinium".

POMs, including simple HPAs and their salts, have good acidic properties. Different HPAs, such as $H_3PW_{12}O_{40}$ (run EL_54, Table 9) [127], $HPW_4Mo_{10}O_x$ (runs EL_55—EL_56, Table 9) [128], and $H_4SiW_{12}O_{40}$ (run EL_57, Table 9) [91], were used for the EL synthesis from fructose, glucose and cellulose, obtaining satisfactory yields, in particular with the first two catalysts. In fact, analogously to the ML synthesis, $H_4SiW_{12}O_{40}$ strongly promoted the formation of the intermediate ethyl glucoside, leading to a corresponding yield up to 59 mol%and an ethanol conversion of 16 mol% to the diethyl ether by-product.However, the main drawback of the HPA employment is their good solubility in the reaction medium that complicates their separation/recycling. As previously observed for ML, most of the studies have dealt with HPA salts, where one or more protons were substituted with larger cations. In particular, as for the ML synthesis, the $PW_{12}O_{40}{}^{-3}$ heteropolyanion is that preferred for studying the alcoholysis of saccharides. For example, Zhao et al. [129] substituted a proton of the Keggin-type $H_3PW_{12}O_{40}$ with larger monovalent cations, such as $K^+$ ($KH_2PW_{12}O_{40}$) and $Ag^+$ ($AgH_2PW_{12}O_{40}$), thus decreasing the starting Brønsted acidity and making the catalyst insoluble (runs EL_58—EL_62, Table 9). $KH_2PW_{12}O_{40}$ was identified as the best catalyst, leading to a similar maximum EL yield than that achieved with $AgH_2PW_{12}O_{40}$, and involving a cheaper synthesis with KCl precursor, instead of $AgNO_3$. Moreover, the authors proved that the addition of toluene strongly increased the EL yield from fructose, from 51 mol% (with pure EtOH) to 69 mol%, which was attributed to the EL extraction into the toluene phase, which prevented the product degradation. Good EL yields were also ascertained from inulin and sucrose, whilst once again unsatisfactory EL yields were obtained from glucose and cellulose. Similarly, Srinivasa et al. [130] exchanged the protons of $H_3PW_{12}O_{40}$ with titanium, thus adding Lewis acid sites, and reached a maximum EL yield of 63 and 21 mol%, starting from fructose and glucose, respectively (runs EL_63—EL_64, Table 9). Pinheiro et al. [93] synthesized several tin salts of $H_4SiW_{12}O_{40}$, which were almost insoluble in an alcohol medium, and compared their performances towards the EL synthesis, starting from different saccharides (runs EL_65—EL_67, Table 9). The authors demonstrated that $Sn_2SiW_{12}O_{40}$ was the most active catalyst, leading to promising EL yields starting from fructose, sucrose, and inulin, and the proton exchange by $Sn^{2+}$ had a beneficial effect on EL selectivity. The very high yield from sucrose (78 mol%), the disaccharide of fructose and glucose, was ascribed to a contribution of glucose unit, which after has been released on the catalytic site, is directly isomerized to fructose and is then converted to EL. However, when glucose was employed as the starting feedstock, the EL yield was very low (about 5 mol%), probably due to the significant formation of by-products from this substrate [93]. Besides, the replacement of protons of HPAs with organic species was proposed by Fang et al. [99], who synthesized different phosphotungstic acid-based solid hybrids, through the reaction between $H_3PW_{12}O_{40}$ and different pyridines. The 3-phenylpyridine-phosphotungstate (3-PhPyPW) resulted to be the most active catalyst, giving the best EL yield of 30 mol%, starting from fructose (run EL_68, Table 9). Recently, ionic liquids havealso been employed for modifying the acidity, polarity, surface properties, and solubility of POMs. A noteworthy example is provided by Chen et al. [96], who employed the ionic liquid *N*,*N*,*N*′,*N*′-tetramethyl-*N*,*N*′-dipropanesulfonic acid-1,6-hexanediammonium (TMEDAPS) for preparing the corresponding POM salt, [TMEDAPS]$_3$ [$PW_{12}O_{40}$]$_2$, which was tested for the ethanolysis of different substrates, achieving the highest EL yield of 80 mol% from fructose (runs EL_69—EL_74, Table 9). Analogously, Song et al. [97] proposed the use of the 1-(3-sulfopropyl)pyridinium as ionic liquid to synthesize the [PyPS]$_3PW_{12}O_{40}$ catalyst, which gave the best EL yield of 57 mol%, starting from cellulose (run EL_75, Table 9).

The performances of acid zeolite-based catalysts have been studied for the synthesis of EL from C6 carbohydrates, as reported in Table 10.

**Table 10.** Zeolite-based catalysts for the EL production from C6 model carbohydrates, under conventional heating.

| Entry | Feedstock | Catalyst | Cat.(g)/EtOH(g) [1] | T (°C) | t (h) | $Y_{EL}$ (mol%) | Ref. |
|-------|-----------|----------|---------------------|--------|-------|------------------|------|
| EL_76 | Fructose | HY | 0.31/18.2 | 230 | 3 | 53 | [131] |
| EL_77 | Glucose | HY | 1.00/64.0 | 170 | 12 | 39 | [132] |
| EL_78 | Glucose | H-β (19) | 0.60/31.6 | 160 | 20 | 28 | [102] |
| EL_79 | Fructose | H-USY (6) | 0.60/31.6 | 160 | 20 | 40 | [102] |
| EL_80 | Glucose | H-USY (6) | 0.60/31.6 | 160 | 20 | 41 | [102] |
| EL_81 | Mannose | H-USY (6) | 0.60/31.6 | 160 | 20 | 44 | [102] |
| EL_82 | Sucrose | H-USY (6) | 0.63/32.9 | 160 | 20 | 35 | [102] |
| EL_83 | Maltose | H-USY (6) | 0.60/31.6 | 160 | 20 | 47 | [102] |
| EL_84 | Cellobiose | H-USY (6) | 0.63/32.9 | 160 | 20 | 44 | [102] |
| EL_85 | Inulin | H-USY (6) | 0.68/35.9 | 160 | 20 | 39 | [102] |
| EL_86 | Glucose | USY | 0.03/39.4 | 180 | 2 | 47 | [118] |
| EL_87 | Glucose | USY + $H_2SO_4$ (100 wt%) | 0.02 + 0.001/39.4 | 180 | 2 | 51 | [133] |
| EL_88 | Glucose | Sn-β + $ZrH_2PW_{12}O_{40}$ | 0.40 + 0.10/26.4 | 180 | 3 | 54 | [134] |
| EL_89 | Fructose | $H_3PW_{12}O_{40}$/H-ZSM-5 [2] | 1.50/39.2 | 160 | 2 | 43 | [127] |
| EL_90 | Glucose | $H_3PW_{12}O_{40}$/H-ZSM-5 [2] | 1.50/39.2 | 160 | 2 | 19 | [127] |
| EL_91 | Sucrose | $H_3PW_{12}O_{40}$/H-ZSM-5 [2] | 1.50/39.2 | 160 | 2 | 27 | [127] |
| EL_92 | Inulin | $H_3PW_{12}O_{40}$/H-ZSM-5 [2] | 1.50/39.2 | 160 | 2 | 37 | [127] |

[1] The amounts of catalyst and EtOH have been normalized to 1 g of feedstock. [2] "$H_3PW_{12}O_{40}$/H-ZSM-5" stands for "$H_3PW_{12}O_{40}$ supported on H-ZSM-5".

Zeolite HY was tested for the EL synthesis from fructose (run EL_76, Table 10) [131] and glucose (run EL_77, Table 10) [132] reaching yields of 53 and 39 mol%, respectively, under the optimized reaction conditions. The catalytic performances of zeolites H-β and H-USY, having different Si/Al ratios, were compared by Saravanamurugan et al. [102] for the EL synthesis from glucose. Zeolites H-β (Si/Al ratio = 19) and H-USY (Si/Al ratio = 6) gave EL yields of 28 and 41 mol%, respectively (runs EL_78—EL_85, Table 10). Zeolite H-USY (6) was effectively adopted for the ethanolysis of other mono- and polysaccharides, leading to similar EL yields (35–47 mol%). Xu et al. [118] investigated the glucose alcoholysis in the presence of the zeolite USY as the catalyst, and compared its performances with those of $H_2SO_4$, in both cases working at 180 °C (compare run EL_86, Table 10 with run EL_2, Table 7). The authors obtained similar EL yields, but the zeolite USY gave the maximum EL yield (47 mol%) after 2 h, whereas $H_2SO_4$ needed of shorter reaction time (0.5 h) to obtainthe maximum EL yield (45 mol%), being this difference ascribed to mass transfer limitations occurring between the solid zeolite and the liquid phase. However, zeolite USY led to a remarkable higher selectivity to EL, whereas diethyl ether formation was almost negligible. The combined use of zeolites and other co-catalysts has been proposed to better tune the bulk Brønsted-Lewis acidities. For instance, Chang et al. [133] proved that the addition of $H_2SO_4$ to the zeolite USY strongly improved the EL yield from glucose, which increased from 38 to 51 mol% (run EL_87, Table 10). USY zeolite resulted in being efficient for the conversion of glucose to 5-ethoxymethylfurfural, but the stronger Brønsted acidity of $H_2SO_4$ enhanced the overall alcoholysis, improving the EL yield. The zeolites were also employed in combination with POMs. For example, Mulik et al. [134] used Sn-β zeolite with $ZrH_2PW_{12}O_{40}$ for the synthesis of EL from glucose (run EL_88, Table 10), demonstrating that this synergy was fundamental to achieving the best EL yields. The POM salt contributed to the catalysis with both Brønsted and Lewis acidity, but the latter was insufficient, and this lack is offset just by the Sn-Beta zeolite. In particular, the ratio $ZrH_2PW_{12}O_{40}$/Sn-β of 80/20 wt/wt, corresponding to the Brønsted/Lewis ratio of 3.7, gave the best EL yield (54 mol%). Lastly, cheap H-ZSM-5 zeolite was employed as support of high surface area for anchoring the $H_3PW_{12}O_{40}$, thus overcoming the solubility drawback of the latter (runs EL_89—EL_92, Table 10) [127]. However, after the immobilization, the catalytic activity of $H_3PW_{12}O_{40}$ decreased, and the highest EL yield of 43 mol% was achieved, starting from fructose, a lower value than that obtained in the presence of unsupported HPA (50 mol%) (run EL_54, Table 9) [127], although the advantage of this immobilized catalyst lies in its good recyclability. Moreover, this catalytic system was also successfully employed by the authors for the EL synthesis from glucose, sucrose, and inulin, achieving moderate EL yields.

Metal oxides, also in combination with zeolites, resulted in exploitable catalysts for the synthesis of EL from fructose and glucose, and Table 11 reports the most interesting results.

**Table 11.** Metal oxides and tungsten disulfide catalysts for the EL production from C6 model carbohydrates.

| Entry | Feedstock | Catalyst | Cat.(g)/EtOH(g) [1] | T (°C) | t (h) | Heat [2] | $Y_{EL}$ (mol%) | Ref. |
|---|---|---|---|---|---|---|---|---|
| EL_93 | Fructose | $TiO_2$ nanoparticles | 0.56/87.8 | 150 | 3 | Conv. | 71 | [113] |
| EL_94 | Fructose | MCM-41 | 0.80/78.0 | 150 | 3.5 | Conv. | 25 | [124] |
| EL_95 | Fructose | $SO_4^{2-}/ZrO_2$ | 0.80/78.0 | 150 | 3.5 | Conv. | 44 | [124] |
| EL_96 | Glucose | $SO_4^{2-}/ZrO_2$ | 0.02/15.8 | 200 | 3 | Conv. | 29 | [135] |
| EL_97 | Glucose | $SO_4^{2-}/ZrO_2@Al_2O_3$ | 0.50/61.1 | 200 | 5 | Conv. | 37 | [136] |
| EL_98 | Glucose | Grafted $SO_4^{2-}/ZrO_2/SBA$-15 | 0.50/25.6 | 140 | 24 | Conv. | 31 | [114] |
| EL_99 | Glucose | $SO_4^{2-}/ZrO_2$-PMO-$SO_3H$ [2] | 1.00/64.0 | 170 | 12 | Conv. | 42 | [132] |
| EL_100 | Fructose | $Al_2O_3/SBA$-15 | 1.00/78.0 | 190 | 4 | Conv. | 58 | [137] |
| EL_101 | Fructose | DMSi-SA [3] | 0.81/13.3 | 170 | 24 | Conv. | 83 | [138] |
| EL_102 | Glucose | DMSi-SA [3] | 0.81/13.3 | 170 | 24 | Conv. | 62 | [138] |
| EL_103 | Sucrose | DMSi-SA [3] | 0.81/13.3 | 170 | 24 | Conv. | 90 | [138] |
| EL_104 | Glucose | $SnO_2$ + H-USY | 0.03 + 0.48/15.8 | 180 | 3 | Conv. | 81 | [139] |
| EL_105 | Fructose | $WS_2$ | 0.40/21.7 | 160 | 0.5 | MW | 23 | [116] |

[1] The amounts of catalyst and EtOH have been normalized to 1 g of feedstock. [2] "$SO_4^{2-}/ZrO_2$-PMO-$SO_3H$" stands for "Ordered mesoporous sulfonic acid functionalized $ZrO_2$/organosilica".
[3] "DMSi-SA" stands for "Sulfonated dendritic mesoporous silica nanospheres".

Among the commercial metal oxides, $TiO_2$ nanoparticles gave promising results in the EL synthesis from fructose affording the best EL yield of 71 mol% (run EL_93, Table 11) [113]. Gu et al. [124] studied the fructose ethanolysis in the presence of commercial MCM-41 and $SO_4^{2-}$/$ZrO_2$ catalysts, comparing their catalytic performances with those of synthesized sulfonated hyper-cross-linked polymers (HDS-3.6). Both commercial MCM-41 and $SO_4^{2-}$/$ZrO_2$ catalystsled to moderate EL yields, equal to 25 and 44 mol%, respectively (runs EL_94—EL_95, Table 11), whilst the synthesized HDS-3.6 gave the higher EL yield of 70 mol% (run EL_35, Table 8). The lower EL yields with MCM-41 and $SO_4^{2-}$/$ZrO_2$ catalysts were ascribed to the few acid sites and low surface area-pore volume, respectively. Other authors largely investigated the use of $SO_4^{2-}$/$ZrO_2$ for the alcoholysis of carbohydrates. For instance, Peng et al. [135] compared the catalytic activities of $SO_4^{2-}$/$ZrO_2$, $SO_4^{2-}$/$TiO_2$, $SO_4^{2-}$/$ZrO_2$ –$TiO_2$, and $SO_4^{2-}$/$ZrO_2$–$Al_2O_3$ catalysts in the EL synthesis from glucose. $SO_4^{2-}$/$ZrO_2$ and $SO_4^{2-}$/$TiO_2$ exhibited the best activities towards the EL production with 29 mol% yield, but $SO_4^{2-}$/$TiO_2$ increased the formation of undesired diethyl ether, thus $SO_4^{2-}$/$ZrO_2$ resulted in being the best catalyst system for EL production (run EL_96, Table 11). Zhang et al. [136] encapsulated $SO_4^{2-}$/$ZrO_2$ into a mesoporous $Al_2O_3$ ($SO_4^{2-}$/$ZrO_2$ @$Al_2O_3$), obtaining a catalyst with superacid properties. The latter showed both Brønsted and Lewis acid sites, affording the EL yield of 37 mol% (run EL_97, Table 11), which was stable after four recycling tests. Morales et al. [114] grafted $SO_4^{2-}$/$ZrO_2$ on SBA-15. $ZrO_2$ monolayers were grafted on the SBA-15 surface and then the sulfation was carried out with a solution of $H_2SO_4$. The complete coverage of the SBA-15 surface was achieved with two layers of $SO_4^{2-}$/$ZrO_2$ and this catalyst had the best acid density and the appropriate Brønsted/Lewis acid ratio for EL synthesis from glucose in the presence of Lewis acid sites that promote glucose isomerization to fructose. The best EL yield of 31 mol% was obtained with such a catalyst, adopting milder reaction conditions (temperature of 140 °C) (run EL_98, Table 11), but too long reaction time was involved. Song et al. [132] realized the EL synthesis from glucose with an ordered mesoporous sulfonic acid functionalized $ZrO_2$/organosilica catalyst ($SO_4^{2-}$/$ZrO_2$ -PMO-$SO_3H$). This catalyst was characterized by the synergistic effect of super-strong Brønsted acid sites ($SO_4^{2-}$/$ZrO_2$ and –$SO_3H$) and medium Lewis ones ($Zr^{4+}$), and the EL yield of 42 mol% was obtained under the optimized reaction conditions (run EL_99, Table 11). Babaei et al. [137] prepared an alumina-coated mesoporous silica SBA-15 catalyst ($Al_2O_3$/SBA-15), obtaining the maximum EL yield of 58 mol% from fructose (run EL_100, Table 11). When the amount of alumina was increased, the total acidity concentration increased and Lewis/Brønsted ratio decreased, and a lower yield of EL was ascertained, which was ascribed to the humin formation, which is favored by an excess of Brønsted acidity. Besides, the authors proved that the specific surface area of the final catalyst was higher adopting a lower amount of alumina, which guaranteed the better dispersion of the active sites, having a positive effect on the catalytic performances. Jorge et al. [138] prepared SBA-15, KCC-1, MCM-41, and dendritic mesoporous silica nanospheres (DMSi), and the resulting catalyst was functionalized with sulfonic acid. The sulfonated KCC-1, MCM-41, and SBA-15 exhibited moderate catalytic activity towards glucose conversion, obtaining EL yields in the range of 14–19mol%, these resulting significantly lower than that observed with the DMSi-SA catalyst (EL yield of 62 mol%) (run EL_102, Table 11). This catalyst also gave excellent EL yields for fructose and sucrose (EL yields of 83 and 90 mol%, respectively) (runs EL_101 and EL_103, Table 11). The better performances of DMSi-SA catalyst were attributed to its high porosity, which led to a high number of exposed functionalizablesilanol groups. The use of metal oxides in combination with other catalysts was reported by Heda et al. [139], who employed $SnO_2$ with the zeolite H-USY for the synthesis of EL from glucose. Additionally, in this study, the synergy between the different components was fundamental because $SnO_2$ had a high amount of strong Lewis acid sites for glucose-fructose isomerization, while H-USY afforded the proper amount of Brønsted sites. The remarkable yield of 81 mol% was achieved with the catalytic system $SnO_2$+ H-USY (run EL_104, Table 11). Lastly, Quereshi et al. proposedtungsten disulfide [116] for the MW-assisted EL synthesis from fructose, but only a moderate EL yield of 23 mol% was reached (run EL_105, Table 11).

Ionic liquids have been also proposed as efficient catalysts for the synthesis of EL from C6 carbohydrates, but only a few works reported their use for this reaction, due to their high cost, multistep synthesis, and environmental concerns. Table 12 reports the available data.

Saravanamurugan et al. [140] investigated the catalytic performances of several ionic liquids with acidic functionalities for the ethanolysis of fructose and sucrose (runs EL_106—EL_117, Table 12). In particular, they synthesized imidazolium-, pyridinium-, and ammonium-based $SO_3H$ ionic liquids, having $HSO_4^-$ as the anion, and imidazolium-based ionic liquids having $[NTf_2]^-$, $[OMs]^-$ or $[OTf]^-$ as the counter anion. Among these catalysts, $[BMIm-SO_3H][NTf_2]$ gave the highest EL yield of 77 mol% from fructose and 40–43 mol% from sucrose, indicating that only fructose moieties were promptly converted, whilst the isomerization of glucose to fructose was more difficult. In fact, under these reaction conditions, glucose was converted to ethyl-D-glucopyranoside, which is an important non-ionic surfactant that can find applications in cosmetics and pharmaceutical formulations. Amarasekara et al. [141] adopted the Brønsted acid ionic liquid 1-(1-propylsulfonic)-3-methylimidazolium chloride ([PSMIm][Cl]) for the synthesis of EL from cellulose in a (water-EtOH) solution. The authors found that cellulose conversion in sole EtOH was low and highlighted the key role of water co-solvent for promoting the hydrolysis of cellulose to glucose. Once the glucose was formed in the reaction medium, [PSMIm][Cl] catalyzed the dehydration of glucose to HMF, passing through the isomerization to fructose and the synthesis of EL, with a yield of 19 mol% (run EL_118, Table 12).

*2.3. PL Synthesis from Model Carbohydrates*

Table 13 reports the available data for the PL synthesis, including the *n*-PL and *i*-PLisomers, taking into account the C6 model carbohydrates.

**Table 12.** Ionic liquid catalysts for the EL production from C6 model carbohydrates, under conventional heating.

| Entry | Feedstock | Catalyst | Cat.(g)/EtOH(g) [1] | T (°C) | t (h) | $Y_{EL}$ (mol%) | Ref. |
|---|---|---|---|---|---|---|---|
| EL_106 | Fructose | [BMIm-SO$_3$H][HSO$_4$] [2] | 0.11/14.8 | 140 | 24 | 68 | [140] |
| EL_107 | Sucrose | [BMIm-SO$_3$H][HSO$_4$] [2] | 0.11/14.8 | 140 | 24 | 41 | [140] |
| EL_108 | Fructose | [BPyr-SO$_3$H][HSO$_4$] [3] | 0.11/14.8 | 140 | 24 | 70 | [140] |
| EL_109 | Sucrose | [BPyr-SO$_3$H][HSO$_4$] [3] | 0.11/14.8 | 140 | 24 | 43 | [140] |
| EL_110 | Fructose | [NEt$_3$B-SO$_3$H][HSO$_4$] [4] | 0.15/14.8 | 140 | 24 | 74 | [140] |
| EL_111 | Sucrose | [NEt$_3$B-SO$_3$H][HSO$_4$] [4] | 0.15/14.8 | 140 | 24 | 41 | [140] |
| EL_112 | Fructose | [BMIm-SO$_3$H][NTf$_2$] [5] | 0.19/14.8 | 140 | 24 | 77 | [140] |
| EL_113 | Sucrose | [BMIm-SO$_3$H][NTf$_2$] [5] | 0.19/14.8 | 140 | 24 | 43 | [140] |
| EL_114 | Fructose | [BMIm-SO$_3$H][OMs] [6] | 0.11/14.8 | 140 | 24 | 67 | [140] |
| EL_115 | Sucrose | [BMIm-SO$_3$H][OMs] [6] | 0.11/14.8 | 140 | 24 | 40 | [140] |
| EL_116 | Fructose | [BMIm-SO$_3$H][OTf] [7] | 0.15/14.8 | 140 | 24 | 69 | [140] |
| EL_117 | Sucrose | [BMIm-SO$_3$H][OTf] [7] | 0.15/14.8 | 140 | 24 | 42 | [140] |
| EL_118 | Cellulose | [PSMIm][Cl] [8] | 0.33/2.1 [9] | 170 | 12 | 19 | [141] |

[1] The amounts of catalyst and EtOH/co-solvent have been normalized to 1 g of feedstock. [2] "[BMIm-SO$_3$H][HSO$_4$]" stands for "1-methyl-3-(4-sulfobutyl)imidazolium hydrogensulfate". [3] "[BPyr-SO$_3$H][HSO$_4$]" stands for "1-(4-sulfobutyl)pyridiniumhydrogensulfate". [4] "[NEt$_3$B-SO$_3$H][HSO$_4$] " stands for "*N,N,N*-triethyl-4-sulfobutan-ammonium hydrogensulfate". [5] "[BMIm-SO$_3$H][NTf$_2$]" stands for "1-methyl-3-(4-sulfobutyl)imidazolium bis((trifluoromethyl)sulfonyl)amide". [6] "[BMIm-SO$_3$H][OMs]" stands for "1-methyl-3-(4-sulfobutyl)imidazolium methanesulfonate". [7] "[BMIm-SO$_3$H][OTf]" stands for "1-methyl-3-(4-sulfobutyl)imidazolium trifluoromethanesulfonate". [8] "[PSMIm][Cl]" stands for "1-(1-propylsulfonic)-3-methilimidazolium chloride". [9] 1.7 g of water were added to this reaction mixture.

**Table 13.** Catalysts for the PL production from C6 model carbohydrates.

| Entry | Feedstock | Catalyst | Cat.(g)/PrOH(g) [1] | T (°C) | t (h) | Heat [2] | $Y_{PL}$ (mol%) | Ref. |
|---|---|---|---|---|---|---|---|---|
| PL_1 | Cellulose | $H_2SO_4$ (100 wt%) | 0.18/7.0 (*n*) [3] | 190 | 0.25 | Conv. | 35 | [69] |
| PL_2 | Cellulose | $H_2SO_4$ (96 wt%) | 0.42/10.0 (*i*) [4] | 170 | 2 | Conv. | 41 | [121] |
| PL_3 | Fructose | $HReO_4$ | 0.14/21.7 (*i*) [4] | 160 | 16 | Conv. | 22 | [72] |
| PL_4 | Fructose | $Fe_2(SO_4)_3$ | 2.24/9.0 (*i*) [4] | 170 | 2 | Conv. | 61 | [73] |
| PL_5 | Glucose | $Fe_2(SO_4)_3$ | 2.24/9.0 (*i*) [4] | 170 | 2 | Conv. | 55 | [73] |
| PL_6 | Cellulose | $Al_2(SO_4)_3$ | 0.42/22.0 (*i*) [4] | 180 | 1.3 | MW | 54 | [32] |
| PL_7 | Fructose | Amberlyst-15 | 0.40/64.0 (*n*) [3] | 120 | 24 | Conv. | 80 | [80] |
| PL_8 | Cellulose | Acid resin D008 | 2.40/10.0 (*i*) [4] | 170 | 2 | Conv. | 16 | [121] |
| PL_9 | Fructose | PSSA-g-CNT [5] | 0.40/64.0 (*n*) [3] | 120 | 24 | Conv. | 86 | [80] |
| PL_10 | Fructose | PSSA-g-CNF [6] | 0.40/64.0 (*n*) [3] | 120 | 24 | Conv. | 75 | [80] |
| PL_11 | Fructose | BSA-g-CMK-5 [7] | 0.40/64.0 (*n*) [3] | 120 | 24 | Conv. | 68 | [80] |
| PL_12 | Fructose | BSA-g-CNT [8] | 0.40/64.0 (*n*) [3] | 120 | 24 | Conv. | 54 | [80] |
| PL_13 | Cellulose | 5-Cl-SHPAO [9] | 0.25/20.0 (*n*) [3] | 160 | 8 | Conv. | 60 | [84] |
| PL_14 | Glucose | H-USY (6) | 0.60/32.0 (*n*) [3] | 160 | 20 | Conv. | 17 | [102] |
| PL_15 | Fructose | $Sn_2SiW_{12}O_{40}$ | 1.67/66.7 (*n*) [3] | 150 | 2 | Conv. | 74 | [93] |
| PL_16 | Fructose | 3-PhPyPW [10] | 0.50/22.2 (*n*) [3] | 140 | 8 | Conv. | 22 | [99] |
| PL_17 | Fructose | [TMEDAPS]$_3$ [$PW_{12}O_{40}$]$_2$ [11] | 1.00/64.0 (*n*) [3] | 120 | 12 | Conv. | 83 | [96] |
| PL_18 | Cellulose | [PyPS]$_3PW_{12}O_{40}$ [12] | 10.49/98.7 (*n*) [3] | 150 | 5 | Conv. | 37 | [97] |
| PL_19 | Cellulose | [PyPS]$_3PW_{12}O_{40}$ [12] | 10.49/98.7 (*i*) [4] | 150 | 5 | Conv. | 22 | [97] |
| PL_20 | Fructose | $TiO_2$ nanoparticles | 0.56/88.9 (*n*) [3] | 150 | 3 | Conv. | 78 | [113] |
| PL_21 | Fructose | $TiO_2$ nanoparticles | 0.56/88.9 (*i*) [4] | 150 | 3 | Conv. | 13 | [113] |
| PL_22 | Glucose | Grafted $SO_4{}^{2-}$/$ZrO_2$/SBA-15 | 0.50/25.6 (*i*) [4] | 140 | 24 | Conv. | 10 | [114] |

[1] The amounts of catalyst and PrOH have been normalized to 1 g of feedstock. [2] "Conv." and "MW" stand for "Conventional" and "Microwave", respectively. [3] "(*n*)" stands for "*n*-propanol". [4] "(*i*)" stands for "*iso*-propanol". [5] "PSSA-g-CNT" stands for "poly(*p*-styrenesulfonic acid)-grafted carbon nanotubes". [6] "PSSA-g-CNF" stands for "poly(*p*-styrenesulfonic acid)-grafted carbon nanofibers". [7] "BSA-g-CMK-5" stands for "benzenesulfonic acid-grafted carbon mesostructured by KAIST-5". [8] "BSA-g-CNT" stands for "benzenesulfonic acid-grafted carbon nanotubes". [9] "5-Cl-SHPAO" stands for "sulfonated hyperbranched poly(aryleneoxindole) with chloride substituent in the fifth position of isatin". [10] "3-PhPyPW" stands for "3-phenylpyridine phosphotungstate". [11] "TMEDAPS" stands for "*N,N,N′,N′*- tetramethyl-*N,N′*-dipropanesulfonic acid-1,6-hexanediammonium". [12] "PyPS" stands for "1-(3-sulfopropyl)pyridinium".

Garves et al. [69] confirmed similar molar yields for the H$_2$SO$_4$ catalyzed production of *n*-PL and that of the shortest ML, carried under similar reaction conditions (compare run PL_1, Table 13 with run ML_6, Table 1). As observed above for the ML and EL synthesis, the production of *i*-PL from C6 carbohydrates was mainly carried out in the presence of strong Brønsted homogeneous acids, such as H$_2$SO$_4$ (run PL_2, Table 13) [121], HReO$_4$ (run PL_3, Table 13) [72], and with inorganic salts, such as Fe$_2$(SO$_4$)$_3$ (runs PL_4–PL_5, Table 13) [73] and Al$_2$(SO$_4$)$_3$ (run PL_6, Table 13) [32]. However, the use of *i*-PrOH as the solvent/reagent caused a drastic decrease of the *i*-PL yield, with respect to those that were observed with lower alcohols. In fact, the *i*-PL yield was 41, 22 and 54 mol% with H$_2$SO$_4$, HReO$_4$, and Al$_2$(SO$_4$)$_3$, respectively, values that were lower than the EL and ML yields achieved under analogous reaction conditions (runs EL_10–EL_11, Table 7; run ML_11, Table 1; run ML_29, Table 1). Moreover, in the presence of HReO$_4$, also some by-products, such as HMF (4 mol%) and the corresponding 5-alkoxymethylfurfural derivative (20 mol%), were produced, proving that *i*-PrOH is not an excellent solvent for the selective AL synthesis [72]. This can be due to the branched-chain of *i*-PrOH, resulting in a higher steric hindrance. On the other hand, several sulfonic acids have been proposed for the synthesisof both *n*-PL and *i*-PL. The synthesis of *n*-PL from fructose was carried out by Liu et al. [80] in the presence of the commercial Amberlyst-15 resin, achieving the best *n*-PL yield of 80 mol% (run PL_7, Table 13). These catalytic performances were compared with those of synthesized sulfonated carbonaceous materials (PSSA-g-CNT, PSSA-g-CNF, BSA-g-CMK-5, BSA-g-CNT). Promising *n*-PL yields were reached, between 54–86 mol% (runs PL_9–PL_12, Table 13), depending on the adopted catalyst, and comparable with those that were reached employing shorter-chain EtOH (runs EL_31–EL_34, Table 8). Commercial D008 sulfonic acid resin was employed for the synthesis of *i*-PL starting from cellulose [121], but the corresponding molar yield was low (16 mol%, according to run PL_8, Table 13), especially if compared with that reached in the presence of the mineral acid H$_2$SO$_4$ (run PL_2, Table 13). Zhang et al. [84] prepared a new sulfonic acid, the system 5-chloro-sulfonated hyperbranched poly(aryleneoxindole) (5-Cl-SHPAO), which was used for the synthesis of *n*-PL from cellulose, achieving the best yield of 60 mol% (run PL_13, Table 13), a value that is comparable with that obtained from the cellulose ethanolysis (run EL_44, Table 8). Zeolites have been poorly investigated for the synthesis of PL from carbohydrates, leading to unsatisfactory PL yields. For example, zeolite H-USY gave the maximum PL yield of 17 mol% from glucose (run PL_14, Table 13) [102]. On the contrary, POMs have been much considered for the synthesis of the two PL isomers, starting from fructose and cellulose, and obtaining interesting results (runs PL_15–PL_19, Table 13) [93,96,97,99]. In particular, Song et al. [97] compared the performances of [PyPS]$_3$PW$_{12}$O$_{40}$ towards the cellulose alcoholysis, in the presence of *n*-PrOH or *i*-PrOH. The authors obtained the best *n*-PL and *i*-PL yields of 37 and 22 mol%, respectively, thus confirming that the higher steric hindrance of *i*-PrOH negatively affected the alcoholysis reaction. The work of Kuo et al. further supported this conclusion [113], where TiO$_2$ nanoparticles were employed for the fructose alcoholysis, achieving the maximum *i*-PL and *n*-PL yields of 13 and 78 mol%, respectively, working under the same reaction conditions (runs PL_20–PL_21, Table 13). Few other metal oxides have been used for the *i*-PL synthesis from glucose, such as grafted SO$_4^{2-}$/ZrO$_2$/SBA-15, which led to the best yield of 10 mol% (run PL_22, Table 13), lower than that obtained from the ethanolysis reaction (run EL_27, Table 8) [114].

## 2.4. BL and Longer-Chain AL (PeL and HL) Synthesis from Model Carbohydrates

Table 14 reports the most interesting data related to the synthesis of BL, PeL and HL from model C6 carbohydrates.

**Table 14.** Catalysts for the butyl levulinate (BL), pentyl levulinate (PeL), and hexyl levulinate (HL) production from C6 model carbohydrates.

| Entry | Feedstock | Catalyst | Cat.(g)/Alcohol (g) [1] | T (°C) | t (h) | Heat [2] | Y (mol%) | Ref. |
|---|---|---|---|---|---|---|---|---|
| BL_1 | Cellulose | $H_2SO_4$ (100 wt%) | 0.17/7.5 (n) [3] | 210 | 0.17 | Conv. | 40 | [142] |
| BL_2 | Cellulose | $H_2SO_4$ (100 wt%) | 0.18/7.0 (i) [4] | 210 | 0.25 | Conv. | 40 | [69] |
| BL_3 | Cellulose | $H_2SO_4$ (96 wt%) | 25.00/100.0 (n) [3] | 130 | 20 | Conv. | 60 | [143] |
| BL_4 | Cellulose | $H_2SO_4$ (96 wt%) | 42.86/100.0 (n) [3] | 130 | 5 | Conv. | 60 | [143] |
| BL_5 | Cellulose | $H_2SO_4$ (100 wt%) | 0.25/40.0 (n) [3] | 200 | 0.5 | Conv. | 50 | [144] |
| BL_6 | Cellulose | $H_2SO_4$ (100 wt%) | 0.25/40.0 (i) [4] | 200 | 0.5 | Conv. | 45 | [144] |
| BL_7 | Cellulose | $H_2SO_4$ (100 wt%) | 0.25/40.0 (s) [5] | 200 | 0.5 | Conv. | 13 | [144] |
| BL_8 | Fructose | $Fe_2(SO_4)_3$ | 0.20/32.4 (n) [3] | 190 | 3 | Conv. | 63 | [145] |
| BL_9 | Glucose | $Fe_2(SO_4)_3$ | 0.20/32.4 (n) [3] | 190 | 3 | Conv. | 40 | [145] |
| BL_10 | Sucrose | $Fe_2(SO_4)_3$ | 0.20/32.4 (n) [3] | 190 | 3 | Conv. | 50 | [145] |
| BL_11 | Inulin | $Fe_2(SO_4)_3$ | 0.20/32.4 (n) [3] | 190 | 3 | Conv. | 57 | [145] |
| BL_12 | Cellulose | $Fe_2(SO_4)_3$ | 0.20/32.4 (n) [3] | 220 | 3 | Conv. | 30 | [145] |
| BL_13 | Cellulose | $Fe_2(SO_4)_3 + Al_2(SO_4)_3$ | 0.48 + 0.025/40.8 (n) [3] | 194 | 3 | Conv. | 40 | [146] |
| BL_14 | Cellulose | $Cs_2HPW_{12}O_{40}$ | 0.40/40.0 (n) [3] | 200 | 1 | Conv. | 12 | [144] |
| BL_15 | Fructose | 3-PhPyPW [6] | 0.50/22.2 (n) [3] | 140 | 8 | Conv. | 18 | [99] |
| BL_16 | Glucose | Zeolite H-USY (6) | 0.60/32.4 (n) [3] | 160 | 20 | Conv. | 12 | [102] |
| BL_17 | Cellulose | $SO_4^{2-}/ZrO_2$ | 0.40/40.0 (n) [3] | 200 | 1 | Conv. | 13 | [144] |
| BL_18 | Glucose | $SO_4^{2-}/SnO_2$-$ZrO_2$ | 0.44/16.2 (n) [3] | 200 | 2 | Conv. | 33 | [147] |
| BL_19 | Cellulose | $SO_4^{2-}/SnO_2$-$ZrO_2$ | 0.44/16.2 (n) [3] | 200 | 2 | Conv. | 10 | [147] |
| BL_20 | Glucose | $SO_4^{2-}/SnO_2$-$ZrO_2$ + $(COOH)_2$ | 0.44 + 0.01/16.2 (n) [3] | 200 | 2 | Conv. | 36 | [147] |
| BL_21 | Glucose | $SO_4^{2-}/SnO_2$-$ZrO_2$ + $H_2SO_4$ (100 wt%) | 0.44 + 0.01/16.2 (n) [3] | 200 | 2 | Conv. | 40 | [147] |
| BL_22 | Cellulose | $SO_4^{2-}/SnO_2$-$ZrO_2$ + $H_2SO_4$ (100 wt%) | 0.44 + 0.01/16.2 (n) [3] | 200 | 2 | Conv. | 28 | [147] |
| BL_23 | Glucose | $SO_4^{2-}/SnO_2$-$ZrO_2$ + $Fe_2(SO_4)_3$ | 0.44 + 0.01/16.2 (n) [3] | 200 | 2 | Conv. | 35 | [147] |
| BL_24 | Cellulose | $SO_4^{2-}/SnO_2$-$ZrO_2$ + $Fe_2(SO_4)_3$ | 0.44 + 0.01/16.2 (n) [3] | 200 | 2 | Conv. | 18 | [147] |
| BL_25 | Glucose | $SO_4^{2-}/SnO_2$-$ZrO_2$ + $CuSO_4$ | 0.44 + 0.01/16.2 (n) [3] | 200 | 2 | Conv. | 39 | [147] |
| BL_26 | Cellulose | $SO_4^{2-}/SnO_2$-$ZrO_2$ + $CuSO_4$ | 0.44 + 0.01/16.2 (n) [3] | 200 | 2 | Conv. | 19 | [147] |
| BL_27 | Glucose | $SO_4^{2-}/SnO_2$-$ZrO_2$ + PTSA [7] | 0.44 + 0.01/16.2 (n) [3] | 200 | 2 | Conv. | 33 | [147] |
| BL_28 | Cellulose | $SO_4^{2-}/SnO_2$-$ZrO_2$ + PTSA [7] | 0.44 + 0.01/16.2 (n) [3] | 200 | 2 | Conv. | 23 | [147] |
| BL_29 | Cellulose | $[C_4H_8SO_3Hmim]HSO_4$ [8] | 0.10/10.0 (n) [3] | 180 | 0.75 | Conv. | 31 | [148] |
| PeL_1 | Cellulose | $H_2SO_4$ (100 wt%) | 25.4/101.8 | 138 | 3 | Conv. | 69 | [149] |
| PeL_2 | Fructose | $WS_2$ | 0.39/21.7 | 160 | 0.25 | MW | 5 | [116] |
| HL_1 | Cellulose | $H_2SO_4$ (100 wt%) | 25.4/101.8 | 157 | 1 | Conv. | 60 | [149] |
| HL_2 | Glucose | Zn/DFNS [9] | 0.71/58.6 | 200 | 5 | Conv. | 55 | [150] |

[1] The amounts of catalyst and the respective alcohol have been normalized to 1 g of feedstock. [2] "Conv." and "MW" stand for "Conventional" and "Microwave", respectively. [3] "(n)" stands for "n-butanol". [4] "(i)" stands for "iso-butanol". [5] "(s)" stands for "sec-butanol". [6] "3-PhPyPW" stands for "3-phenylpyridine phosphotungstate". [7] "PTSA" stands for "p-toluenesulfonic acid". [8] "$[C_4H_8SO_3Hmim]HSO_4$" stands for "1-(4-sulfobutyl)-3-methyl imidazolium hydrosulfate". [9] "Zn/DFNS" stands for "Zn modified dendritic fibrous nanosilica".

Cellulose butanolysis in the presence of diluted sulfuric acid (about 0.2 mol/L) led to good BL yields, approximately 40 mol%, working under the typical reaction conditions already adopted for the previous shorter-chain ALs (runs BL_1-BL_2, Table 14) [69,142]. The choice of Hishikawa et al. [143] of using much higher concentrations of sulfuric acid (2.0–3.5 mol/L)improved the BL yield up to the maximum of about 60 mol% (runs BL_3-BL_4, Table 14), but such high acid concentrations are not advantageous for the sustainability of the process, additionally causing to corrosion problems of the equipment. Therefore, diluted mineral acids must be certainly preferred, to avoid these important drawbacks. Taking into account the diluted acid approach, the best BL yield (40 mol%, according to runs BL_1—BL_2, Table 14) is similar to the best one obtained for PL (41 mol%, according to run PL_1, Table 13), whilst it is significantly lower for EL (51 mol%, according to EL_10, Table 7) and especially for ML (70 mol%, according to ML_12, Table 1), thus highlighting that the synthesis of ALs in high yield becomes progressively more difficultas the length of the alkyl chain increases. AL yield strongly depends also on the steric hindrance of the alkyl chain, as demonstrated by Démolis et al. [144] forthe cellulose butanolysis. In this context, at first, the authors studied the liquefaction of cellulose in butanol isomers (1-butanol, iso-butanol, *sec*-butanol, *tert*-butanol) in the absence of the acid catalyst, carried out under supercritical conditions (300 °C) and with different reaction times (1 and 2 h), achieving cellulose liquefaction in the range 70–85%, in order to give soluble oligomers and polymers.Despite the similar liquefaction performances of the different butanol isomers, in the case of *tert*-butanol, a significant loss of alcohol (50 wt%) occurred, with the concomitant pressure increase, due to the formation of gaseous products, as a consequence of the significant dehydration of *tert*-butanol to iso-butene, thus anticipating that this butanol isomer is less attractive for developing this reaction. Subsequently, the authors investigated the acid-catalyzed alcoholysis of cellulose in 1-butanol under subcritical conditions, employing very diluted $H_2SO_4$ (0.05 mol/L), and optimizing the reaction temperature and time. The highest BL yield was achieved after 1 h at 200 °C (about 50 mol%), but a significant formation of dibutyl ether occurred (yield of 36 mol%). A reaction time of 0.5 h was found to be the best compromise for reducing the dibutyl ether yield (20 mol%), keeping the BL yield high (50 mol%, according to run BL_5, Table 14). These reaction conditions were considered as those of reference to compare the reactivity of the butanol isomerstowards the cellulose butanolysis. Iso-butanol and *sec*-butanol isomers gave corresponding BL yields of 45 and 13 mol% (runs BL_6—BL_7, Table 14), whilst *tert*-butanol onlyled to traces of BL. Therefore, both 1-butanol and iso-butanol primary alcohols showed better performances than *sec*-butanol secondary one, due to the higher steric hindrance of the latter. In addition, from the complementary perspective of the by-product formation, 1-butanol and iso-butanol primary alcohols gave different dibutyl ether yields of 20 and 2 mol%, respectively, proving the higher reactivity of the linear isomer. Instead, *tert*-butanol was rapidly dehydrated to gaseous olefins, with remarkable loss of this alcohol (about 80 wt%), once again confirming the poor attractiveness of this butanol isomer towards this reaction. Based on the above discussion, iso-butanol represents an excellent alcohol to use for the butanolysis, to develop a new kind of BL-based bio-products. Instead, if *sec*- or *tert*- BLs are desired, the alcoholysis route is not so appropriate, whilst, at the actual state of the art, their synthesis can be more effectively realized by adopting 1-butene or iso-butene as the corresponding alkylating agents and levulinic acid as starting feedstocks, as demonstrated by Démolise t al. [151]. As for the previous ALs, the catalytic activity of a series of cheap metal sulfates was proposed by An et al. [145], instead of the traditional sulfuric acid. Among the investigated metal sulfates, $Fe_2(SO_4)_3$ resulted in being particularly efficient for BL production from simpler model compounds, achieving the maximum yield of about 60 mol%, starting from simple C6 carbohydrates, and about 30 mol%, starting from the more recalcitrant cellulose feedstock (runs BL_8—BL_12, Table 14). However, $Al_2(SO_4)_3$ was also efficient for this purpose, especially in combination with $Fe_2(SO_4)_3$ (run BL_13, Table 14) [146], allowing the enhancement of the BL yield up to the maximum value of about 40 mol%, thus reaching similar performances to those with the sulfuric acid, although much longer reaction times were required in the former case. As for the previous ALs, the use of Cs-exchanged HPAs was also proposed by Démolis et al. [144] for BL synthesis from cellulose, but achieving unsatisfactory BL molar yields (run BL_14,

Table 14). Few other catalysts have been considered, such as 3-PhPyPW or zeolite H-USY (6), but the reported BL molar yields data, already starting from simple monosaccharides, were not particularly noteworthy (runs BL_15—BL_16, Table 14) [99,102]. $SO_4^{2-}/ZrO_2$ was properly modified by Liu et al. [147] to $SO_4^{2-}/SnO_2$-$ZrO_2$, which was used alone, or in combination with organic/inorganic acidsor sulfates. In particular, the combination of $SO_4^{2-}/SnO_2$-$ZrO_2$ with small amounts of$H_2SO_4$, $(HCOOH)_2$, PTSA, $Fe_2(SO_4)_3$ or $CuSO_4$, was effective for improving this reaction (runs BL_17—BL_28, Table 14). Lastly, the good BL yield of about 30 mol% was reported by Ma et al. [148] for the cellulose conversion in the acidic ionic liquid $[C_4H_8SO_3Hmim]HSO_4$ (run BL_29, Table 14), which was identified as a stable and water-tolerant catalyst.

On the other hand, only very few works have studied the synthesis of PeL and HL from C6 model feedstocks, due to their lower reactivity. Yamada et al. [149] carried out the synthesis of PeL and HL starting from cellulose and working at the boiling point of the respective solvent (runs PeL_1 and HL_1, Table 14). Very interesting yields were ascertained for both PeL and HL, equal to 69 and 60 mol%, respectively, but the concentration of $H_2SO_4$ was too high (2.1 mol/L), so that the process could be considered sustainable. Additionally, heterogeneous catalysts have been adopted for the production of PeL and HL. Quereshi et al. [116] carried out the alcoholysis of fructose under MW heating, employing $WS_2$ as the catalyst and PeOH as the solvent/reagent, anyway achieving the PeL yield of only 5 mol% (run PeL_2, Table 14). On the other hand, Mohammadbagheri et al. [150] reported the good HL yield of 55 mol% starting from glucose, in the presence of Zn/dendritic fibrous nanosilica as the catalyst (run HL_2, Table 14), which resulted in being effective for this purpose, due to the balanced Brønsted/Lewis acidity.

## 3. AL Synthesis from Real Biomass

### 3.1. ML Synthesis from Real Biomass

ML production from real biomass has been less investigated with respect to that from model compounds, due to its complexity and consequent much more difficult study of the fate of the large number of components. Table 15 summarizes the best data reported in the literature.

Table 15. Available catalysts for the ML production from real biomasses.

| Entry | Feedstock | Catalyst | Cat.(g)/MeOH(g) [1] | T (°C) | t (h) | Heat [2] | $Y_{ML}$ (wt%) | Ref. |
|---|---|---|---|---|---|---|---|---|
| ML_157 | Bamboo | HCl (37 wt%) | 0.20/8.0 | 180 | 0.5 | MW | 2 | [68] |
| ML_158 | Bamboo | $H_2SO_4$ (96 wt%) | 0.20/5.0 | 180 | 0.5 | MW | 25 | [68] |
| ML_159 | Bamboo | $H_2SO_4$ (96 wt%) | 0.25/8.0 | 180 | 0.67 | MW | 29 | [68] |
| ML_160 | Straw | $H_2SO_4$ (96 wt%) | 0.25/8.0 | 180 | 0.67 | MW | 22 | [68] |
| ML_161 | Eucalyptus | $H_2SO_4$ (96 wt%) | 0.25/8.0 | 180 | 0.67 | MW | 24 | [68] |
| ML_162 | Poplar | $H_2SO_4$ (96 wt%) | 0.25/8.0 | 180 | 0.67 | MW | 25 | [68] |
| ML_163 | Pine | $H_2SO_4$ (96 wt%) | 0.25/8.0 | 180 | 0.67 | MW | 26 | [68] |
| ML_164 | Bagasse | $H_2SO_4$ (96 wt%) | 0.25/8.0 | 180 | 0.67 | MW | 28 | [68] |
| ML_165 | Corn stover | $H_2SO_4$ (96 wt%) | 0.17/5.0 | 160 | 1 | MW | 9 | [152] |
| ML_166 | Corn stover | $H_2SO_4$ (100 wt%) | 0.003/29.8 | 160 | 1 | MW | 9 | [74] |
| ML_167 | BM-corn stover [3] | $H_2SO_4$ (100 wt%) | 0.003/29.8 | 160 | 1 | MW | 11 | [74] |
| ML_168 | Paper sludge | $H_2SO_4$ (100 wt%) | 0.06/15.8 | 222 | 3.58 | Conv. | 27 | [153] |
| ML_169 | *Chlorella sp.* KR-1 | $H_2SO_4$ (100 wt%) | 2.75/6.5 | 130 | 2 | Conv. | 7 | [154] |
| ML_170 | *Nannochloropsisgaditana* | $H_2SO_4$ (100 wt%) | 2.75/6.5 | 130 | 2 | Conv. | 2 | [154] |
| ML_171 | Corn stover | $Al_2(SO_4)_3$ | 0.33/29.8 | 160 | 1 | MW | 8 | [74] |
| ML_172 | BM-corn stover [3] | $Al_2(SO_4)_3$ | 0.33/29.8 | 160 | 1 | MW | 12 | [74] |
| ML_173 | Wheat straw | $CuSO_4$ | 0.16/7.1 | 182 | 3.3 | Conv. | 16 | [55] |
| ML_174 | Pretreated wheat straw [4] | $CuSO_4$ | 0.15/7.1 | 183 | 3.9 | Conv. | 20 | [55] |
| ML_175 | Peanut shells | $Al_2(SO_4)_3$ + $H_2SO_4$ (100 wt%) | 0.51 + 0.02/13.2 | 160 | 3.6 | Conv. | 18 | [155] |
| ML_176 | Bamboo | PTSA [5] | 0.20/8.0 | 180 | 0.5 | MW | 19 | [68] |
| ML_177 | Cedar | Al(acac)$_3$ + PTSA [5] | 0.12 + 0.06/31.6 | 180 | 5 | Conv. | 28 | [79] |
| ML_178 | Eucalyptus | Al(acac)$_3$ + PTSA [5] | 0.12 + 0.06/31.6 | 180 | 5 | Conv. | 25 | [79] |
| ML_179 | Cedar | In(OTf)$_3$ + BSA [6] | 0.04 + 0.06/31.6 | 200 | 5 | Conv. | 31 | [78] |
| ML_180 | Pine | In(OTf)$_3$ + BSA [6] | 0.04 + 0.06/31.6 | 200 | 5 | Conv. | 26 | [78] |
| ML_181 | Eucalyptus | In(OTf)$_3$ + BSA [6] | 0.04 + 0.06/31.6 | 200 | 5 | Conv. | 24 | [78] |
| ML_182 | Bagasse | In(OTf)$_3$ + BSA [6] | 0.04 + 0.06/31.6 | 200 | 5 | Conv. | 25 | [78] |
| ML_183 | Bamboo | $H_3PW_{12}O_{40}$ | 0.20/8.0 | 180 | 0.5 | MW | 15 | [68] |
| ML_184 | Corn straw | $[PyPS]_3PW_{12}O_{40}$ [7] | 3.40/31.6 | 170 | 4.5 | Conv. | 18 | [97] |
| ML_185 | Bagasse | $[PyPS]_3PW_{12}O_{40}$ [7] | 3.40/31.6 | 170 | 4 | Conv. | 14 | [97] |
| ML_186 | Bamboo | $[HSO_3BMIM]HSO_4$ [8] | 0.20/8.0 | 180 | 0.5 | MW | 19 | [68] |
| ML_187 | Duckweed | $[C_3H_6SO_3HPy]HSO_4$ | 0.80/31.6 | 170 | 5 | Conv. | 24 | [156] |

[1] The amounts of catalyst and MeOH have been normalized to 1 g of feedstock. [2] "Conv." and "MW" stand for "Conventional" and "Microwave", respectively. [3] "BM" stands for "Ball-Milled". [4] Wheat straw was pretreated in decanol for fractionating hemicelluloses, and the solid residue in 3.0 wt% NaOH for isolating lignin. [5] "PTSA" stands for "*p*-toluenesulfonic acid". [6] "BSA" stands for "benzenesulfonic acid". [7] "PyPS" stands for "1-(3-sulfopropyl)pyridinium". [8] "[HSO$_3$BMIM]HSO$_4$" stands for "1-methyl-3-(4-sulfobutyl)imidazolium hydrogensulfate".

Several biomass feedstocks, such as bamboo, straw, eucalyptus, poplar, pine, and bagasse, were tested by Feng et al. [68] for the ML production adopting the efficient MW heating, mainly in the presence of diluted sulfuric acid (runs ML_158—ML_164, Table 15), whilst hydrochloric acid resulted in being inefficient for this purpose (run ML_157, Table 15). All of these biomasses have very similar cellulose content, approximatively 40 wt%, and this similarity makes easy the comparison of the ML yield results, expressed as wt% with respect to the starting weight of dry biomass. About 80–85 wt% of the starting biomass was liquefied within 40 min. of reaction, achieving, in most cases, ML yields within the range 20–30 wt%, and negligible dimethyl ether yields, due to the low sulfuric acid concentration, whilst other by-products, such as furfurals (5-methoxymethyl furfural and furfural) and methyl glucoside, were significatively produced, with corresponding yields of about 15–20 and 10–20 wt%(runs ML_157—ML_164, Table 15). Among the investigated biomasses, bamboo was better liquefied and it gave the highest ML yield (run ML_159, Table 15) and, for this reason, it was studied more in-depth, when comparing the hydrolysis and methanolysis routes. Approximately 84 wt% of the bamboo biomass was converted into liquefied products at 180 °C with MeOH, whilst only 25 wt% of bamboo was converted with water medium. The selective catalytic conversion of biomass was found to be efficient for the ML production (reaching a maximum yield of approximately 30 wt%), higher than the levulinic acid yield (14 wt%), with the latter obtained carrying out the reaction in aqueous solution, under the same reaction conditions. The authors also demonstrated that PTSA, $H_3PW_{12}O_{40}$ or [$HSO_3BMIM$]$HSO_4$ ionic liquid gave acceptable ML yields (runs ML_176, ML_183, and ML_184, Table 15), and confirmed the key role of MeOH for the solubilization of the high molecular weight polar products, due to their low dielectric constants, which could efficiently prevent the re-polymerization and re-condensation of liquefied products on the surface of biomass itself, thus significantly improving the diffusion and reactivity of the alcohol. However, also in the presence of these catalysts, the corresponding yields of furfurals and methyl glucoside were not negligible, in particular 20 and 18 wt% with PTSA, 13 and 16 wt% with $H_3PW_{12}O_{40}$, and 14 and 19 wt% with [$HSO_3BMIM$]$HSO_4$ [68]. Xiao et al. [152] proposed the sulfuric acid-catalyzed conversion of corn stover, reporting a maximum ML yield of 9 wt%, under MW irradiation (run ML_165, Table 15). Despite the noteworthy positive effect of MW towards ML production, the obtained ML yields are not competitive with those that were obtained from the above considered biomasses. Even the addition of a preliminary ball-milling treatment did not allow for the improvement of the ML yield from corn stover (runs ML_165—ML_167, Table 15) [74], which is not a suitable feedstock for this reaction. Peng et al. proposed the conversion of a paper sludge [153], in the presence of a very low concentration of sulfuric acid (≤0.05 mol/L). Response Surface Methodology (RSM) with a four-factor, five-level central composite rotatable design was employed to optimize the reaction. The yields of ML and dimethyl ether, as a function of the process variables, were fitted to second-order polynomial models through the application of multiple regression analyses, achieving a good agreement between the experimental and modeled data. When the dimethyl ether yield was lower than 20 mol%, a maximum ML yield of about 55 mol% was achieved, corresponding to a mass yield of 27 wt%, calculated with respect to the weight of dry paper sludge (run ML_168, Table 15). The authors concluded that sulfuric acid concentration and temperature were crucial for increasing ML production and reducing that of dimethyl ether [153]. As examples of unconventional biomass feedstocks for ML production, *Nannochloropsis* and *Chlorella* microalgal strains were compared by Kim et al. [154], leading to low ML yields (2–7 wt%) (runs ML_168—ML_170, Table 15). As already discussed for the model C6 feedstocks, inorganic salts can represent a good and cheap alternative to the active sulfuric acid. The good performances of $Al_2(SO_4)_3$ were demonstrated by Chen et al. [74] for the conversion of raw and ball-milled corn stover (runs ML_171—ML_172, Table 15), resulting in agreement with the best results that were obtained with sulfuric acid (runs ML_166—ML_167, Table 15). Wheat straw was effectively converted into ML, in the presence of $CuSO_4$ as a cheap and active catalyst (runs ML_173—ML_174, Table 15) [55]. In this regard, metal sulfates of the IA and IIA groups, such as $Na_2SO_4$, $MgSO_4$, $K_2SO_4$ and $CaSO_4$, were not active towards ML production, due to their too low acidities, as well as $ZnSO_4$, $MnSO_4$, and

$NiSO_4$. On the other hand, $Al_2(SO_4)_3$, $Fe_2(SO_4)_3$, $Ti_2(SO_4)_3$, and, especially, $CuSO_4$ exhibited good catalytic activity, due to the contemporary presence of Lewis acid sites from metal ions and Brønsted ones, whose formation occurs via the hydrolysis/methanolysis of metal ions. The authors developed a two-stage pretreatment process, where the hemicellulose fraction was first converted into decyl pentoside bio-surfactant by an acidic decanol-based pretreatment, carried out under mild conditions and, subsequently, the lignin component was extracted in the second stage by sodium hydroxide treatment [55]. After this fractionation, the residual wheat straw, resulting enriched in cellulose, was used as starting feedstock for ML production, and a maximum yield of 20 wt% was obtained, under the optimized reaction conditions (run ML_174, Table 15). Taking into account the interactions between reaction time and reaction temperature or catalyst dosage, the choice of long reaction time was not in favor of the process. Moreover, the catalyst recycling experiments showed that copper sulfate was stable, and it can be reused more than five times. The combination of sulfuric acid with extremely low concentration and $Al_2(SO_4)_3$ was identified as the efficient mixed acid catalytic system for ML production from waste peanut shells, reaching a ML yield of 17 wt% (run ML_175, Table 15) [155]. Additionally, the combination of PTSA and $Al(acac)_3$ was found to be an efficient catalytic system, applied to the conversion of cedar and eucalyptus feedstocks (runs ML_177—ML_178, Table 15) [79]. The solvolysis of the C6 fraction of the biomass was catalyzed by Brønsted acidic PTSA and the conversion of these sugars to ML was significantly enhanced by Lewis acidic $Al(OTs)_3$, formed in situ from Al compounds and PTSA. Among metal triflates of group 13, $In(OTf)_3$, used in combination with BSA, resulted in being particularly efficient for the ML production, from lignocellulosic biomasses as cedar, pine, eucalyptus and bagasse (runs ML_179—ML_182, Table 15) [78]. Remarkably, the highest ML yield (31 wt%) was achieved from cedar, working at 200 °C for 5 h (run ML_179, Table 15). As above reported for the conversion of model compounds, heteropolyanion-based ionic liquid $[PyPS]_3PW_{12}O_{40}$ was proposed by Song et al. [97] as an efficient catalyst for the conversion of corn straw and bagasse into ML, with yields of 18 and 14 wt%, respectively (runs ML_184—ML_185, Table 15). As previously reported for the alcoholysis of model compounds, the ascertained catalytic performances were ascribed to the high acidic strength of the sulfonic-functionalized cation and its synergistic effects with the corresponding heteropolyanion, which favored the whole process. Duckweed, a typical fast-growing aquatic microalgae, was converted into ML in the presence of acidic ionic liquid $[C_3H_6SO_3HPy]HSO_4$ (run ML_187, Table 15) [156]. Under the best reaction conditions (170 °C, 5 h), about 88% of the starting duckweed feedstock was converted, leading to the maximum ML yield of 24 wt% and levulinic acid yield of 2 wt%, with these good catalytic performances being partly attributed to the low lignin content of the starting biomass. In agreement with the previous data, it was confirmed that the solvent had a remarkable intensified effect on the process efficiency (as levulinic acid yield plus ML yield), which dramatically decreased from 82 to 54 mol% when MeOH was replaced by water.

### 3.2. EL Synthesis from Real Biomass

Up to now, also for the EL synthesis from real biomasses, homogeneous catalysts have been preferred over heterogeneous ones. Table 16 reports the most promising available data.

**Table 16.** Catalysts for the EL production from real biomasses.

| Entry | Feedstock | Catalyst | Cat.(g)/EtOH(g) [1] | T (°C) | t (h) | Heat [2] | $Y_{EL}$ (wt%) | Ref. |
|---|---|---|---|---|---|---|---|---|
| EL_119 | Grey pine wood | $H_2SO_4$ (96 wt%) | 0.04/5.0 | 190 | 1.6 | Conv. | 16 | [157] |
| EL_120 | Paper pulp | $H_2SO_4$ (96 wt%) | 0.04/5.0 | 190 | 1.6 | Conv. | 26 | [157] |
| EL_121 | Switchgrass | $H_2SO_4$ (96 wt%) | 0.04/5.0 | 190 | 1.6 | Conv. | 14 | [157] |
| EL_122 | Wheat straw | $H_2SO_4$ (100 wt%) | 0.51/19.8 | 183 | 0.6 | Conv. | 18 | [158] |
| EL_123 | Chipped laminated particleboard | $H_2SO_4$ (96 wt%) | 0.19/7.4 | 200 | 0.5 | Conv. | 24 | [159] |
| EL_124 | *Chlorella sp.* KR-1 | $H_2SO_4$ (100 wt%) | 2.75/6.5 | 130 | 2 | Conv. | 11 | [154] |
| EL_125 | *Nannochloropsisgaditana* | $H_2SO_4$ (100 wt%) | 2.75/6.5 | 130 | 2 | Conv. | 3 | [154] |
| EL_126 | Mandarin peels | $H_2SO_4$ (100 wt%) | 0.90/5.3 [3] | 150 | 2 | Conv. | 28 | [160] |
| EL_127 | Cassava | $H_2SO_4$ (96 wt%) | 0.25/3.9 [4] | 160 | 3 | Conv. | 14 | [161] |
| EL_128 | Cassava [5] | $H_2SO_4$ (96 wt%) | 0.25/3.9 [4] | 160 | 3 | Conv. | 21 | [161] |
| EL_129 | Cassava [5] | $H_2SO_4$ (96 wt%) | 0.17/5.3 [6] | 160 | 5 | Conv. | 27 | [161] |
| EL_130 | Bamboo | $H_2SO_4$ (96 wt%) | 0.04/30.1 | 210 | 2.1 | Conv. | 51 | [162] |
| EL_131 | Bamboo [7] | $H_2SO_4$ (96 wt%) | 0.04/30.1 | 210 | 2.1 | Conv. | 63 | [162] |
| EL_132 | Corn stover | $H_2SO_4$ (96 wt%) | 0.42/15.0 | 190 | 0.5 | Conv. | 7 | [163] |
| EL_133 | Corn stover | $H_2SO_4$ (96 wt%) | 0.42/15.0 | 190 | 0.5 | MW | 17 | [163] |
| EL_134 | Corn stover | $H_2SO_4$ (96 wt%) | 0.20/20.0 | 180 | 0.5 | MW | 13 | [164] |
| EL_135 | Corn stover [8] | $H_2SO_4$ (96 wt%) | 0.20/20.0 | 180 | 0.5 | MW | 14 | [164] |
| EL_136 | Corn stover | $H_2SO_4$ (96 wt%) | 0.15/5.0 | 180 | 0.5 | MW | 12 | [25] |
| EL_137 | Wheat straw | $H_2SO_4$ (96 wt%) | 0.15/5.0 | 180 | 0.5 | MW | 11 | [25] |
| EL_138 | Rice straw | $H_2SO_4$ (96 wt%) | 0.15/5.0 | 180 | 0.5 | MW | 11 | [25] |
| EL_139 | Rape straw | $H_2SO_4$ (96 wt%) | 0.15/5.0 | 180 | 0.5 | MW | 6 | [25] |
| EL_140 | Poplar wood | $H_2SO_4$ (96 wt%) | 0.15/5.0 | 180 | 0.5 | MW | 9 | [25] |
| EL_141 | Cassava | $H_2SO_4$ (96 wt%) | 0.10/19.0 | 200 | 6 | Conv. | 31 | [165] |
| EL_142 | Cassava | $NaHSO_4$ | 0.10/19.0 | 200 | 6 | Conv. | 15 | [165] |
| EL_143 | Cassava | $Al_2(SO_4)_3$ | 0.10/19.0 | 200 | 6 | Conv. | 36 | [165] |
| EL_144 | Cassava | $Fe_2(SO_4)_3$ | 0.10/19.0 | 200 | 6 | Conv. | 9 | [165] |
| EL_145 | DHFW [9] | $H_2SO_4$ (96 wt%) +$AlCl_3 \cdot 6H_2O$ | 0.16 + 0.16/15.8 | 180 | 4 | Conv. | 15 | [166] |
| EL_146 | KW [10] | $H_2SO_4$ (96 wt%) +$AlCl_3 \cdot 6H_2O$ | 0.16 + 0.16/15.8 | 180 | 4 | Conv. | 32 | [166] |
| EL_147 | FVS [11] | $H_2SO_4$ (96 wt%) +$AlCl_3 \cdot 6H_2O$ | 0.16 + 0.16/15.8 | 180 | 4 | Conv. | 11 | [166] |
| EL_148 | OFMSW [12] | $H_2SO_4$ (96 wt%) +$AlCl_3 \cdot 6H_2O$ | 0.16 + 0.16/15.8 | 180 | 4 | Conv. | 14 | [166] |
| EL_149 | Coniferous wood | 1,5-NSA [13] | 0.20/7.9 | 200 | 4 | Conv. | 46 | [167] |
| EL_150 | Coniferous wood | 2-NSA [14] | 0.20/7.9 | 200 | 4 | Conv. | 49 | [167] |
| EL_151 | Furfural residue | USY + $H_2SO_4$ (96 wt%) | 0.03 + 0.04/39.9 | 219 | 1.8 | Conv. | 19 | [133] |
| EL_152 | Wheat straw | [$HSO_3$-BMIM][$HSO_4$] | 0.26/15.0 | 200 | 1 | Conv. | 16 | [168] |
| EL_153 | OPEFB [15,16] | [HMIM][$HSO_4$] | n.a. [17] | 90 | 12 | Conv. | 12 | [169] |

**Table 16.** *Cont.*

| Entry | Feedstock | Catalyst | Cat.(g)/EtOH(g) [1] | T (°C) | t (h) | Heat [2] | $Y_{EL}$ (wt%) | Ref. |
|---|---|---|---|---|---|---|---|---|
| EL_154 | OPMF [18,16] | [HMIM][HSO₄] | n.a. [17] | 90 | 12 | Conv. | 14 | [169] |
| EL_155 | OPEFB [15,19] | [HMIM][HSO₄] + InCl₃ | n.a. [17] | 90 | 10 | Conv. | 13 | [170] |
| EL_156 | OPMF [18,19] | [HMIM][HSO₄] + InCl₃ | n.a. [17] | 90 | 10 | Conv. | 15 | [170] |
| EL_157 | OPEFB [15,20] | [HMIM][HSO₄] + InCl₃ | n.a. [17] | 105 | 12.2 | Conv. | 19 | [171] |
| EL_158 | OPMF [18,20] | [HMIM][HSO₄] + InCl₃ | n.a. [17] | 105 | 12.2 | Conv. | 21 | [171] |
| EL_159 | Corn stover [21] | [HSO₃-BMIM][HSO₄] + Al₂(SO₄)₃ | 0.51 + 0.14/20.0 | 170 | 2 | MW | 11 | [172] |

[1] The amounts of catalyst and EtOH have been normalized to 1 g of feedstock. [2] "Conv." and "MW" stand for "Conventional" and "Microwave", respectively. [3] 10.0 g of chloroform were added to this reaction mixture. [4] 4.8 g of water were added to this reaction mixture. [5] The biomass was pre-hydrolyzated at 100 °C for 1 h. [6] 3.2 g of water were added to this reaction mixture. [7] The biomass was pretreated in water with MgO under O2 pressure (1 MPa) at 170 °C for 3 h. [8] The biomass was pretreated by 120 min. of ball milling. [9] "DHFW" stands for "Dried household food waste". [10] "KW" stands for "Kitchen waste". [11] "FVS" stands for "Fruit and vegetables scraps". [12] "OFMSW" stands for "Organic fraction of municipal solid waste". [13] "1,5-NSA" stands for "1,5-naphthalenesulfonic acid". [14] "2-NSA" stands for "2-naphthalenesulfonic acid". [15] "OPEFB" stands for "Oil palm empty fruit bunch". [16] The biomass was previously depolymerized at 160 °C for 3 h. [17] "n.a." stands for "not-available". [18] "OPMF" stands for "Oil palm mesocarp fiber". [19] The biomass was previously depolymerized at 160 °C for 5 h. [20] The biomass was previously depolymerized at 177 °C for 4.8 h. [21] The biomass was pretreated by 60 min. of ball milling.

Additionally, in this case, sulfuric acid is the catalyst of greatest practical interest. Different biomasses, such as wheat straw, chipped laminated particleboard, algae, cassava, bamboo, grey pine wood, paper pulp, switchgrass, mandarin peels, corn stover, rice straw, rape straw, and poplar wood, have been adopted as starting feedstocks, in the presence of $H_2SO_4$ as the catalyst, achieving EL yields in the range 6–63 wt%, evaluated with respect to the weight of the starting dry biomass. This wide range of the best EL yields depends on the different cellulose content of the adopted biomasses. Le Van Mao et al. [157] compared the ethanolysis of three different biomasses (grey pine wood, paper pulp, switchgrass) (runs EL_119–EL_121, Table 16), demonstrating that EL yield decreased in the order paper pulp > grey pine wood > switchgrass, retracing the same trend of cellulose amount in the three biomasses, equal to 78, 42, and 35 wt%, respectively. The authors also confirmed the presence of some interesting reaction by-products, such as ethyl formate, with yields of 8, 7, and 5 wt%, starting from paper pulp, grey pine wood and switchgrass, respectively, and diethyl ether, with corresponding yields of 3, 2, and 2 wt%, both properly exploitable after their efficient separation. Chang et al. [158] carried out the conversion of wheat straw to EL, optimizing the reaction conditions through a Box-Behnken experimental design, claiming the highest EL yield of 18 wt%, working at 183 °C for 36 min. with much more concentrated acid (run EL_122, Table 16). Olson et al. [159] performed the ethanolysis of chipped laminated particleboard, reaching an optimal EL yield of 24 wt% (run EL_123, Table 16). Starting from the results of ML synthesis, Kim et al. [154] alsostudied that of EL from *Chlorella* and *Nannochloropsis* microalgal strains, achieving the EL yield of 11 and 3 wt%, respectively, with this difference of reactivity being ascribed to the higher carbohydrate content in the *Chlorella* strain (runs EL_124–EL_125, Table 16). Yang et al. [160] proposed the conversion of mandarin peels, employing chloroform as co-solvent for increasing the EL yield (from 4 to 28 wt%, under the optimized reaction conditions) (run EL_126, Table 16). This improvement was ascribed to the higher solubilization of EL in chloroform, rather than in the (water-EtOH) system, where chloroform enables as a continuous extraction medium for EL, at the same time limiting the humin formation. EL yield was also improved by adopting appropriate biomass pre-treatments of the starting raw biomass in order to overcome its recalcitrance and improve the interaction between the catalyst and the cellulose. On this basis, Zhao et al. [161] pre-hydrolyzed cassava biomass at 100 °C for 1 h and, subsequently, carried out its ethanolysis at 160 °C for 3 h, when comparing the results with those achieved starting from the untreated cassava (runs EL_127–EL_129, Table 16). The authors found that, when cassava was pre-hydrolyzed, the maximum achieved EL yield was higher, and the EL yield from pre-hydrolyzed cassava was further increased up to the maximum value of 27 wt%, after having properly optimized the reaction conditions. Gong et al. [162] hydrothermally pre-treated bamboo biomass, working in the presence of MgO and $O_2$ pressure (1 MPa), at 170 °C for 3 h. As a consequence of this pre-treatment, the authors obtained a significant removal of the lignin fraction, thus making the next sulfuric acid-catalyzed ethanolysis step easier (runs EL_130–EL_131, Table 16). Zhang et al. [163] investigated the corn stoverethanolysis, exploiting the efficient MW heating, which improved the EL production (17 wt%) with respect to the conventional heating (7 wt%) (runs EL_132–EL_133, Table 16). Additionally, Liu et al. [164] preferred the MW heating for studying the ethanolysis of ball-milled corn stover, adopting similar reaction conditions of Zhang et al. [163], anyway not obtaining appreciable improvements in the EL yield (runs EL_134–EL_135, Table 16). Zhao et al. [25] compared the MW-assisted conversion of different biomasses (corn stover, wheat straw, rice straw, rape straw, poplar wood) (runs EL_136–EL_140, Table 16). In this work, the authors proved that EL yield depended not only on the cellulose content, but also on the crystallinity index of the cellulose fraction. In fact, the cellulose amount decreased, as follows: rice straw ≈ poplar wood > corn stover> wheat straw > rape straw. However, the EL yield decreased, as follows: corn stover> rice straw ≈ wheat straw > poplar wood > rape straw. In particular, the EL yield from poplar wood was lower than that obtained from rice straw, despite themhavinga similar cellulose amount (37 wt%), and this difference was explained with the higher crystallinity index of poplar wood, leading to a lower EL yield. The bifunctional Bronsted–Lewis acid behavior of inorganic salts has also been exploited for the EL synthesis from real biomasses. Tan et al. [165] compared the catalytic

activity of H$_2$SO$_4$ and several inorganic salts towards the ethanolysis of cassava (runs EL_141–EL_144, Table 16). The authors found that Al$_2$(SO$_4$)$_3$gave the highest EL yield of 36 wt%, thanks to its better synergistic effect between Brønsted and Lewis acidity, generated from the salt hydrolysis/ethanolysis. Analogously to the ethanolysis of model compounds, also for the real biomasses, H$_2$SO$_4$ was employed in combination with inorganic salts, for example, by Di Bitonto et al. [166], who choose the catalytic system (H$_2$SO$_4$ + AlCl$_3$·6H$_2$O) for the conversion of organic wastes, such as dried household food waste (DHFW), kitchen waste, fruit (KWF), fruit and vegetable scraps (FVS), and organic fractions of municipal solid waste (OFMSW), obtaining the maximum EL yields of 15, 32, 11, and 14 wt%, respectively (runs EL_145–EL_148, Table 16).

Other catalysts, such as sulfonic acids, zeolites, and ionic liquids, have been tested for this reaction. Regarding the sulfonic acids, such as 1,5-NSA and 2-NSA, Bianchi et al. discussedtheir use for the coniferous wood ethanolysis [167]. The authors claimed the highest EL yields of 46 and 49 wt%, employing 1,5-NSA acid and 2-NSA, respectively (runs EL_149–EL_150, Table 16). Chang et al. [133] employed a combination of zeolite USY and H$_2$SO$_4$ for optimizing, through a Box–Behnken experimental design, the ethanolysis of a cellulosic waste deriving from the furfural factory, achieving the highest EL yield of 19 wt%, working at 219 °C for 107 min. (run EL_151, Table 16). Guan et al. [168] compared the catalytic performances of three ionic liquids ([BMIM][Cl], [BMIM][HSO4] and [HSO$_3$-BMIM][HSO$_4$]) for the conversion of wheat straw, and selected [HSO$_3$-BMIM][HSO$_4$] thanks to its higher efficiency (runs EL_152, Table 16). An in-depth investigation of the synthesis of EL with ionic liquids, starting from oil palm empty fruit bunch (OPEFB) and oil palm mesocarp fiber (OPMF), was carried out by Tiong et al. [169]. The authors developed a one-pot cascade approach, which provided (i) a depolymerization step, which was carried out in the presence of the ionic liquid at 160 °C for 5 h in a 20 wt% of aqueous EtOH and (ii) an esterification step in EtOH excess, working under reflux at 90 °C for 12 h (runs EL_153–EL_154, Table 16). This study demonstrated that [HMIM][HSO$_4$] was the most efficient catalyst, leading to the best EL yields of 12 and 14 wt%, starting from OPEFB and OPMF, respectively. The same authors added the inorganic salt InCl$_3$ to the ionic liquid [HMIM][HSO$_4$] and performed the reaction carrying out the depolymerization step with InCl$_3$-[HMIM][HSO$_4$] at 160 °C for 3 h in a 20 wt% aqueous–EtOH solution and the esterification step in EtOH excess, under reflux at 90 °C for 10 h (runs EL_155–EL_156, Table 16) [170]. The addition of InCl$_3$ improved the reaction, weakening the glycosidic bonds of polysaccharides, thus promoting the depolymerization and conversion of cellulose to levulinic acid, the direct EL precursor. By this way, higher EL yields of 13 and 15 wt%were obtained starting from OPEFB and OPMF, respectively, after shorter reaction time, employing InCl$_3$-[HMIM][HSO$_4$] as the catalytic system. Lastly, the authors optimized both the depolymerization and esterification steps, through the central composite design (CCD) model, achieving the maximum EL yields of 19 and 21 wt%, starting from OPEFB and OPMF, respectively (runs EL_157–EL_158, Table 16) [171]. Liu et al. [172] employed the ionic liquid [HSO$_3$-BMIM][HSO$_4$] together with Al$_2$(SO$_4$)$_3$ as catalyst for the MW-assisted EL synthesis from ball-milled corn stover. When Al$_2$(SO$_4$)$_3$ was applied as the only catalyst, under the adopted reaction conditions ethyl-D-glucoside was the main product, achieving a yield up to 16 wt%, whilst levoglucosenone yield up to 4 wt% was obtained with [HSO$_3$-BMIM][HSO$_4$], due to the strong acidity of the ionic liquid. Instead, the combined use of the ionic liquid and inorganic salt showed a synergistic positive effect, leading to the highest EL yield of 11 wt% (run EL_159, Table 16).

### 3.3. PL, BL and Longer-Chain AL (PeL and HL) Synthesisfrom Real Biomass

Up to now, the most investigated ALs from real biomass are ML and EL, whilst few data have been reported for the synthesis of PL, BL, PeL, and HL. Table 17 summarized these data.

Table 17. Catalysts for the PL, BL, PeL, and HL production from real biomasses.

| Entry | Feedstock | Catalyst | Cat.(g)/Alcohol (g) [1] | T (°C) | t (h) | Heat [2] | Y (wt%) | Ref. |
|---|---|---|---|---|---|---|---|---|
| PL_23 | Coniferous wood | 2-NSA [3] | 0.20/8.0 (*i*) [4] | 200 | 4 | Conv. | 46 | [167] |
| PL_24 | Duckweed | [C$_3$H$_6$SO$_3$HPy]HSO$_4$ [5] | 0.80/32.0 (*n*) [6] | 170 | 5 | Conv. | 20 | [156] |
| BL_30 | Softwood Kraft pulp | H$_2$SO$_4$(100 wt%) | 42.86/100.0 (*n*) [7] | 117 | 3 | Conv. | 53 | [149] |
| BL_31 | Hardwood Kraft pulp | H$_2$SO$_4$(100 wt%) | 42.86/100.0 (*n*) [7] | 117 | 3 | Conv. | 43 | [149] |
| BL_32 | Papermaking sludge "A" | H$_2$SO$_4$(100 wt%) | 42.86/100.0 (*n*) [7] | 117 | 3 | Conv. | 11 | [149] |
| BL_33 | ADW *Eucalyptus Nitens* [8] | H$_2$SO$_4$(100 wt%) | 0.10/3.9 (*n*) [7] | 183 | 2.4 | MW | 38 | [52] |
| BL_34 | *Arundo donax* | H$_2$SO$_4$(100 wt%) | 0.10/6.0 (*n*) [7] | 190 | 0.25 | MW | 37 | [173] |
| BL_35 | Coniferous wood | 2-NSA [3] | 0.20/8.1 (*n*) [7] | 200 | 4 | Conv. | 43 | [167] |
| PeL_3 | Softwood Kraft pulp | H$_2$SO$_4$(100 wt%) | 25.4/101.8 | 138 | 3 | Conv. | 62 | [149] |
| PeL_4 | Hardwood Kraft pulp | H$_2$SO$_4$(100 wt%) | 25.4/101.8 | 138 | 3 | Conv. | 56 | [149] |
| PeL_5 | Papermaking sludge "G" | H$_2$SO$_4$(100 wt%) | 25.4/101.8 | 138 | 3 | Conv. | 45 | [149] |
| HL_3 | Softwood Kraft pulp | H$_2$SO$_4$(100 wt%) | 25.4/101.8 | 157 | 1 | Conv. | 59 | [149] |
| HL_4 | Hardwood Kraft pulp | H$_2$SO$_4$(100 wt%) | 25.4/101.8 | 157 | 1 | Conv. | 54 | [149] |
| HL_5 | Papermaking sludge "G" | H$_2$SO$_4$(100 wt%) | 25.4/101.8 | 157 | 1 | Conv. | 42 | [149] |

[1] The amounts of catalyst and the respective alcohol have been normalized to 1 g of feedstock. [2] "Conv." and "MW" stand for "Conventional" and "Microwave", respectively. [3] "2-NSA" stands for "2-naphthalenesulfonic acid". [4] "(*i*)" stands for "*iso*-propanol". [5] "[C$_3$H$_6$SO$_3$HPy]HSO$_4$" stands for "1-(3-sulfopropyl)-pyridinium bisulfate". [6] "(*n*)" stands for "*n*-propanol". [7] "(*n*)" stands for "*n*-butanol". [8] "ADW" stands for "autohydrolyzed-delignified wood".

Bianchi et al. [167] proposed the use of 2-NSA as a homogeneous catalyst for the synthesis of *i*-PL from coniferous wood, obtaining a promising PL yield of 46 wt% (run PL_23, Table 17). Moreover, Chen et al. [156] directly synthesized *n*-PL from duckweed biomass, employing the ionic liquid [$C_3H_6SO_3HPy$]$HSO_4$ as the reaction catalyst (run PL_24, Table 17). Under the best reaction conditions, the authors achieved the maximum *n*-PL yield of 20 wt%, which was significantly lower than those that were achieved by the authors with shorter-chain alcohols, such as MeOH and EtOH.

Few data are available for the conversion of real biomasses to BL. Sulfuric acid has been the almost exclusively used catalyst, highlighting that the interest in the heterogeneous catalyst is rather limited by the difficulties of performing out this reaction and achieving satisfactory yields. Some interesting biomasses derivedfrom the papermaking production were tested for this reaction, in the presence of sulfuric acid as the reaction catalyst, achieving BL yields in the range 40–50 wt%, under the best circumstances (runs BL_30—BL_32, Table 17) [149]. However, the authors' choice of employing a low reaction temperature (117 °C) involved the requirement of a very high acid concentration, about 3.5 mol/L, which is not sustainable for a sustainable industrial process. To solve this drawback, a higher reaction temperature should be preferred, thus allowing the use of lower acid concentrations. Very recently, Antonetti et al. [52] and Raspolli Galletti et al. [173] preferred this last approach, carrying out the MW-assisted butanolysis of ADW *Eucalyptus Nitens* and *Arundo donax*, respectively. In particular, the former optimized the reaction with a multivariate approach, achieving a maximum BL yield of about 40 wt% (run BL_33, Table 17), in agreement with the other best BL data that were reported in Table 17. Lastly, in a patent of Bianchi et al. [167], coniferous wood was converted in the presence of dilute 2-NSA (0.1 mol/L), also, in this case, achieving a good BL yield (run BL_35, Table 17).

As in the previous cases, also for the production of PeL and HL, sulfuric acid has been the only adopted catalyst. To the best of our knowledge, only Yamada et al. [149] havedirectly considered the synthesis of these two longer-chain ALfrom real biomasses (Table 17). The authors carried out the alcoholysis of different biomasses, softwood kraft pulp, hardwood kraft pulp, some papermaking sludges, by refluxing PeOH and HeOH at their boiling points, in the presence of highly concentrated acid, and optimizing the reaction time (runs PeL_3–PeL_5, Table 17; runs HL_3–HL_5, Table 17). For both of these ALs, the highest yields were ascertained starting from the softwood kraft pulp and the yield from the different substrates followed the trend: softwood kraft pulp > hardwood kraft pulp > papermaking sludge "G". This was mainly due to the very different hexose content of the three biomasses that was about 90, 80, and 50 wt% for softwood kraft pulp, hardwood kraft pulp, and papermaking sludge "G", respectively.

## 4. Considerations on the Catalysis Issues, Main Process Bottlenecks and Improvable Aspects

It is necessary to more critically consider the overall available data and, consequently, evaluate the most appropriate choices, especially those related to the target als, the starting feedstocks, and the catalysts of greatest interest, in order to identify the main bottlenecks and improvable aspects of the alcoholysis process.

Regarding the choice of the most suitable ALs, the comparison among the available data from both model and real feedstocks confirms that ML and EL are undoubtedly the most studied esters and, consequently, the most successful candidates for the possible development of valuable applications in the immediate future. The first pioneering work of Silva et al. [17] on the economic feasibility of the EL production on a greater scale, together with the acknowledged use of ML and EL as bio-additives for improving the properties of different transportation fuels [47], indirectly confirm this tendency. In particular, Tian et al. [174] reported that blends of ML or EL (10 vol.%) in Euro 95 gasoline have a superior anti-knock quality to the reference Euro 95 gasoline. However, even if the antiknock index of ML is high, its full miscibility with water and separation from gasoline at cold temperatures makes practically disadvantageous its use for this purpose [47]. Additionally, in the field of diesel fuels, the use of EL as bio-based cold-flow improvers in bio-diesel fuels has been demonstrated [47,175], but hindered by miscibility issues [176]. In order to further increase the AL miscibility in diesel

fuels, the attention isnow moving rather towards higher molecular-weight alcohols. In this context, the use of PL as a fuel-blender does not lead to remarkable advantages on fuel properties, and this poor interest is indirectly confirmed by the few available catalytic data, whilst that of BL has recently aroused great interest, allowing for cleaner combustion, mainly in terms of low CO and soot emissions [52,53]. Therefore, longer-chain ALs are certainly more appropriate for diesel-fuel blends [177], but, according to the available data, their syntheses are more difficult, being realized mostly starting from the expensive pure levulinic acid, thus making more appropriate the use of these ALs for niche applications, rather than for high-volume ones. Definitely, among the possible candidates, EL and BL represent the most promising ALs for the real development of high-volume fuel applications, already in the immediate future.

The choice of the appropriate starting feedstock is fundamental and strategic for improving the alcoholysis process, for the exploitation of the C6 fraction of biomass. Most of the available data reported in this review have been obtained adopting model C6 feedstocks, especially soluble mono- and disaccharides, with loadings that are generally below 4 wt%. The preference of soluble C6 mono- and disaccharides is certainly convenient for demonstrating the good performances ofad hocsynthesized catalysts, thus minimizing substrate-catalyst mass transfer problems. Besides, recycling tests of the synthesized heterogeneous catalysts generally provide satisfactory results, because the issue of solid humin formation for the alcoholysis of these model carbohydrates is not sorelevant as in the hydrothermal process. Therefore, the use of these convenient feedstocks minimizes the char formation, enabling a more agile recovery of the catalysts, which generally maintain almost unaltered their physicochemical properties, after many recycling runs. Anyway, from an applicative perspective, the adoption of these model feedstocks is concretely unsustainable for the development of this process intensification, which should, instead, prefer the use of cheap or, even better, waste real biomasses: this is certainly the main bottleneck to face for the development of the alcoholysis process [61]. Already moving towards the more complex model cellulose, the issue of by-product formation is significant, and this issue becomes much more considerable by using the real lignocellulosic biomass, where the final solid residue includes the contribution of both humin and lignin components. For these reasons, catalyst heterogenization becomes a difficult issue to solve, also taking into account that the new catalysts should have increasingly satisfactory performances and maintain prolonged recyclability. Based on these considerations, in the case of real biomass alcoholysis, the use of very dilute mineral acids (in particular $H_2SO_4$) is preferred over that of heterogeneous catalysts, whose application is, instead, almost limited to AL synthesis from simple model carbohydrates. The range of the catalyst loadingadopted for the conversion of model compounds and real biomasses is wide (0.02–30 wt%), with the chosen value depending on the type of catalyst and the reaction conditions. Lower catalyst loadings are generally used for the mineral acids, requiring a higher reaction temperature to achieve the highest AL yield, approximately 50–60 mol% in the most difficult cases, e.g., starting from the recalcitrant cellulose or real biomass. The preference of the homogeneous acids for the alcoholysis of real biomass also favors the use of higher feedstock loadings (6–10 wt%), ascomparred with those of the model compounds. The use of higher loadings is strategic for increasing the AL concentration in the reaction mixture, in agreement with the *high gravity* approach [63], reducing the AL separation costs and making the whole process economically more profitable.

According to the available data, the chemical properties of the catalyst can be adjusted and improved. Alkyl lactate and 1,1,2-trialkoxyethane are formed with an excess of Lewis acidity, while humins with an excess of Brønsted or Lewis acids. In addition to these reaction by-products, the preference of a very low concentration of Brønsted acids is of paramount importance for limiting the dialkyl ether formation, which is responsible for a significant consumption and loss of the solvent, with dialkyl ether yields up to 60 mol% [41,43]. Anyway, a limited formation of dialkyl ether is allowed, thanks to its valuable fuel properties [178], and its separation from the reaction mixture is generally simple. The proper balance between Brønsted and Lewis acidity appears to be of paramount importance for maximizing the AL production and, consequently, minimizing that of by-products.

Many authors have strategically chosen fructose as starting feedstock, avoiding the Lewis-catalyzed isomerization from glucose to fructose, thus proposing Brønsted acids and obtaining very high yields in AL. Again, this solution is academically convenient, but industrially uninteresting, whilst the use of real feedstocks must take the Lewis-catalysed stepsinto account. Taking the alcoholysis of the model celluloseinto account, which is the most difficult model substrate to convert into ALs, the combined use of traditional Brønsted acids (such as $H_2SO_4$, PTSA, NSA, or $H_3PO_4$) and Lewis acids, in particular aluminum derivatives (e.g., $Al_2(SO_4)_3$, $Al(acac)_3$, $Al(OEt)_3$, $Al(OH)_3$) [79], and metal triflates (e.g., $In(OTf)_3$ or $Y(OTf)_3$) [27,122], is particularly promising, given the very high AL yields (70–75 mol%), if compared with the sole Brønsted acid (50–60 mol%, for the synthesis of ALs with the most efficient $H_2SO_4$). In this context, it is interesting that already the Brønsted–Lewis acidity of the sole $Al_2(SO_4)_3$ is enoughto achieve the highest AL yields (about 70 mol% for ML and EL) [32], thus avoiding the addition of mineral acid as co-catalyst, and achieving considerable advantages on the separation/recovery of the spent catalyst. Eventually, it is possible to further modulate the Brønsted–Lewis acidity of $Al_2(SO_4)_3$,by using other metal salts as co-catalysts, such as with the system [$Al_2(SO_4)_3$+$Fe_2(SO_4)_3$], adopted for the BL synthesis from model cellulose, reaching performances similar to those of sulfuric acid (40 mol%, in both cases) [146]. Lastly, the ML synthesis in the presence of a sulfonated hydrochar as the heterogeneous catalyst appears interesting [85]. Despite the reported ML yield from cellulose is not particularly high (30 mol%) and the bio-char was obtained from a different process (biomass pyrolysis), the possibility of sulfonating the char recovered from the same solvothermal processes deserves further research and development, allowing for the valorization of waste by-products produced within the same process, rather than using more costly acid sulfonic resins, thus supporting the biorefinery and the intensification concepts [61]. In this context, the addition of Lewis acids as the reaction co-catalysts, should also be evaluated in this case, extending the research towards the synthesis of longer-chain ALs.

Moving towards the alcoholysis of real biomasses, the comparison of the catalyst performances becomes more difficult, due to the different chemical composition of the starting biomass feedstocks, which is often untold. As a general consideration, the use of diluted Brønsted acids (in particular $H_2SO_4$, PTSA, and BSA), together with Lewis ones (especially Al salts, e.g., $Al_2(SO_4)_3$, $Al(acac)_3$, or $AlCl_3$, but also $CuSO_4$), actually represents the best and simplest solution for maximizing the production of short-chain ALs, e.g., ML and EL, achieving similar yields to those that were obtained with the traditional $H_2SO_4$, generally in the range 20–30 wt%. Eventually, the sole use of inorganic salts, endowed with Lewis/Brønsted acidity, deserves further investigation for this one-pot approach, given the acceptable AL yields reported in the literature [55,74], thus completely avoiding the use of $H_2SO_4$, and preferring the conversion of biomasses thatare rich in cellulose and poor in lignin. When longer-chain ALs are desired, the one-pot alcoholysis of the real biomass requires stronger acid catalysts, which is why $H_2SO_4$ is still the preferred catalyst, thus giving less importance to the careful balance of the Brønsted–Lewis acidity. Moreover, the addition of any pretreatment steps, for example, to improve the accessibility of the cellulose and/or fractionate the biomass components (e.g., mechanical ball-milling, organosolv, or ionic liquid pre-treatments), should lead to an improvement of the AL yields, by this way compensating for the added costs. However, the reported data for the different ALs clearly show that, up to now, these pre-treatments do not lead to such striking improvements, further supporting the proposed simpler choices, e.g., the preference of the direct biomass conversion, aided by cheap diluted and tunable catalysts. Moreover, especially in the case of real low-cost or waste biomass, the unjustified addition of expensive treatments would create evident contradictions.

## 5. Conclusions

Alkyl levulinates are valuable chemicals having strong market potential, mainly as oxygenated bio-fuels. However, in order to enhance their production on a larger industrial scale, it is necessary to start directly from cheap precursors, such as C6 carbohydrates and, even better, lignocellulosic or waste biomasses, rather than from more expensive and pure levulinic acid. Therefore, a direct cascade approach should allow to postpone and reduce the number of purification steps, with remarkable

advantages on the alkyl levulinate yield and, more generally, on the total costs of the process. For these prime reasons, this review has been focused on the use of model C6 carbohydrates and real biomasses in the direct alcoholysis to give different levulinates, dealing with the most interesting and recent advances in catalysis. Diluted sulfuric acid results the most performing catalyst for the alkyl levulinate production, leading to the highest yield of approximately 90 mol%, and the reference for the development of new catalysts. The use of model C6 carbohydrates has been preferred by researchers to demonstrate the effectiveness of ad hoc synthesized catalysts, which are often very elegant, but too expensive and of little practical use. In this context, the choice of fructose as the starting substrate makes it possible to simplify the catalysis, which is mainly of Brønsted-acid type, thus generally achieving very high yields, whilst glucose and its polymers require an additional isomerization step, as catalyzed by Lewis acids. The tunability between these different acidities has been the subject of many studies and, among many proposed catalytic systems, the use of metal salts, especially $Al_2(SO_4)_3$, alone or in combination with very diluted mineral acids, represents the best compromise between catalytic performances and costs. These catalysts result in also being attractive for the alcoholysis of the more recalcitrant cellulose and even for that of real biomass, reaching performances that are similar to that of the traditional sulfuric acid, whilst other systems, such as heteropolyacids, sulphonic acids, zeolites, clays, sulfated metal oxides, and properly modified catalysts, are still poor performing for the alcoholysis of these tough substrates. Moreover, in the case of the real biomass alcoholysis, the issue of recovery/recycling of the heterogeneous catalyst becomes even more problematic, due to the co-presence of char, which includes carbonaceous degradation products, mainly humins and lignin. In principle, the use of sulfonated char, which is produced within the same solvothermal process, as a heterogeneous catalyst for the alcoholysis reaction, could satisfy the concepts of sustainability and biorefinery, but low alkyl levulinate yield from soluble carbohydrates hinders its use for the biomass alcoholysis shortly. On this basis, dilute sulfuric acid remains the preferred choice for the alcoholysis of real substrates, leading to yields of up to about 60 wt%. To favor the intensification of the process, the conversion of cellulose-rich biomasses, such as wastes derived from the papermaking process, should be preferred for improving the yield to alkyl levulinates, being less recalcitrant towards the alcoholysis. Moreover, the use of cellulose-rich feedstocks, which do not include the lignin component, is particularly appropriate for improving the employment of heterogeneous catalysts, simplifying their recovery. Therefore, the topic is surely of great interest and offers real prospects for industrial applications, but it is necessary to deepen the research by both proposing appropriate feedstocks and carefully tuning the adopted catalytic systems and reaction conditions.

**Author Contributions:** All authors equally contributed to the preparation of this manuscript. All authors have read and agreed to the published version of the manuscript.

**Funding:** This research was funded by MIUR in the framework of the research project VISION PRIN 2017, grant number FWC3WC_002. Fondazione Cassa di Risparmio di Lucca funded this research in the framework of the research project GREENFIBER.

**Conflicts of Interest:** The authors declare no conflict of interest.

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
