# Peer review of "Direct Alcoholysis of Carbohydrate Precursors and Real Cellulosic Biomasses to Alkyl Levulinates: A Critical Review"

_catalysts, doi:10.3390/catal10101221_

Round 1

Reviewer 1 Report

The work presented by the authors is very well documented, and presents a very exquisite comparison of other works, however it is an extremely extensive work, the amount of impressive tables, they are fine but the amount I think is exaggerated. The work should reduce the number of tables, to make a reading a little more compact. On the other hand, in Figure 3 the image should be improved, by improving the contrast the information can be better appreciated for the reader. In conclusion good work but too extensive.

Reviewer 2 Report

This review deals with the direct alcoholysis of carbohydrate precursors to alkyl levulinates. The manuscript is comprehensive and includes almost publications in this research field. The paper is well written. However, some small changes are needed or should be considered for improving. Below the authors will find some comments:

In general, the introduction section has not a completely clear structure and there are small repetitions. Please consider subsection. These different subsections could help to show the structure and the red thread of the introduction.

Who did draw these Figures? Is a permission necessary or did you collect the information from different papers and drew these Figures or is a citation of the different papers needed?

There is no connection between the text and Figure 1. Please insert e.g. Figure 1 also in the text.

Line 75: Which Figure did you mean with “above Figure”?

Line 150: The term “ML” was not defined before. Please mention, what do you mean with this term (e.g. methyl levulinate)?

Line 1101: I guess that it should be “é” instead of “e” in Demolis et al. [149].

Line 1290 to 1292: Please check the journal guide lines for citations. There are two citations in one sentence, e.g. Yang et al. [160] proposed the conversion … (run EL_126, Table 17) [160].

Please check also the line 1435 to 1436 concerning the citation.

Please check the journal guide lines for the citation of webpages, Line 1626 to 1630; In my opinion, you could also add Anonymous (2020) Ethyl levulinate market size….if you do not know the author of this webpage.

Round 2

Reviewer 1 Report

I have seen the corrections made by the authors, I still think that it is a very long article but very well documented. I would recommend your publication